# TeamTR: Trust-Region Fine-Tuning for Multi-Agent LLM Coordination

**Yi Xie** [1] **Siao Liu** [2] **Falong Fan** [1] **Yuanqi Yao** [3] **Siyang Cao** [1] **Yue Zhao** [4] **Bo Liu** [1]

## Abstract

Multi-agent LLM systems have shown promise for complex reasoning, yet recent evaluations reveal they often underperform single-model baselines. We identify a structural failure mode in sequential fine-tuning of shared-context teams: updating one agent shifts the team's context distribution, and when subsequent updates are evaluated on cached rollouts, this mismatch compounds. We formalize this as the *compounding occupancy shift* and prove that stale-occupancy evaluation incurs a penalty that scales quadratically with the number of agents. In contrast, intermediate-occupancy evaluation reduces this to linear scaling. We propose *TeamTR*, a trust-region framework that resamples trajectories after each component update and enforces per-agent divergence control, yielding rigorous per-update and per-stage improvement lower bounds. Experiments show that TeamTR outperforms single-agent and sequential baselines with 7.1% on average, mitigates coordination regressions, and supports plug-and-play component replacement. Code is available at https://github.com /Yydc/TeamTR.

## 1. Introduction

Multi-agent LLM systems coordinate role-specialized components for complex reasoning and task execution (Yao et al., 2022; Wu et al., 2023; Hong et al., 2023; Du et al., 2023). Despite their success, recent evaluations reveal that such teams often underperform single strong models with best-of-$N$ sampling (Kim et al., 2025). These failures are attributed to suboptimal coordination protocols (Cemri et al., 2025). We identify that the training process of the MA-LLMs sys-

tem can introduce this bias and undermine coordination.

Most multi-agent LLM systems are *shared-context* teams, in which agents interact turn by turn over a common textual state (Wu et al., 2023; Hong et al., 2023). Fine-tuning such teams may proceed via joint updates (all agents simultaneously) or sequential updates (one agent at a time). The instability issues of joint updates are known in MARL, where coupled policy changes make optimization difficult to control (Foerster et al., 2017; Kuba et al., 2021) and also MA-LLMs (Liu et al., 2025a) (Fig. 1, left). Sequential training offers a more stable alternative and is increasingly adopted for modular optimization (Subramaniam et al., 2025). However, sequential methods introduce a failure mode: each update shifts the context distribution seen by subsequent agents, and when rollouts are cached at stage start to reduce sampling cost, this mismatch compounds (Fig. 1, middle).

We formalize this as *compounding occupancy shift*: when later agents are updated using rollouts collected before earlier agents were updated, the resulting distribution mismatch accumulates across the update sequence. Under per-step trust regions of radii $\{\delta_i\}_{i=1}^n$, stale-occupancy evaluation incurs a penalty scaling as $O(n^2\sqrt{\bar{\delta}})$, whereas intermediate-occupancy evaluation reduces this to $O(n\sqrt{\bar{\delta}})$. This gap explains why naive sequential fine-tuning can regress coordination even when each update appears locally beneficial.

To address this, we propose *TeamTR*, a stage-wise trust-region framework that resamples trajectories after each component update. This ensures that each agent is trained under the distribution induced by the partially updated team, thereby eliminating the stale-occupancy penalty. Per-agent trust regions control the occupancy drift introduced by each update, keeping the distribution shift bounded. TeamTR yields rigorous lower bounds on improvement: surrogate gain minus explicit penalties for occupancy shift and estimation error. The certificate applies to any update order, and penalties can be tracked empirically in the training process.

We instantiate TeamTR under a turn-taking protocol where one agent is active per step. This yields a factorization: team-level divergence reduces to single-agent divergence on visited states, enabling tractable trust regions. A token decomposable behavior-to-updated KL yields a monitorable trust region constraint, estimated from on-policy rollouts via sampled log-probability differences. Empirically, TeamTR

[1]Department of Electrical & Computer Engineering, University of Arizona [2]Future Science and Engineering College, Soochow University [3]INSAIT, Sofia University "St. Kliment Ohridski" [4]Department of Electrical and Computer Engineering, Stony Brook University. Correspondence to: Bo Liu <boliu@arizona.edu>.

*Proceedings of the 43rd International Conference on Machine Learning*, Seoul, South Korea. PMLR 306, 2026. Copyright 2026 by the author(s).

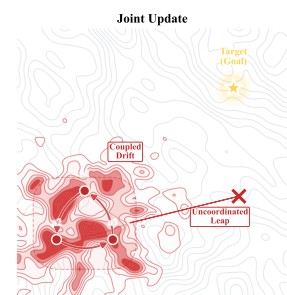 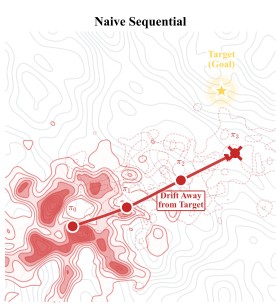 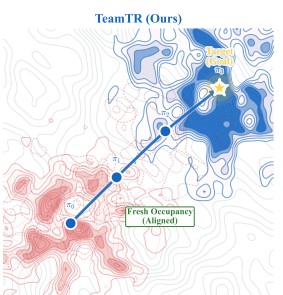 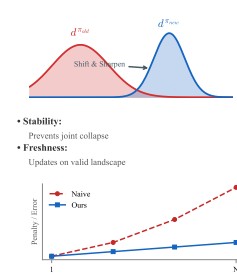

*Figure 1.* Update trajectories on the team objective landscape. **Left**: Joint updates suffer from coupled drift and uncoordinated leaps. **Middle**: Naive sequential updates with cached rollouts drift away from the target due to stale occupancy. **Right**: TeamTR resamples under fresh occupancy after each update, reaching the target stably. **Inset**: Occupancy distributions (top), penalty term scaling (bottom).

improves performance relative to sequential baselines, stabilizes coordination, and supports plug-and-play of components. Our contribution can be concluded as follows:

- We formalize the *compounding occupancy shift* in shared-context multi-LLM fine-tuning and prove that stale-occupancy evaluation incurs an $O(n^2)$ penalty (Sec.3).

- We propose TeamTR, which reduces this penalty to $O(n)$ via intermediate-occupancy evaluation and provides rigorous per-update improvement lower bounds (Sec.4).

- We empirically validate that TeamTR outperforms existing single and multi-agent baselines by stabilizing training and enabling components to be plug-and-play (Sec.5).

## 2. Related Work

**Multi-agent LLM systems.** Multi-agent LLM systems have evolved along two axes: inference-time orchestration and training-time coordination. Inference-time approaches deploy frozen models under hand-crafted protocols, including debate and consensus frameworks (Du et al., 2023; Liang et al., 2024), structured role-play pipelines (Hong et al., 2023; Qian et al., 2024), and general-purpose orchestration libraries (Wu et al., 2024). Training-time approaches aim to internalize coordination through supervised fine-tuning on interaction trajectories (Xie et al., 2026; Zeng et al., 2024) or reinforcement learning with social or preference signals (Liu et al., 2023a; Lee et al., 2023). These methods typically assume frozen components or joint/independent training; none explicitly addresses the occupancy shift that arises under sequential fine-tuning.

**Trust regions in multi-agent RL.** Trust-region methods guarantee monotonic improvement by constraining policy divergence (Schulman et al., 2015). Extending these guarantees to multi-agent LLMs settings is nontrivial due to non-stationarity from simultaneous updates. HATRPO and HAPPO (Kuba et al., 2021) derive a multi-agent advantage decomposition showing that sequential updates preserve monotonic improvement in heterogeneous teams, but operate on low-dimensional continuous control and do not

address the autoregressive, token-level structure of LLM message spaces. TeamTR adapts insights to shared-context LLM teams by defining trust regions based on token-level KL divergences computed from on-policy rollouts.

**Distribution shift and modular evolution.** Distribution shift under sequential updates is a known challenge in both MARL (Foerster et al., 2017) and RLHF (Casper et al., 2023), often addressed via importance sampling or replay buffers. For modular system evolution, model-merging techniques (Ilharco et al., 2022; Wortsman et al., 2022) compose capabilities by manipulating weight vectors but assume independent training distributions and static composition. TeamTR provides a dynamic alternative: it bounds the distribution shift induced by each sequential update via intermediate-occupancy trust regions, yielding theoretically guaranteed improvement that extends to agent plug-and-play. Extended discussions are provided in Appendix A.

## 3. Theoretical Framework

We develop a framework for fine-tuning multi-agent LLM teams under sequential component updates. The central challenge is *compounding occupancy shift*: each update changes the team's state distribution, and reusing for next agent with stale rollouts incurs an additional certificate penalty term that scales as $O(n^2\sqrt{\bar{\delta}})$ (for fixed trust region radii), where $n$ is the number of agents. Our analysis identifies this failure mode and shows intermediate-occupancy evaluation reduces the dominant penalty to $O(n\sqrt{\bar{\delta}})$.

### 3.1. Shared-context team as a message-action MDP

We model team execution as a discounted MDP $\mathcal{M} = (\mathcal{S}, \{\mathcal{A}_j\}_{j=1}^n, P, r, \gamma)$ with $\gamma \in (0, 1)$ and bounded reward $|r| \leq R_{\max}$. The state $s \in \mathcal{S}$ is the shared textual context (the prompt plus accumulated messages) and can optionally include the active agent's ID selected by a router. We treat the router as fixed and fold it into the environment dynamics. Each agent $j$ chooses a macro-action $a_j \in \mathcal{A}_j$ corresponding to a message (token sequence), and the team policy

factorizes as $\pi(\mathbf{a} \mid s) = \prod_{j=1}^{n} \pi^{(j)}(a_j \mid s)$. We focus on the turn-taking protocol: at each decision step, exactly one agent is active and emits a message that augments the shared context; all other agents take a fixed no-op action.

**Lemma 3.1** (Active-factor reduction in turn taking protocol). *Assume turn-taking protocol: for each state $s$, there is an active agent index $j(s)$ such that all inactive agents deterministically take noop under both $\pi$ and $\pi'$. Then for any two team policies $\pi', \pi$,*

$$D_{\mathrm{KL}}\big(\pi'(\cdot \mid s)\|\pi(\cdot \mid s)\big) = D_{\mathrm{KL}}\big(\pi'^{(j(s))}(\cdot \mid s)\|\pi^{(j(s))}(\cdot \mid s)\big),$$

*and expectations about functions that depend only on the active action reduce to expectations about the active agent.*

Lemma 3.1 addresses a practical obstacle: a team-level trust region over the joint action space $\prod_j \mathcal{A}_j$ is intractable for LLM message distributions, while the turn-taking protocol reduces the divergence constraint to a single-agent quantity. The detailed proof can be found in Appendix B.1. We study within-stage sequential updates. Let $\pi_{\mathrm{cur}}$ be the team at stage start and $\sigma$ an update order (any permutation). Denote by $\pi[j \leftarrow \pi']$ the team obtained by replacing agent $j$ in $\pi$ with $\pi'$. We define the intermediate policies as follows:

$$\pi_{\mathrm{cur}} = \hat{\pi}^0 \xrightarrow[\mathrm{Update}]{\substack{\sigma(1)\leftarrow \\ \pi_{\mathrm{tar}}^{\sigma(1)}}} \hat{\pi}^1 \to \cdots \to \hat{\pi}^{n-1} \xrightarrow[\mathrm{Update}]{\substack{\sigma(n)\leftarrow \\ \pi_{\mathrm{tar}}^{\sigma(n)}}} \hat{\pi}^n = \bar{\pi}$$

where $\pi_{\mathrm{tar}}^{\sigma(i)}$ denotes the updated policy for agent $\sigma(i)$ obtained by (approximately) maximizing the surrogate objective in Section 3.3. Let $d^\pi$ be the discounted occupancy induced by $\pi$, and $J(\pi)$ its discounted return. Within each stage, each agent is updated at most once; thus at step $i$ the pre-update policy for agent $j = \sigma(i)$ is the corresponding factor in $\hat{\pi}^{i-1}$, which we denote by $\pi_{\mathrm{cur}}^{\sigma(i)}$ for brevity.

### 3.2. Token-decomposed trust regions

To control occupancy shift, we constrain the *behavior-to-updated* KL divergence, which can be directly estimated from trajectories generated by the pre-update policy, without requiring policy sampling from the updated one.

For a reference policy $\rho$ and policies $\pi, \pi'$, we define:

$$D_{\mathrm{KL}\mathrm{tok}}^{\rho}(\pi\|\pi') := \mathbb{E}_{s\sim d^\rho} D_{\mathrm{KL}}\big(\pi(\cdot \mid s)\|\pi'(\cdot \mid s)\big).$$

Here $\mathrm{tok}$ emphasizes *token-decomposability* for autoregressive messages: the message-level KL decomposes exactly into token-level KLs via the chain rule. Let $\pi_u := \pi^{(j)}(\cdot \mid s, x_{<u})$ and $\pi'_u := \pi'^{(j)}(\cdot \mid s, x_{<u})$. Then we have:

$$D_{\mathrm{KL}}\big(\pi^{(j)}(\cdot|s) \| \pi'^{(j)}(\cdot|s)\big) = \mathbb{E}_{\substack{m\sim \\ \pi^{(j)}(\cdot|s)}} \sum_{u=1}^{T(m)} D_{\mathrm{KL}}(\pi_u \| \pi'_u), \tag{1}$$

where $T(m)$ is the (random) message length. Computing each token-level KL term exactly requires a full-vocabulary sum; in practice, we therefore use a sampled estimator based on token log-probability differences along rollouts from the behavior policy (Details in Appendix B.2 and F.6).

At update step $i$, TeamTR constrains the updated factor:

$$D_{\mathrm{KL}\mathrm{tok}}^{\hat{\pi}^{i-1}}\big(\pi_{\mathrm{cur}}^{\sigma(i)}\|\pi_{\mathrm{tar}}^{\sigma(i)}\big) \leq \delta_i. \tag{2}$$

The radius $\delta_i$ controls the allowed per-step policy change: smaller $\delta_i$ reduces occupancy-shift penalties (which scale as $\sqrt{\delta_i}$) but restricts policy movement. Since each within-stage step updates a single factor under the turn-taking protocol, Eq. (2) also implies a team-level step bound $D_{\mathrm{KL}\mathrm{tok}}^{\hat{\pi}^{i-1}}(\hat{\pi}^{i-1}\|\hat{\pi}^i) \leq \delta_i$. (formalized in Lemma 3.5).

### 3.3. Intermediate-occupancy surrogates and error

Let $A^\pi(s, \mathbf{a})$ be the macro-action advantage. When updating agent $\sigma(i)$, we consider the population surrogate evaluated under the intermediate occupancy:

$$L_i^{\mathrm{seq}} = \frac{1}{1-\gamma} \mathbb{E}_{s\sim d^{\hat{\pi}^{i-1}}, \mathbf{a}\sim \hat{\pi}^i(\cdot|s)}\Big[\widehat{A}_{i-1}(s, \mathbf{a})\Big], \tag{3}$$

where $\widehat{A}_{i-1}$ is an estimator constructed from rollout data collected under $\hat{\pi}^{i-1}$. In practice, TeamTR approximately maximizes a clipped objective (see Appendix E.1 for a concrete instantiation); the certificate statements below hold for any realized update that satisfies the trust-region constraint. We track the mismatch between the estimator and the true advantage. Let $\mathcal{D}_{i-1}$ denote the rollout data used to construct $\widehat{A}_{i-1}$ under $\hat{\pi}^{i-1}$. Assume $|A^\pi(s, \mathbf{a})| \leq A_{\max}$ and enforce $|\widehat{A}_{i-1}| \leq A_{\max}$ via clipping, we define:

$$\zeta_i := \left| \mathbb{E}_{\substack{s\sim d^{\hat{\pi}^{i-1}} \\ \mathbf{a}\sim \hat{\pi}^i(\cdot|s)}} \Big[\mathbb{E}_{\mathcal{D}_{i-1}}\big[\widehat{A}_{i-1}(s, \mathbf{a})\big] - A^{\hat{\pi}^{i-1}}(s, \mathbf{a})\Big] \right|. \tag{4}$$

### 3.4. Occupancy-shift and improvement lower bounds

The following bound connects expected divergence to occupancy shift, explaining why trust regions yield explicit control over distribution mismatch. Proofs for this subsection are provided in Appendix B.3, B.4, C.1, C.2, and C.3.

**Lemma 3.2** (Occupancy shift under expected divergence). *For any policies $\pi', \pi$ and any bounded measurable $f$,*

$$\big|\mathbb{E}_{d^{\pi'}}[f] - \mathbb{E}_{d^\pi}[f]\big| \leq \frac{\sqrt{2}\gamma}{1-\gamma} \sqrt{D_{\mathrm{KL}\mathrm{tok}}^{\pi}(\pi\|\pi')} \, \|f\|_\infty.$$

**Proposition 3.3** (Quadratic-to-linear reduction). *Let $\hat{\pi}^0$ be the stage-start team and $\hat{\pi}^{i-1}$ the intermediate team before step $i$. Assume the per-step trust regions in Eq. (2). Then*

*for any bounded measurable $f$ and any $i \geq 1$,*

$$\left| \mathbb{E}_{d^{\hat{\pi}^{i-1}}}[f] - \mathbb{E}_{d^{\hat{\pi}^0}}[f] \right| \leq \frac{\sqrt{2}\gamma}{1-\gamma} \|f\|_\infty \sum_{k<i} \sqrt{\delta_k}. \quad (5)$$

*Then we can find the stale-occupancy surrogate by evaluating on stage-start occupancies:*

$$L_i^{\text{stale}} = \frac{1}{1-\gamma} \mathbb{E}_{s \sim d^{\hat{\pi}^0}, \mathbf{a} \sim \hat{\pi}^i(\cdot|s)} \left[ \widehat{A}_{i-1}(s, \mathbf{a}) \right]. \quad (6)$$

$$\left| L_i^{\text{seq}} - L_i^{\text{stale}} \right| \leq \frac{\sqrt{2}\gamma}{(1-\gamma)^2} A_{\max} \sum_{k<i} \sqrt{\delta_k}. \quad (7)$$

*Thus, stale-occupancy evaluation incurs a cumulative penalty of $O(n^2\sqrt{\bar{\delta}})$ with $\bar{\delta} = \max_k \delta_k$, whereas intermediate-occupancy evaluation achieves $O(n\sqrt{\bar{\delta}})$.*

**Theorem 3.4** (Single-step improvement lower bound)**.** *For step $i \in \{1, \ldots, n\}$, assume the trust region in Eq. (2). Then*

$$J(\hat{\pi}^i) - J(\hat{\pi}^{i-1}) \geq L_i^{\text{seq}} - \frac{\sqrt{2}\gamma}{(1-\gamma)^2} A_{\max} \sqrt{\delta_i} - \frac{\zeta_i}{1-\gamma}.$$

**Lemma 3.5** (Per-step KL reduction under factorized updates)**.** *The per-step reduction satisfies:*

$$D_{\text{KLtok}}^{\hat{\pi}^{i-1}}(\hat{\pi}^{i-1} \| \hat{\pi}^i) = D_{\text{KLtok}}^{\hat{\pi}^{i-1}}(\pi_{\text{cur}}^{\sigma(i)} \| \pi_{\text{tar}}^{\sigma(i)}) \leq \delta_i, \quad \forall i.$$

*Consequently, summing Theorem 3.4 over $i = 1, \ldots, n$ yields the stage-wise certificate in Theorem 3.6.*

**Theorem 3.6** (Stage-wise improvement lower bound)**.** *The improvement is bounded as follows:*

$$J(\bar{\pi}) - J(\pi_{\text{cur}}) \geq \sum_{i=1}^{n} L_i^{\text{seq}} \quad (under (2) \; \forall i)$$
$$- \frac{\sqrt{2}\gamma A_{\max}}{(1-\gamma)^2} \sum_{i=1}^{n} \sqrt{\delta_i} - \frac{1}{1-\gamma} \sum_{i=1}^{n} \zeta_i.$$

A high-probability empirical version (accounting for minibatch estimation and ratio clipping) is in Appendix E.3.

**Theorem 3.7** (Stage-wise bound under stale-occupancy surrogates)**.** *Using stale surrogates, we have:*

$$J(\bar{\pi}) - J(\pi_{\text{cur}}) \geq \sum_{i=1}^{n} \left( L_i^{\text{stale}} - \frac{\zeta_i}{1-\gamma} \right) \quad (under (2))$$
$$- \frac{\sqrt{2}\gamma A_{\max}}{(1-\gamma)^2} \sum_{i=1}^{n} \left( \sqrt{\delta_i} + \sum_{k<i} \sqrt{\delta_k} \right). \quad (8)$$

*For $\delta_i \equiv \bar{\delta}$, the stale-occupancy penalty scales as $\Theta(n^2\sqrt{\bar{\delta}})$, versus $\Theta(n\sqrt{\bar{\delta}})$ for intermediate-occupancy evaluation.*

---

**Algorithm 1** TeamTR: Stage-wise Sequential Fine-tuning

---

**Require:** Team $\pi_{\text{cur}} = \{\pi^{(j)}\}_{j=1}^{n}$, trust-region radii $\{\delta_i\}$, prompt distribution $\mathcal{D}$, group size $G$, router $\mathcal{R}$
**Ensure:** Updated team $\pi_{\text{cur}}$
 1: **for** stage $k = 1, 2, \ldots$ **do**
 2:     Sample batch $\mathcal{B} \subset \mathcal{D}$; choose update order $\sigma$
 3:     $\hat{\pi}^0 \leftarrow \pi_{\text{cur}}$
 4:     **for** $i = 1$ to $n$ **do**
 5:         Collect rollouts under $\hat{\pi}^{i-1}$ with router $\mathcal{R}$
 6:         Compute advantages $\tilde{A}$ via Eq. (9)
 7:         Update $\pi^{(\sigma(i))}$ via Eq. (10) until $\widehat{D_{\text{KLtok}}} \leq \delta_i$
 8:         $\hat{\pi}^i \leftarrow \hat{\pi}^{i-1}[\sigma(i) \leftarrow \pi_{\text{new}}^{(\sigma(i))}]$
 9:     **end for**
10:     $\pi_{\text{cur}} \leftarrow \hat{\pi}^n$
11: **end for**

---

### 3.5. Plug-and-play upgrades

The trust region is defined per agent and evaluated on the team's occupancy, plug-and-play replacement can be handled by aligning the new component within a trust region around the replaced agent, then resuming intermediate occupancy updates. The proofs are detailed in Appendix D.1.

**Proposition 3.8** (Certified resumability after replacement)**.** *Replacing agent $j$ by a new parameterization and then performing TeamTR updates that satisfy Eq. (2). Theorems 3.4–3.6 continue to hold for subsequent updates. This does not guarantee that the replacement step itself is non-decreasing in $J$; it only states that subsequent TeamTR updates preserve the same lower bound. Detailed in Appendix D.2.*

**Proposition 3.9** (Certificate tightening)**.** *If an upgraded agent achieves a higher surrogate value within the same radius, or the same surrogate value with a smaller radius, then the lower bound in Theorem 3.6 does not decrease.*

## 4. Algorithm

We instantiate TeamTR as a stage-wise training loop that implements the theoretical framework in Sec. 3. Each within-stage update resamples rollouts within the current intermediate team, optimizes a clipped surrogate objective, and enforces a per-agent trust region via a decomposable KL constraint. Algorithm 1 summarizes the procedure.

**Group-normalized advantages.** We use a sequence-level REINFORCE signal with group normalization. For each

prompt, we sample $G$ rollouts and compute

$$\tilde{A}_g := \mathrm{clip}\left(\frac{R_g - \mu}{\sigma}, -A_{\mathrm{clip}}, A_{\mathrm{clip}}\right),$$

$$\mu = \frac{1}{G}\sum_{g=1}^{G} R_g, \quad \sigma = \sqrt{\frac{1}{G}\sum_{g=1}^{G}(R_g - \mu)^2 + \epsilon_{\mathrm{norm}}},$$

$$(9)$$

where $R_g$ is the discounted episode return. The scalar $\tilde{A}_g$ is applied to all message log-probabilities within the trajectory. Bias from normalization and clipping is absorbed into $\zeta_i$;.

**Per-agent update with trust-region control.** At step $i$ update agent $j = \sigma(i)$ using rollouts from $\hat{\pi}^{i-1}$, likelihood ratio for message $m$ passed by agent $j$ at state $s$ is:

$$w(m; s) := \frac{\pi_{\mathrm{new}}^{(j)}(m \mid s)}{\pi_{\mathrm{cur}}^{(j)}(m \mid s)} = \prod_{u=1}^{|m|} \frac{\pi_{\mathrm{new}}^{(j)}(x_u \mid s, x_{<u})}{\pi_{\mathrm{cur}}^{(j)}(x_u \mid s, x_{<u})}.$$

We optimize the objective with an adaptive KL penalty, where $\mathrm{clip}_\epsilon(w) \triangleq \mathrm{clip}(w, 1-\epsilon, 1+\epsilon)$:

$$\mathcal{L}^{(j)} = \mathbb{E}_{m \sim \hat{\pi}^{i-1}}\left[\min\left(w\tilde{A}, \ \mathrm{clip}_\epsilon(w)\tilde{A}\right)\right] - \beta\widehat{D_{\mathrm{KLtok}}}.$$

$$(10)$$

the expectation is taken over messages passed by agent $j$ in the rollout batch under $\hat{\pi}^{i-1}$, $\epsilon$ is the ratio clip threshold, and $\beta$ is adjusted to satisfy the trust-region constraint.

**Token-decomposed KL monitoring.** Let $\mathcal{M}_j$ denote the messages passed by agent $j$ in the rollout batch. We monitor the sampled forward KL, and tune $\beta$ adaptively or early-stop when $\widehat{D_{\mathrm{KLtok}}}$ approaches $\delta_i$:

$$\widehat{D_{\mathrm{KLtok}}} := \frac{1}{|\mathcal{M}_j|} \sum_{m \in \mathcal{M}_j} \sum_{u=1}^{|m|} \log \frac{\pi_{\mathrm{cur}}^{(j)}(x_u \mid s, x_{<u})}{\pi_{\mathrm{new}}^{(j)}(x_u \mid s, x_{<u})} \le \delta_i.$$

$$(11)$$

**Agent plug and play replacement.** For agent replacement (Sec. 3.5), we first align the new agent to satisfy the trust-region constraint on a probe set sampled from the current team's occupancy, then resume standard TeamTR updates. Details for this are provided in Appendix D.

# 5. Experiments

We evaluate TeamTR to validate the theory in Sec. 3. We first report end-task performance under matched budgets, then diagnose the within-stage staleness predicted by Proposition 3.3, analyze training stability and certificate tracking under the monitored token-KL trust region, and finally test plug-and-play component replacement. Additional experiments and analyses are deferred to Appendix G, including

scaling with team size, compute, and wall-clock time, comparisons with parameter-matched single models, inference-time sampling, and selected ablation studies.

## 5.1. Setup

### 5.1.1. MODELS AND DATASETS.

We fine-tune LLMs with 1.7B-8B parameters, including Qwen2.5 (1.5B/3B/7B-Instruct) (Qwen et al., 2025), Qwen3 (1.7B/4B/8B-Instruct) (Yang et al., 2025), and LLaMA-3.3 (3B-Instruct) (Grattafiori et al., 2024). For reference, we also compare against larger models in the 30B-72B range, including Qwen2.5 (32B/72B-Instruct), Qwen3 (30B-A3B, 32B), LLaMA-3.3 (70B-Instruct), and QwQ (Team, 2025), DeepSeek-R1 (Guo et al., 2025), GPT-o4-mini. We evaluate on six benchmarks spanning mathematical reasoning (AIME 2024, AIME 2025, MATH-500 (Hendrycks et al., 2021)), logical reasoning (ZebraLogic (Lin et al., 2025), AutoLogi (Zhu et al., 2025)), and active reasoning (AR-Bench (Zhou et al., 2025) and PlanBench (Valmeekam et al., 2023)). Details of benchmarks are provided in Appendix H.

### 5.1.2. TRAINING AND EVALUATION PROTOCOL.

We implement training with VeRL (Sheng et al., 2025), using a temperature of 0.8, a top-p of 1.0, and a maximum output length of 32,768 tokens, unless otherwise specified. For math reasoning, we use training data from DeepScaleR and DAPO (Luo et al., 2025; Yu et al., 2025); for AR-Bench and PlanBench, we use their official training sets. For ZebraLogic, which is comparably challenging to MATH and lacks dedicated training or development sets, we train for each iteration using the same number of epochs that achieved the best performance on MATH (Prasad et al., 2024). Following standard practice for high-variance reasoning tasks, we sample multiple solutions per instance and report both pass@K (the proportion of instances with at least one correct solution among K samples) and avg@K (the average correctness across K samples). We set K=64 for AIME and ZebraLogic, K=4 for MATH-500, K=25 for AR-Bench, and K=8 for PlanBench. Mathematical verification uses the DeepScaleR verifier; external baseline numbers are reported as citations when verifiers may differ. Within each benchmark, we match the rollout and sampling budgets.

### 5.1.3. COMPARED METHODS.

We compare TeamTR against baselines spanning inference-time prompting, single-agent fine-tuning, multi-agent coordination, and strong reasoning systems. Inference-time baselines include Chain-of-Thought prompting (Wei et al., 2022), Self-Consistency with majority voting (Wang et al., 2022), and Tree-of-Thoughts (Yao et al., 2023). Single-agent fine-tuning baselines include PPO (Schulman et al., 2017), GRPO (Shao et al., 2024), and DAPO (Yu et al.,

2025), all trained under the same budget as the per-agent allocation in TeamTR. Multi-agent baselines include debate with and without a judge (Du et al., 2023), and naive sequential fine-tuning reuses stage-start rollouts for all within-stage updates, testing whether intermediate-occupancy evaluation mitigates the compounding degradation in Proposition 3.3.

### 5.1.4. TEAMTR SETUP.

We use a team of three agents with a fixed round-robin router. The trust-region radius $\delta_i$ controls the trade-off between update magnitude and certificate tightness (Theorem 3.4): smaller values yield tighter bounds but slower learning. We set $\delta_i = 0.01$ based on preliminary experiments; sensitivity analysis in Sec. 5.6 shows that performance is stable across $\delta_i \in [0.005, 0.02]$. The KL penalty coefficient $\beta$ in Eq. (10) is tuned adaptively to satisfy the constraint. All agents share the same base architecture but have separate parameter sets. For plug-and-play experiments, Stage-0 alignment uses 500 probe contexts from the current team's rollout distribution.

### 5.2. Main Results

Table 1 reports performance across general reasoning, active reasoning, and planning benchmarks with single-agent, multi-agent fine-tuning and strong thinking LLM baselines. Across benchmarks, TeamTR outperforms sequential fine-tuning baselines instantiated with PPO/GRPO/DAPO and multi-agent fine-tuning baselines (debate and role-play) under the same team budget. The heterogeneous configuration (1.7B+8B+14B) excels on challenging benchmarks (AIME, ZebraLogic), and the homogeneous configuration (3×8B) performs better on simpler tasks (MATH-500, AutoLogi), where consistent capacity across agents is beneficial.

### 5.3. Verifying Compounding Occupancy Shift

Section 3 predicts that, under within-stage sequential updates, evaluating later updates on stage-start rollouts can introduce a growing mismatch between the stage-start occupancy and the current intermediate occupancy. We quantify this effect by measuring the surrogate disagreement between rollouts collected at stage start and rollouts collected under the current intermediate team within the same training stage.

For a training stage $k$ and within-stage update index $i$, let $\widehat{L}_i^{\text{stale}}$ denote the surrogate evaluated using rollouts from the stage-start team ($d^{\hat{\pi}^0}$), and let $\widehat{L}_i^{\text{inter}}$ denote the surrogate evaluated using rollouts from the intermediate team before the $i$-th update ($d^{\hat{\pi}^{i-1}}$). We define the stale-occupancy gap as $\text{Gap}_i := \left| \widehat{L}_i^{\text{stale}} - \widehat{L}_i^{\text{inter}} \right|$. Figure 2 reports $\text{Gap}_i$ across training stages for $i = 2$ and $i = 3$. For baselines that reuse stage-start rollouts, the gap is systematically larger for the later within-stage update ($i = 3$) than for $i = 2$, indicating that stage-start rollouts become increasingly misaligned

after earlier updates are applied. This trend is consistent with the compounding mismatch implied by Remark 3.3. TeamTR avoids optimizing on stale rollouts by resampling under the intermediate team before each update, and empirically maintains a much smaller surrogate disagreement.

### 5.4. Scaling, Ablations, and Compute Trade-offs

Proposition 3.3 predicts a gap between stale and intermediate-occupancy evaluation that grows with team size. We examine this prediction beyond the default three-agent setting and summarize the results in Table 2. On MATH-500 with homogeneous Qwen3-1.7B teams, stale sequential updates exhibit near-quadratic growth in both the stale surrogate gap and empirical occupancy drift, with fitted exponents 1.94 and 1.91, respectively. Under TeamTR, the corresponding exponents are 1.07 and 1.05, matching the expected near-linear behavior. The scaling difference is also reflected in end-task performance: at $n = 8$, TeamTR reaches $87.9\%$ accuracy, while naive sequential training drops to $58.7\%$.

The same occupancy-shift pattern appears at larger model scales. For 3×Qwen3-8B teams, TeamTR improves AIME24 accuracy from $71.1\%$ to $88.1\%$ and reduces $\Delta_{\text{stale}}$ from 0.31 to 0.08. For heterogeneous 8B+14B+32B teams, TeamTR improves AIME24 accuracy from $77.8\%$ to $92.5\%$ and reduces $\Delta_{\text{stale}}$ from 0.29 to 0.06. Thus, stronger individual models do not remove the stale-occupancy mismatch; controlling intermediate occupancy remains beneficial for larger and heterogeneous teams.

Table 2(b) further separates the two components of TeamTR. Removing intermediate-occupancy resampling recovers the naive sequential baseline and leads to a large stale gap. Resampling every two updates reduces this mismatch, but remains 8.8 points below full TeamTR on AIME24, indicating that stale drift still accumulates within a stage. Trust-region control is also necessary. An adaptive KL-penalty-only variant improves over naive sequential training, but it trails full TeamTR by 3.2 points and exhibits larger stale gaps and higher instability. Without any trust-region constraint, the update becomes the least stable and produces the weakest coordination score. These ablations suggest that TeamTR's gains come from the combination of fresh intermediate rollouts and explicit token-level trust-region enforcement.

The additional sampling cost is modest relative to the stability gain. On AIME24 with 3×Qwen3-8B over 40 stages, TeamTR uses 158.3M tokens and 49.2K rollouts, compared with 142.5M tokens and 44.3K rollouts for naive sequential training. This corresponds to about $+11.1\%$ overhead in both token and rollout accounting. The measured per-update wall-clock cost is 192.8s for TeamTR and 172.0s for naive sequential training, giving a $1.12\times$ overhead. KL monitoring contributes 6.2s per update, so the dominant

*Table 1.* Performance comparison across reasoning and planning benchmarks. Green/red indicate improvement/decline relative to TeamTR (3×8B). Best results are **bold**; second-best are underlined.

| Method | Size | General Reasoning (%) | | | | | Active Reasoning (%) | | | Planning (%) |
|---|---|---|---|---|---|---|---|---|---|---|
| | | AIME24 | AIME25 | MATH-500 | ZebraLogic | AutoLogi | DC | SP | GN | PlanBench |
| *Proprietary Models* | | | | | | | | | | |
| GPT-4o-mini | – | $8.1_{-80.0}$ | $8.8_{-69.5}$ | $78.2_{-21.1}$ | $20.1_{-74.6}$ | $62.5_{-28.0}$ | $44.0_{-2.7}$ | $40.8_{-2.3}$ | $43.6_{-1.7}$ | $34.7_{-9.4}$ |
| o4-mini | – | $79.6_{-8.5}$ | $74.8_{-3.5}$ | $98.0_{-1.3}$ | $88.9_{-5.8}$ | $86.3_{-4.2}$ | $46.7$ | $43.1$ | $45.1_{-0.2}$ | $41.2_{-2.9}$ |
| *Open-Source Thinking Models* | | | | | | | | | | |
| Qwen3-A3B | 30B | $80.4_{-7.7}$ | $70.9_{-7.4}$ | $98.0_{-1.3}$ | $89.5_{-5.2}$ | $88.1_{-2.4}$ | $43.1_{-3.6}$ | $38.4_{-4.7}$ | $42.3_{-3.0}$ | $39.1_{-5.0}$ |
| QwQ | 32B | $79.5_{-8.6}$ | $69.5_{-8.8}$ | $98.0_{-1.3}$ | $76.8_{-17.9}$ | $86.3_{-4.2}$ | $41.9_{-4.8}$ | $36.1_{-7.0}$ | $39.5_{-5.8}$ | $37.9_{-6.2}$ |
| Qwen3 (thinking) | 32B | $81.4_{-6.7}$ | $72.9_{-5.4}$ | $97.2_{-2.1}$ | $88.8_{-5.9}$ | $87.3_{-3.2}$ | $42.7_{-4.0}$ | $38.9_{-4.2}$ | $41.1_{-4.2}$ | $40.3_{-3.8}$ |
| DeepSeek-R1 Distill | 70B | $70.0_{-18.1}$ | $56.3_{-22.0}$ | $94.5_{-4.8}$ | $71.3_{-23.4}$ | $83.5_{-7.0}$ | $42.1_{-4.6}$ | $35.4_{-7.7}$ | $40.7_{-4.6}$ | $39.1_{-5.0}$ |
| *Single-Agent Fine-tuning (Qwen3-8B)* | | | | | | | | | | |
| GRPO | 8B | $39.1_{-49.0}$ | $27.9_{-50.4}$ | $90.7_{-8.6}$ | $35.4_{-59.3}$ | $80.3_{-10.2}$ | $40.1_{-6.6}$ | $35.2_{-7.9}$ | $38.9_{-6.4}$ | $33.1_{-11.0}$ |
| DAPO | 8B | $41.3_{-46.8}$ | $30.1_{-48.2}$ | $91.5_{-7.8}$ | $36.7_{-58.0}$ | $80.9_{-9.6}$ | $40.7_{-6.0}$ | $35.9_{-7.2}$ | $39.3_{-6.0}$ | $33.5_{-10.6}$ |
| *Multi-Agent Fine-tuning (3×Qwen3-8B)* | | | | | | | | | | |
| Debate | 3×8B | $68.7_{-19.4}$ | $59.1_{-19.2}$ | $95.2_{-4.1}$ | $84.1_{-10.6}$ | $85.1_{-5.4}$ | $43.1_{-3.6}$ | $38.0_{-5.1}$ | $42.3_{-3.0}$ | $38.3_{-5.8}$ |
| Debate + Judge | 3×8B | $71.1_{-17.0}$ | $61.7_{-16.6}$ | $95.9_{-3.4}$ | $86.3_{-8.4}$ | $85.9_{-4.6}$ | $44.3_{-2.4}$ | $39.4_{-3.7}$ | $43.5_{-1.8}$ | $39.1_{-5.0}$ |
| Role-play | 3×8B | $69.9_{-18.2}$ | $60.5_{-17.8}$ | $95.7_{-3.6}$ | $85.1_{-9.6}$ | $85.5_{-5.0}$ | $43.7_{-3.0}$ | $38.7_{-4.4}$ | $43.1_{-2.2}$ | $38.7_{-5.4}$ |
| *TeamTR (Ours)* | | | | | | | | | | |
| Homogeneous | 3×8B | $88.1$ | $78.3$ | **$99.3$** | $94.7$ | **$90.5$** | **$46.7$** | **$43.1$** | **$45.3$** | $44.1$ |
| Heterogeneous | 1.7B+8B+14B | **$89.7_{+1.6}$** | **$80.1_{+1.8}$** | $98.1_{-1.2}$ | **$96.3_{+1.6}$** | $88.7_{-1.8}$ | $45.5_{-1.2}$ | $42.3_{-0.8}$ | $44.5_{-0.8}$ | **$44.5_{+0.4}$** |

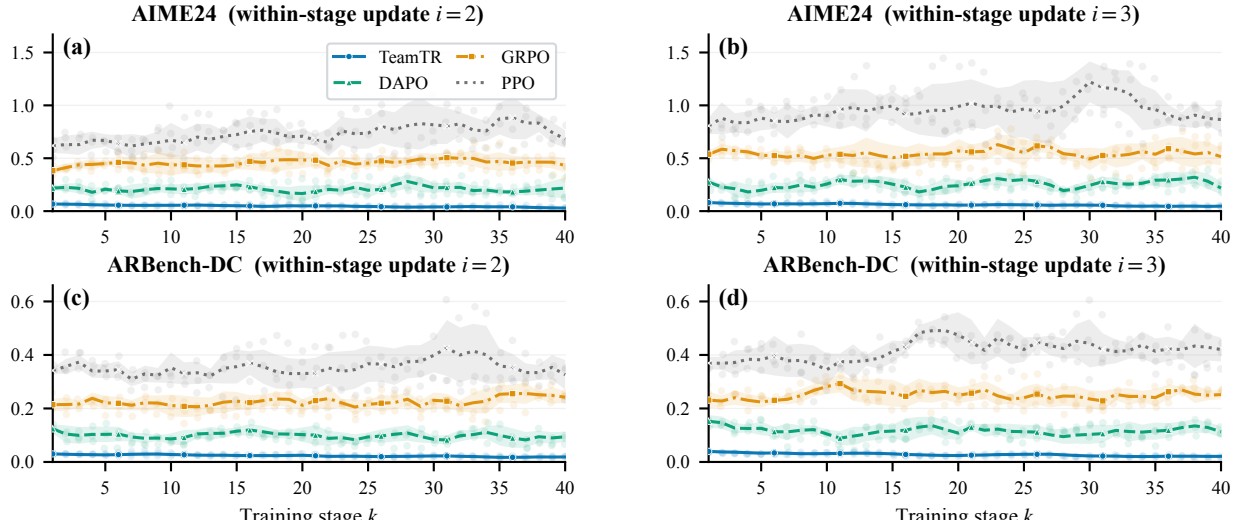

*Figure 2.* **Stale-occupancy gap within a training stage.** We plot the occupancy gap across training stages $k$ for within-stage update indices $i = 2$ and $i = 3$. For the baselines-reuse stage, the gap is larger for the later update ($i = 3$), consistent with Remark 3.3.

cost remains rollout generation rather than trust-region monitoring.

The turn-taking assumption remains the formal scope of the guarantee. Lemma 3.1 relies on a single active agent at each decision step, and does not directly cover concurrent generation, overlapping tool calls, or branching blackboard-style interaction. The stale-rollout diagnosis, however, is not restricted to text-only reasoning. In a sequential tool-mediated setting on $\tau$-bench, TeamTR improves average pass@1 / pass@4 from $45.6/13.8$ under naive sequential training to $52.9/19.7$ with the same 3×8B team. This suggests that intermediate-occupancy control remains useful when the shared context includes tool-mediated state changes, even

though the formal theorem is stated for turn-taking interaction. The empirical certificate in Sec. 5.6 should be interpreted as a training diagnostic rather than a literal per-stage guarantee. It combines logged surrogate gains, monitored KL terms, and a proxy for the estimation-error penalty $\zeta_i$. In our logs, the $\widehat{\zeta}$ proxy contributes $18\%$ of the total penalty on average, and the near-threshold flip rate of the sampled token-KL monitor is $2.4\%$ under $50\%$ token-position subsampling. The certificate therefore provides a useful conservative signal during training, while its tightness depends on the proxy quality and on the reliability of sampled token-KL monitoring.

*Table 2.* **Scaling behavior, ablations, and overhead.** Panel (a) reports team-size scaling and larger-model results. For model-scale rows, $\Delta_{\text{stale}}$ is reported as AIME24 / MATH-500. Panel (b) isolates the effects of intermediate-occupancy resampling and trust-region control.

**(a) Team-size and model-scale behavior**

| Setting | Method | AIME24 | MATH-500 | Drift summary |
|---|---|---|---|---|
| $n$-scaling | Naive Seq. | – | 58.7 ($n=8$) | $\alpha_\Delta$=1.94, $\alpha_D$=1.91 |
| $n$-scaling | TeamTR | – | 87.9 ($n=8$) | $\alpha_\Delta$=1.07, $\alpha_D$=1.05 |
| 3×8B | Naive Seq. | 71.1 | 95.1 | 0.31/0.18 |
| 3×8B | TeamTR | 88.1 | 99.3 | 0.08/0.03 |
| 1.7B+8B+14B | TeamTR | 89.7 | 98.1 | 0.07/0.04 |
| 8B+14B+32B | Naive Seq. | 77.8 | 97.1 | 0.29/0.16 |
| 8B+14B+32B | TeamTR | 92.5 | 99.4 | 0.06/0.02 |

**(b) Ablations on AIME24**

| Variant | Acc. (%) | $\Delta_{\text{stale}} \downarrow$ | Stab.$\downarrow$ | Coord. (%) |
|---|---|---|---|---|
| TeamTR (full) | **88.1**±1.2 | **0.08** | **1.9** | **89.1** |
| KL-penalty only, adaptive $\beta$ | 84.9±1.6 | 0.14 | 2.7 | 83.1 |
| KL-penalty only, fixed $\beta$ | 80.7±2.0 | 0.20 | 3.5 | 77.2 |
| Resample every 2 updates | 79.3±1.8 | 0.18 | 2.8 | 79.2 |
| No resampling (= Naive Seq.) | 71.1±2.8 | 0.31 | 4.2 | 71.5 |
| No trust region ($\delta \to \infty$) | 68.3±3.5 | 0.42 | 6.1 | 62.3 |

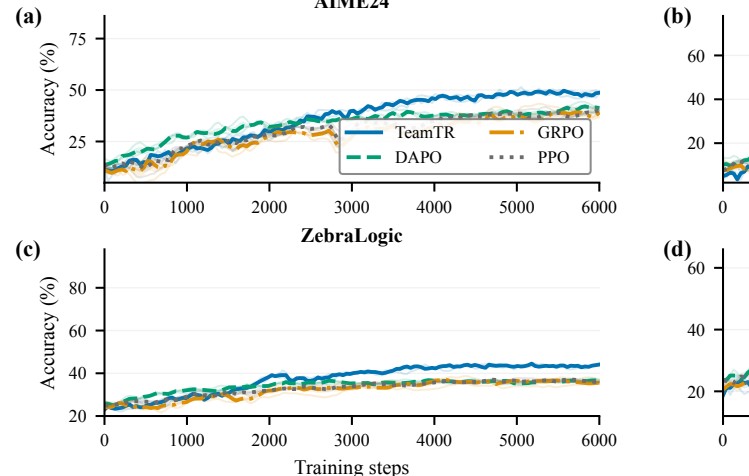

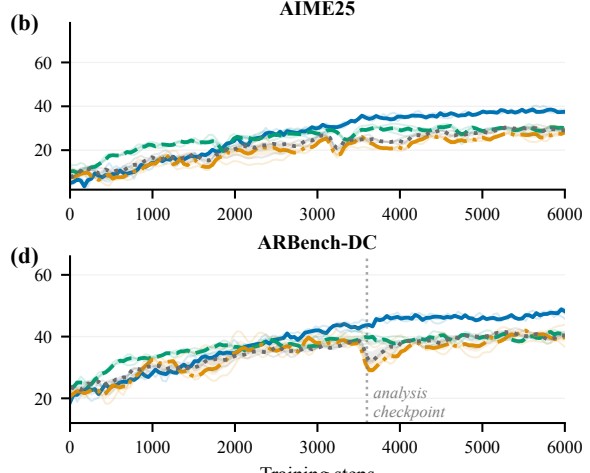

*Figure 3.* **Training dynamics under matched rollout budgets.** We compare TeamTR against sequential baselines instantiated with PPO/GRPO/DAPO. Baselines exhibit occasional regressions during training, whereas TeamTR maintains more stable improvement by evaluating each update on intermediate rollouts and enforcing per-update trust regions. Shaded regions indicate variation across seeds.

## 5.5. Training Dynamics

Figure 3 shows held-out accuracy as a function of training steps under matched rollout budgets. Across benchmarks, TeamTR improves steadily and achieves higher final performance than sequential baselines trained with PPO/GRPO/DAPO. The baselines exhibit non-monotonic trajectories with occasional regressions, which are accompanied by large stale-occupancy gaps measured in Sec. 5.3. These trends align with Remark 3.3: reusing stage-start rollouts can lead to compounding mismatch, while intermediate-occupancy evaluation localizes the occupancy-shift penalty.

## 5.6. Trust-Region Enforcement and Certificate Tracking

Theorems 3.4–3.6 lower bound improvement under the per-update trust-region condition $\widehat{D_{\text{KLtok}}} \leq \delta_i$. We examine whether updates stay within the prescribed KL region and whether the resulting empirical certificate is informative.

Figure 4(a) shows the distribution of per-update token-level KL divergences on AIME25, with the trust-region threshold $\delta$ marked by the red dashed line. The percentages above each method report the out-of-region rate, i.e., the fraction of updates with $\widehat{D_{\text{KLtok}}} > \delta$. TeamTR keeps most

updates in-region, whereas PPO/GRPO/DAPO variants exhibit substantially higher out-of-region rates. Figure 4(b) compares the cumulative measured improvement across training stages with the cumulative certificate lower bound obtained by plugging logged surrogates and KL terms into Theorem 3.6. Figure 4(c) further plots the per-stage certificate value against the corresponding empirical improvement, reporting rank correlation and the violation rate (the fraction of points where the certificate exceeds the measured improvement). On AIME25, the certificate remains conservative while tracking progress with a consistent gap.

## 5.7. Token-Level Analysis of Shift

Our per-update trust region constrains behavior-to-updated divergence through the token-level decomposition in Eq. (1). To localize where probability mass moves, Fig. 5 compares token logits before and after an update at a representative ARBench checkpoint, with tokens ordered by their pre-update probabilities (remaining vocabulary aggregated as `other`). In-region updates (monitored by $\widehat{D_{\text{KLtok}}} \leq \delta$) primarily modify a small set of high-probability tokens, resulting in controlled, localized changes. Out-of-region updates ($\widehat{D_{\text{KLtok}}} > \delta$) induce larger reshuffling among

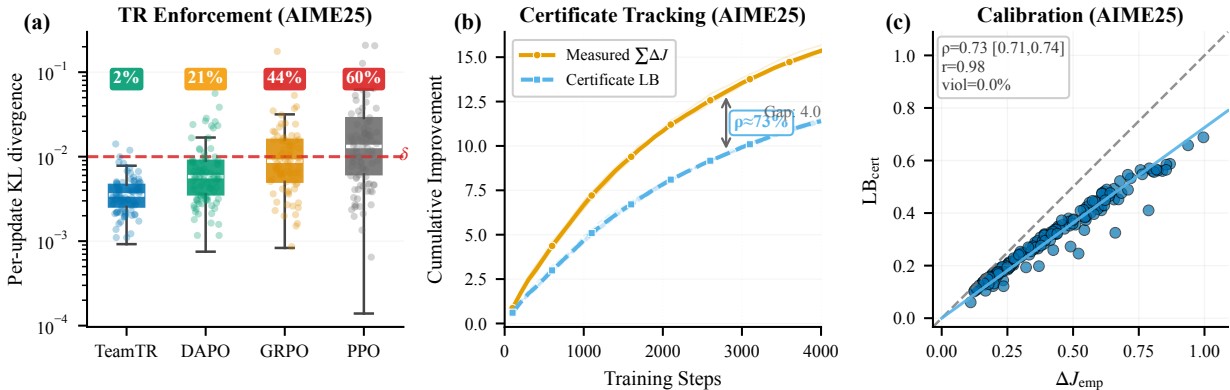

*Figure 4.* **Trust-region enforcement and certificate tracking (AIME25).** (a) Distribution of per-update token-level KL divergence; the red dashed line marks the threshold $\delta$, and percentages indicate out-of-region rates $(\widehat{D_{\mathrm{KLtok}}} > \delta)$. (b) Cumulative measured improvement versus certificate lower bound (Theorem 3.6); $\rho$ denotes rank correlation. (c) Per-stage calibration of certificate values against empirical improvements; "viol" indicates the fraction of stages where the bound is violated.

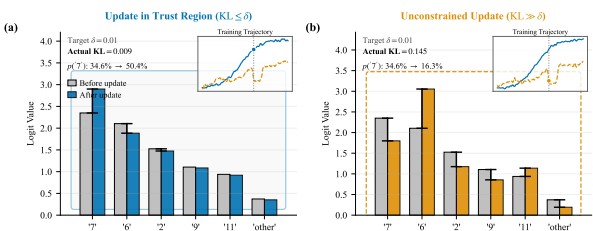

*Figure 5.* **Token-level logit shifts by pre-update probability rank.** Left: an in-region update $(\widehat{D_{\mathrm{KLtok}}} \leq \delta)$ produces localized changes on top tokens. Right: an out-of-region update $(\widehat{D_{\mathrm{KLtok}}} > \delta)$ reshuffles probability mass among alternatives.

close alternatives and can flip the dominant token, consistent with the instabilities observed in Sec. 5.5.

### 5.8. Plug-and-Play Component Replacement

Proposition 3.8 implies that after replacing an agent, once the new component is aligned to satisfy the per-agent trust-region interface, subsequent TeamTR updates continue to admit the same improvement-certificate form (the replacement step itself is not certified). We evaluate this plug-and-play by initializing a team with Qwen2.5-Instruct (1.5B/3B/7B) and, at stage $k_{\mathrm{swap}} = 20$, replacing the 1.5B agent with Qwen3-8B, which requires protocol alignment (see Appendix G.7 for details). We compare three strategies: *Direct swap*, which replaces the agent and immediately resumes training; *Stage-0 aligned swap*, which first aligns the new agent on 500 probe contexts so that the monitored token-KL satisfies $\widehat{D_{\mathrm{KLtok}}} \leq \delta$ on the probe distribution before resuming training; and *Retrain-from-scratch*, which resets the team after the swap and retrains for the remaining budget. Figure 6 reports scores on AIME24 and ARBench-DC. A direct swap causes an immediate performance drop at $k_{\mathrm{swap}}$ ($-18\%$ on AIME24) and a gradual recovery thereafter. In contrast, Stage-0 alignment largely eliminates the swap shock and enables post-swap improvement, achieving substantially

higher final performance under the same post-swap budget (+27% and +24% on AIME24 and ARBench-DC, respectively). Retraining from scratch avoids reusing pre-swap components but is less efficient: within the constrained budget, it remains below the aligned-swap trajectory.

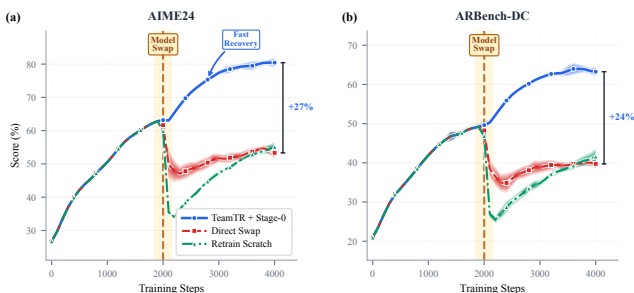

*Figure 6.* **Plug-and-play component replacement.** A Qwen2.5-Instruct (1.5B/3B/7B) team is trained over 20 stages; and the 1.5B agent is replaced with Qwen3-8B. Stage-0 alignment mitigates the swap shock and achieves the best performance.

## 6. Conclusion and Limitations

We analyzed sequential fine-tuning of shared-context LLM teams and identified *compounding occupancy shift*: later component updates can be optimized or evaluated under stale rollouts, yielding a certificate penalty that scales quadratically with team size. TeamTR addresses this by evaluating each update under the partially updated team's occupancy and enforcing per-agent trust regions via a token-decomposable behavior-to-updated divergence monitored by a sampled estimator. This yields lower bounds on per-update and per-stage improvement and improves coordination stability and accuracy in practice, including a plug-and-play component with a brief alignment step. **Limitations:** Our current analysis targets a single-active-agent protocol; We rely on per-update resampling to avoid long-horizon importance weighting.

## Impact Statement

This paper presents work aimed at advancing the field of machine learning. There are many potential societal consequences of our work, none of which we consider necessary to highlight here.

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

# Appendix contents

# A. Extended Related Work

**Inference-time multi-agent LLM systems.** Recent advancements have shifted the deployment of LLMs from monolithic predictors to collaborative multi-agent systems (MAS) that operate entirely at inference time. In this paradigm, frozen LLMs serve as cognitive controllers, interacting via natural language without parameter updates. Unlike traditional MARL, which relies on training task-specific policies from scratch (Rashid et al., 2020; Yu et al., 2022), these systems orchestrate collaboration solely through prompt engineering and context management. Pioneering works such as CAMEL (Li et al., 2023) and Generative Agents (Park et al., 2023) have demonstrated that, by using role-playing prompts and memory streams, agents can autonomously decompose tasks and simulate complex social behaviors using only their pre-trained knowledge. Beyond simulation, training-free collaborative strategies have been deployed to enhance reasoning. (Du et al., 2023) and (Liang et al., 2024) propose test-time debate and consensus-building frameworks, which reduce hallucinations by aggregating diverse perspectives from fixed models. To address complex tasks such as software engineering, systems like MetaGPT (Hong et al., 2023) and ChatDev (Qian et al., 2024) incorporate hard-coded Standard Operating Procedures (SOPs) into the context. By enforcing structured sequential handoffs via system prompts, these frameworks stabilize the output of heterogeneous teams without requiring gradient-based optimization. General-purpose libraries like AutoGen (Wu et al., 2024) further formalize these inference-only interaction topologies. Despite these empirical successes, these deployments remain constrained by the capabilities of the base models and fixed interaction rules, thereby forfeiting opportunities for training-time optimization that could yield more intrinsic and stable collaboration.

**Training multi-agent LLM systems and learning to coordinate.** Beyond inference-time orchestration, recent research has pivoted towards updating model parameters to internalize coordination capabilities. Early efforts largely used Supervised Fine-Tuning (SFT) to specialize agents for distinct roles or to enhance general agentic capabilities. For instance, large-scale interaction trajectories are constructed to fine-tune Llama-based models, which significantly outperform prompting baselines on complex planning and tool-use tasks (Chen et al., 2023; Zeng et al., 2024; Yi et al., 2025). Other works focus on distilling specific functional roles; for example, Shepherd (Wang et al., 2023) and Gorilla (Patil et al., 2024) demonstrate that fine-tuning dedicated modules can establish effective heterogeneous teams via static supervision. Moving toward reinforcement learning and preference optimization, researchers aim to teach agents to coordinate dynamically. Frameworks such as LLM-Blender (Jiang et al., 2023) and RLAIF (Lee et al., 2023) employ ranking mechanisms and AI-generated feedback to fuse outputs or refine behaviors, thereby aligning agents with collective goals. Furthermore, (Liu et al., 2023a) explores training socially aligned agents within simulated societies using reinforcement signals derived from social interactions. However, these training paradigms pose significant challenges due to non-stationarity. Sequential updates in a shared context often lead to pathological behaviors such as sycophancy—where agents agree with users or teammates regardless of correctness to maximize rewards (Wei et al., 2023; Perez et al., 2023; Sharma et al., 2023). Existing methods typically rely on naive joint training or independent fine-tuning, lacking a unified framework to explicitly control these distribution shifts and guarantee monotonic improvement in collaboration.

**Trust regions and block-coordinate optimization in multi-agent RL.** Trust-region methods establish the theoretical foundation for stable reinforcement learning by guaranteeing monotonic policy improvement. Seminal works like TRPO (Schulman et al., 2015) and PPO (Schulman et al., 2017) rely on the monotonic improvement bounds originally derived by Kakade and Langford (Kakade & Langford, 2002), utilizing KL-divergence constraints to prevent destructive parameter updates. In the multi-agent domain, extending these guarantees is non-trivial due to the non-stationarity introduced by simultaneous updates. While methods like MAPPO (Yu et al., 2022), ACORN (Xie et al., 2025a) empirically demonstrate the effectiveness of PPO in cooperative games, they often lack strict convergence guarantees in heterogeneous settings. To address this, recent theoretical advancements have formulated multi-agent learning as a block-coordinate ascent problem. Most notably, HATRPO and HAPPO (Kuba et al., 2021) derive a multi-agent advantage decomposition lemma, proving that sequential updates of agents—rather than simultaneous joint optimization—are necessary to preserve the monotonic improvement property in heterogeneous teams. This sequential paradigm has been successfully applied to continuous control benchmarks like StarCraft II (Rashid et al., 2020; Kuba et al., 2021). However, these algorithms operate on low-dimensional state spaces and have not yet been adapted to the high-dimensional, discrete, and autoregressive nature of Large Language Models, where "trust regions" must be defined over token-level probability distributions within an evolving textual context.

**Distribution shift under sequential updates and modular system evolution.** Training collaborative systems introduces non-stationarity, where sequential updates create a moving target for optimization. While classical MARL addressed this with importance sampling (Foerster et al., 2017; Espeholt et al., 2018; Xie et al., 2025b; Liu et al., 2025b; 2023b), recent

analyses in RLHF (Casper et al., 2023; Singhal et al., 2023) reveal that such distribution shifts in LLMs often lead to mode collapse and reward hacking. Controlling this shift is crucial not only for stability but also for enabling modular system evolution—combining or upgrading capabilities without retraining. In the era of LLMs, modularity has evolved from simple adapters to model merging techniques. Seminal works like Task Arithmetic (Ilharco et al., 2022) and Model Soups (Wortsman et al., 2022) demonstrate that agent capabilities can be composed by linearly manipulating weight vectors. Furthermore, LoRAHub (Huang et al., 2023) extends this to the dynamic composition of parameter-efficient modules. However, these static merging techniques assume independent training distributions and often require heuristic tuning of merging coefficients. TeamTR provides a complementary dynamic framework: it leverages trust regions to certify that the distribution shift induced by updating or swapping one module (agent) remains within a safe bound, effectively enabling "certified model arithmetic" during active collaboration.

# Correspondence Between Main Text and Appendix

*Table 3.* Main theoretical results and their detailed proofs.

| Main Text | Appendix | Description |
|---|---|---|
| Lemma 3.2 | B.3 | Occupancy shift bounds (expected divergence form; reverse-KL compatible) |
| Proposition 3.3 | B.4 | Stale-occupancy surrogate mismatch and quadratic-in-$n$ compounding term |
| Theorem 3.4 | C.1 | Single-step improvement lower bound under sequential trust-region update |
| Lemma 3.5 | C.2 | KL identity for single-factor update; token-KL reduction (reverse KL) |
| Theorem 3.6 | C.2 | Joint-stage improvement lower bound (order-robust form) |
| Theorem 3.7 | C.3 | Stage-wise improvement lower bound under stale-occupancy surrogates (explicit double-sum penalty) |
| Proposition 3.8 | D.2 | Certified resumability after replacement (post-alignment) |
| Proposition 3.9 | D.2 | Certificate tightening |
| **Supporting / Implementation Material** | | |
| Token-level KL trust region | B.2 | Reverse token-level KL functional and exact autoregressive decomposition (rollout-estimable) |
| Stage-0 alignment | D.1 | Reverse-KL projection / distillation objective for heterogeneous upgrades |
| Group-based advantages | E.1 | Group-standardized, hard-clipped sequence-level advantages and bias proxy |
| Concentration bounds | E.3 | Finite-sample concentration for group-based surrogates, and ratio-clipping bias control |
| Implementation notes | F.6 | Token-KL token-sum, std normalization, hard clip, ratio clip |
| Diagnostics | F.7 | Certificate tightness plots and training monitors |
| Information-theoretic bounds | F.1, F.2 | Oracle upper envelope and budget-aware lower envelope (auxiliary) |
| Smoothness & optimization | F.4 | Smoothness and projected-gradient ascent convergence (auxiliary) |

## Notation

We work with a discounted MDP $\mathcal{M} = (\mathcal{S}, \mathcal{A}, P, r, \gamma)$ with $\gamma \in (0, 1)$ and bounded rewards $|r| \leq R_{\max}$. The analysis also applies to finite-horizon sequence generation by setting an effective horizon $H \approx (1 - \gamma)^{-1}$.

$\pi$          joint (factorized) team policy. At each state $s$, $\pi(\cdot \mid s) = \prod_{k=1}^{n} \pi^{(k)}(\cdot \mid s)$.

$\mu$          initial-state distribution at $t = 0$ (used in the discounted visitation fixed-point equation).

$d^{\pi}$          discounted visitation distribution: $d^{\pi}(s) = (1 - \gamma) \sum_{t \geq 0} \gamma^t \Pr_{\pi}(s_t = s)$.

$J(\pi)$          discounted return: $J(\pi) = \mathbb{E}_{\tau \sim \pi}\left[ \sum_{t \geq 0} \gamma^t r_t \right]$.

$Q^{\pi}, V^{\pi}, A^{\pi}$

         action-value, value, and advantage under $\pi$.

$\sigma$          update order (a permutation of $\{1, \ldots, n\}$) within a stage.

$\hat{\pi}^i$          intermediate joint policy after $i$ factor updates in a stage: $\hat{\pi}^0 = \pi_{\text{cur}}$ and $\hat{\pi}^i = \hat{\pi}^{i-1}[\sigma(i) \leftarrow \pi_{\text{tar}}^{\sigma(i)}]$; $\bar{\pi} = \hat{\pi}^n$.

$D_{\text{TV}}^{\max}(\pi' \| \pi)$

         $\sup_{s \in \mathcal{S}} D_{\text{TV}}(\pi'(\cdot \mid s), \pi(\cdot \mid s))$.

$D_{\text{KL}_{\text{tok}}}^{\rho}(\pi \| \pi')$

         token-level (expected statewise) *reverse* KL with reference policy $\rho$ (Definition B.1).

$\delta_i$          trust-region radius for step $i$: $D_{\text{KL}_{\text{tok}}}^{\hat{\pi}^{i-1}}(\pi_{\text{cur}}^{\sigma(i)} \| \pi_{\text{tar}}^{\sigma(i)}) \leq \delta_i$ (Eq. (2) in the main text).

$\widehat{A}_{\text{GRP}}^{i-1}$          group-standardized, hard-clipped sequence-level advantage (Appendix E.1).

$\zeta_i$          surrogate-estimation error term at step $i$: any quantity satisfying

$$\zeta_i \geq \left| \mathbb{E}_{s \sim d^{\hat{\pi}^{i-1}}, \mathbf{a} \sim \hat{\pi}^i(\cdot | s)}\left[ \mathbb{E}\left[\widehat{A}_{i-1}(s, \mathbf{a})\right] - A^{\hat{\pi}^{i-1}}(s, \mathbf{a}) \right] \right|.$$

(A conservative sufficient choice is $\zeta_i = \sup_{s, \mathbf{a}} |\mathbb{E}[\widehat{A}_{i-1}(s, \mathbf{a})] - A^{\hat{\pi}^{i-1}}(s, \mathbf{a})|$.)

$A_{\max}$      advantage bound: $|A^{\pi}(s, \mathbf{a})| \leq A_{\max}$ with $A_{\max} \leq 2R_{\max}/(1 - \gamma)$.

$A_{\mathrm{clip}}$      clipping threshold ensuring $|\widehat{A}_{\mathrm{GRP}}^{i-1}| \leq A_{\mathrm{clip}}$ almost surely.

$N_i$      number of independent prompt-groups used for step $i$ (each group contains $G$ rollouts).

$G$      number of rollouts sampled per prompt-group.

$w_i(s, \mathbf{a})$      step-$i$ per-decision probability ratio used for importance weighting: $w_i(s, \mathbf{a}) = \hat{\pi}^i(\mathbf{a} \mid s)/\hat{\pi}^{i-1}(\mathbf{a} \mid s)$ (under the turn-taking protocol, this depends only on the updated factor).

$w_i^{\mathrm{clip}}$      ratio-clipped weight $w_i^{\mathrm{clip}} = \mathrm{clip}(w_i, 1 - \epsilon_w, 1 + \epsilon_w)$ (Appendix E.3, F.6).

## B. Occupancy Shift and Token-Level Trust Regions

### B.1. Proof of Lemma 3.1

*Proof.* Fix a state $s$ and let $j = j(s)$ be the active agent. Under the turn-taking protocol, for all $k \neq j$ and for both team policies $\pi, \pi'$, the inactive factor is deterministic: $\pi^{(k)}(a_k \mid s) = \pi'^{(k)}(a_k \mid s) = \mathbb{1}\{a_k = \mathrm{noop}\}$.

Therefore, the joint action distributions at state $s$ factorize as

$$\pi(\mathbf{a} \mid s) = \pi^{(j)}(a_j \mid s) \prod_{k \neq j} \mathbb{1}\{a_k = \mathrm{noop}\}, \qquad \pi'(\mathbf{a} \mid s) = \pi'^{(j)}(a_j \mid s) \prod_{k \neq j} \mathbb{1}\{a_k = \mathrm{noop}\}.$$

The support of both distributions is restricted to $\{a_k = \mathrm{noop} \ \forall k \neq j\}$, and on this set the ratio simplifies:

$$\log \frac{\pi'(\mathbf{a} \mid s)}{\pi(\mathbf{a} \mid s)} = \log \frac{\pi'^{(j)}(a_j \mid s)}{\pi^{(j)}(a_j \mid s)}.$$

Thus,

$$D_{\mathrm{KL}}\big(\pi'(\cdot \mid s)\|\pi(\cdot \mid s)\big) = \sum_{\mathbf{a}} \pi'(\mathbf{a} \mid s) \log \frac{\pi'(\mathbf{a} \mid s)}{\pi(\mathbf{a} \mid s)} = \sum_{a_j} \pi'^{(j)}(a_j \mid s) \log \frac{\pi'^{(j)}(a_j \mid s)}{\pi^{(j)}(a_j \mid s)} = D_{\mathrm{KL}}\big(\pi'^{(j)}(\cdot \mid s)\|\pi^{(j)}(\cdot \mid s)\big).$$

The expectation reduction for functions that depend only on the active action follows from the same support argument. $\qquad\square$

### B.2. Token-Level KL Trust Regions and Autoregressive Chain Rule

This section formalizes the bridge between the implementation choice—*token-level reverse KL*—and the theoretical trust-region radius used in our certificates. The key point is that when each macro-action is an autoregressive message, the *reverse* message-level KL equals the expected *sum* of token-level reverse KLs (chain rule), with the expectation taken under the reference/current policy. This matches on-policy rollouts under the intermediate team and avoids a token-occupancy mismatch.

**Definition B.1** (Discounted token-level reverse-KL functional with a reference policy). For a reference policy $\rho$ and two policies $\pi, \pi'$, define

$$D_{\mathrm{KL}_{\mathrm{tok}}^{\rho}}(\pi\|\pi') := \mathbb{E}_{\tau \sim \rho} \left[ (1 - \gamma) \sum_{t \geq 0} \gamma^t D_{\mathrm{KL}}\big(\pi(\cdot \mid s_t)\|\pi'(\cdot \mid s_t)\big) \right].$$

**Lemma B.2** (Token-level KL equals expected statewise KL under $d^{\rho}$). *For any $\rho, \pi, \pi'$,*

$$D_{\mathrm{KL}_{\mathrm{tok}}^{\rho}}(\pi\|\pi') = \mathbb{E}_{s \sim d^{\rho}} D_{\mathrm{KL}}\big(\pi(\cdot \mid s)\|\pi'(\cdot \mid s)\big).$$

*Proof.* Recall that the discounted state-visitation distribution induced by $\rho$ is the (discounted) occupancy measure

$$d^{\rho}(\mathrm{d}s) := (1 - \gamma) \sum_{t=0}^{\infty} \gamma^t \Pr_{\tau \sim \rho} (s_t \in \mathrm{d}s),$$

i.e., for any measurable set $B \subseteq \mathcal{S}$,

$$d^\rho(B) = (1 - \gamma) \sum_{t=0}^{\infty} \gamma^t \Pr_{\tau \sim \rho} (s_t \in B).$$

Equivalently, for any measurable function $g : \mathcal{S} \to [0, \infty]$,

$$\begin{aligned}
\mathbb{E}_{s \sim d^\rho}[g(s)] &= \int_{\mathcal{S}} g(s) \, d^\rho(\mathrm{d}s) \\
&= (1 - \gamma) \sum_{t=0}^{\infty} \gamma^t \int_{\mathcal{S}} g(s) \Pr_{\tau \sim \rho} (s_t \in \mathrm{d}s) \\
&= (1 - \gamma) \sum_{t=0}^{\infty} \gamma^t \mathbb{E}_{\tau \sim \rho}\big[g(s_t)\big].
\end{aligned} \tag{12}$$

(The interchange of the sum and the integral/expectation is justified by Tonelli's theorem since $g \geq 0$; if one works in finite spaces, this follows from elementary manipulations.)

Now apply (12) to

$$g(s) \;=\; D_{\mathrm{KL}}\big(\pi(\cdot \mid s) \| \pi'(\cdot \mid s)\big) \in [0, \infty],$$

(where the value may be $+\infty$ if $\pi(\cdot \mid s)$ is not absolutely continuous w.r.t. $\pi'(\cdot \mid s)$; the identity still holds in the extended reals). Then

$$\begin{aligned}
\mathbb{E}_{s \sim d^\rho} D_{\mathrm{KL}}\big(\pi(\cdot \mid s) \| \pi'(\cdot \mid s)\big) &= (1 - \gamma) \sum_{t=0}^{\infty} \gamma^t \mathbb{E}_{\tau \sim \rho}\big[D_{\mathrm{KL}}\big(\pi(\cdot \mid s_t) \| \pi'(\cdot \mid s_t)\big)\big] \\
&= \mathbb{E}_{\tau \sim \rho}\left[(1 - \gamma) \sum_{t=0}^{\infty} \gamma^t D_{\mathrm{KL}}\big(\pi(\cdot \mid s_t) \| \pi'(\cdot \mid s_t)\big)\right] \\
&= D_{\mathrm{KL}_{\mathrm{tok}}}^\rho(\pi \| \pi'),
\end{aligned}$$

where the second equality again uses Tonelli/linearity (the summand is nonnegative). This proves the claim. $\qquad\square$

**Autoregressive messages.** Fix an agent $j$ and a state $s$. Let a macro-action (message) be a variable-length token sequence $m = (x_1, \ldots, x_T)$ terminated by an EOS token, with conditional distributions

$$\pi^{(j)}(m \mid s) \;=\; \prod_{u=1}^{T(m)} \pi^{(j)}(x_u \mid s, x_{<u}).$$

Define $\pi'^{(j)}$ similarly.

**Lemma B.3** (Chain rule (reverse): message KL equals expected sum of token KLs)**.** *For any $s$,*

$$D_{\mathrm{KL}}\big(\pi^{(j)}(\cdot \mid s) \| \pi'^{(j)}(\cdot \mid s)\big) = \mathbb{E}_{m \sim \pi^{(j)}(\cdot | s)}\left[\sum_{u=1}^{T(m)} D_{\mathrm{KL}}\big(\pi^{(j)}(\cdot \mid s, x_{<u}) \| \pi'^{(j)}(\cdot \mid s, x_{<u})\big)\right].$$

*Proof.* Write

$$\log \frac{\pi^{(j)}(m \mid s)}{\pi'^{(j)}(m \mid s)} = \sum_{u=1}^{T(m)} \log \frac{\pi^{(j)}(x_u \mid s, x_{<u})}{\pi'^{(j)}(x_u \mid s, x_{<u})}.$$

Taking expectation over $m \sim \pi^{(j)}(\cdot \mid s)$ gives the KL on the left. Then apply the tower property to recognize each term as a conditional KL. $\qquad\square$

**Practical estimator.** Lemma B.3 shows that summing per-token reverse KL along tokens *sampled from the reference/current policy* yields an unbiased estimate of the reverse message-level KL at that state. This aligns with TeamTR's per-step trust-region monitoring during rollouts from $\hat{\pi}^{i-1}$.

**Lemma B.4** (Unweighted token-average vs. token-sum (explicit constants)). *Assume message lengths are bounded:* $1 \leq T(m) \leq T_{\max}$ *almost surely under* $\pi^{(j)}(\cdot \mid s)$*. Let* $\kappa_u(m, s) \geq 0$ *denote the token KL at position* $u$ *as in Lemma B.3. Define the token-sum and token-average random variables*

$$S(m, s) := \sum_{u=1}^{T(m)} \kappa_u(m, s), \qquad A(m, s) := \frac{1}{T(m)} \sum_{u=1}^{T(m)} \kappa_u(m, s).$$

*Then* $S(m, s) = T(m) A(m, s)$ *and hence*

$$\mathbb{E}[S(m, s)] \leq T_{\max} \mathbb{E}[A(m, s)], \qquad \mathbb{E}[S(m, s)] \geq \mathbb{E}[A(m, s)].$$

*Consequently, enforcing an average-token cap* $\mathbb{E}[A(m, s)] \leq \bar{\delta}$ *implies a message-KL cap* $D_{\mathrm{KL}}(\pi^{(j)}(\cdot \mid s) \| \pi'^{(j)}(\cdot \mid s)) \leq T_{\max} \bar{\delta}$.

*Proof.* Fix a state $s$ and draw a message $m \sim \pi^{(j)}(\cdot \mid s)$. By assumption, the random length $T(m)$ satisfies $1 \leq T(m) \leq T_{\max}$ almost surely. Moreover, $\kappa_u(m, s) \geq 0$ for every $u$, hence

$$S(m, s) = \sum_{u=1}^{T(m)} \kappa_u(m, s) \geq 0, \qquad A(m, s) = \frac{1}{T(m)} \sum_{u=1}^{T(m)} \kappa_u(m, s) \geq 0.$$

By definition, for every realized $(m, s)$ with $T(m) \geq 1$,

$$S(m, s) = T(m) A(m, s).$$

Using the bounds on $T(m)$ and nonnegativity of $A(m, s)$, we obtain the following *pointwise* inequalities:

$$A(m, s) \ \leq \ S(m, s) \ = \ T(m) A(m, s) \ \leq \ T_{\max} A(m, s).$$

Taking expectation with respect to $m \sim \pi^{(j)}(\cdot \mid s)$ (equivalently, with respect to the joint token-generation randomness underlying $m$) and using monotonicity/linearity of expectation yields

$$\mathbb{E}[S(m, s)] \leq T_{\max} \mathbb{E}[A(m, s)], \qquad \mathbb{E}[S(m, s)] \geq \mathbb{E}[A(m, s)].$$

(These inequalities remain valid in the extended reals even if some KL terms are $+\infty$.)

Finally, by Lemma B.3 (the chain rule decomposition of message-level KL into token-level KLs),

$$D_{\mathrm{KL}}\big(\pi^{(j)}(\cdot \mid s) \| \pi'^{(j)}(\cdot \mid s)\big) = \mathbb{E}_{m \sim \pi^{(j)}(\cdot \mid s)} \left[ \sum_{u=1}^{T(m)} \kappa_u(m, s) \right] = \mathbb{E}[S(m, s)].$$

Therefore, enforcing an average-token cap $\mathbb{E}[A(m, s)] \leq \bar{\delta}$ implies

$$D_{\mathrm{KL}}\big(\pi^{(j)}(\cdot \mid s) \| \pi'^{(j)}(\cdot \mid s)\big) = \mathbb{E}[S(m, s)] \leq T_{\max} \mathbb{E}[A(m, s)] \leq T_{\max} \bar{\delta},$$

as claimed. $\square$

**Lemma B.5** (From expected (token) KL to an occupancy shift penalty). *Let* $\bar{\delta} = D_{\mathrm{KL}_{\mathrm{tok}}^{\pi}}(\pi \| \pi')$ *with reference* $\pi$*. For any bounded* $f$,

$$\left| \mathbb{E}_{d^{\pi'}}[f] - \mathbb{E}_{d^{\pi}}[f] \right| \leq \frac{2\gamma}{1-\gamma} \sqrt{\frac{\bar{\delta}}{2}} \|f\|_\infty.$$

*Proof.* Let $f : \mathcal{S} \to \mathbb{R}$ be bounded, and denote $\|f\|_\infty = \sup_s |f(s)|$. By Lemma B.8, we have an occupancy-shift bound of the form

$$\left| \mathbb{E}_{d^{\pi'}}[f] - \mathbb{E}_{d^\pi}[f] \right| \leq \frac{2\gamma}{1-\gamma} \|f\|_\infty \, \mathbb{E}_{s \sim d^\pi}\Big[ D_{\mathrm{TV}}\big(\pi(\cdot \mid s), \pi'(\cdot \mid s)\big) \Big]. \tag{13}$$

Next, apply Pinsker's inequality pointwise in $s$:

$$D_{\mathrm{TV}}\big(\pi(\cdot \mid s), \pi'(\cdot \mid s)\big) \leq \sqrt{\frac{1}{2} D_{\mathrm{KL}}\big(\pi(\cdot \mid s) \| \pi'(\cdot \mid s)\big)}.$$

Taking expectation over $s \sim d^\pi$ and using Jensen's inequality (since $x \mapsto \sqrt{x}$ is concave on $\mathbb{R}_+$) gives

$$\mathbb{E}_{s \sim d^\pi}\Big[ D_{\mathrm{TV}}\big(\pi(\cdot \mid s), \pi'(\cdot \mid s)\big) \Big] \leq \mathbb{E}_{s \sim d^\pi}\left[ \sqrt{\frac{1}{2} D_{\mathrm{KL}}\big(\pi(\cdot \mid s) \| \pi'(\cdot \mid s)\big)} \right]$$

$$\leq \sqrt{\frac{1}{2} \mathbb{E}_{s \sim d^\pi} D_{\mathrm{KL}}\big(\pi(\cdot \mid s) \| \pi'(\cdot \mid s)\big)}.$$

Finally, by Lemma B.2 with reference policy $\rho = \pi$,

$$\mathbb{E}_{s \sim d^\pi} D_{\mathrm{KL}}\big(\pi(\cdot \mid s) \| \pi'(\cdot \mid s)\big) = D_{\mathrm{KL\,tok}}^{\pi}(\pi \| \pi') = \bar{\delta}.$$

Plugging the above into (13) yields

$$\left| \mathbb{E}_{d^{\pi'}}[f] - \mathbb{E}_{d^\pi}[f] \right| \leq \frac{2\gamma}{1-\gamma} \sqrt{\frac{\bar{\delta}}{2}} \|f\|_\infty,$$

which proves the lemma. $\qquad\square$

**Lemma B.6** (Token-level KL reduction under a single-factor update (reverse)). *At step $i$, $\hat{\pi}^{\,i}$ and $\hat{\pi}^{\,i-1}$ differ only in factor $\sigma(i)$. With reference $\hat{\pi}^{\,i-1}$,*

$$D_{\mathrm{KL\,tok}}^{\hat{\pi}^{\,i-1}}\big(\hat{\pi}^{\,i-1} \| \hat{\pi}^{\,i}\big) = D_{\mathrm{KL\,tok}}^{\hat{\pi}^{\,i-1}}\Big(\pi_{\mathrm{cur}}^{\sigma(i)} \| \pi_{\mathrm{tar}}^{\sigma(i)}\Big).$$

*Proof.* Let $k := \sigma(i)$. We view the (joint) action/message space as a product $\mathcal{M} = \prod_{j=1}^J \mathcal{M}^{(j)}$, and write $m = (m^k, m^{-k})$ for the $k$-th component and the remaining components. By assumption, $\hat{\pi}^{\,i}$ and $\hat{\pi}^{\,i-1}$ differ only in factor $k$, i.e., for every state $s$,

$$\hat{\pi}^{\,i-1}(\mathrm{d}m \mid s) = \pi_{\mathrm{cur}}^k(\mathrm{d}m^k \mid s) \otimes \Big( \bigotimes_{j \neq k} \pi^{(j)}(\mathrm{d}m^{(j)} \mid s) \Big),$$

$$\hat{\pi}^{\,i}(\mathrm{d}m \mid s) = \pi_{\mathrm{tar}}^k(\mathrm{d}m^k \mid s) \otimes \Big( \bigotimes_{j \neq k} \pi^{(j)}(\mathrm{d}m^{(j)} \mid s) \Big),$$

where all factors with $j \neq k$ are identical.

Fix an arbitrary $s$. If $\pi_{\mathrm{cur}}^k(\cdot \mid s) \not\ll \pi_{\mathrm{tar}}^k(\cdot \mid s)$, then also $\hat{\pi}^{\,i-1}(\cdot \mid s) \not\ll \hat{\pi}^{\,i}(\cdot \mid s)$ (because the common product factor on $m^{-k}$ cannot restore absolute continuity), hence both $D_{\mathrm{KL}}(\hat{\pi}^{\,i-1}(\cdot \mid s) \| \hat{\pi}^{\,i}(\cdot \mid s))$ and $D_{\mathrm{KL}}(\pi_{\mathrm{cur}}^k(\cdot \mid s) \| \pi_{\mathrm{tar}}^k(\cdot \mid s))$ equal $+\infty$, and the desired equality holds.

Otherwise, $\pi_{\mathrm{cur}}^k(\cdot \mid s) \ll \pi_{\mathrm{tar}}^k(\cdot \mid s)$ and thus $\hat{\pi}^{\,i-1}(\cdot \mid s) \ll \hat{\pi}^{\,i}(\cdot \mid s)$. By the Radon–Nikodym rule for product measures, the likelihood ratio cancels all identical factors:

$$\frac{\mathrm{d}\hat{\pi}^{\,i-1}(\cdot \mid s)}{\mathrm{d}\hat{\pi}^{\,i}(\cdot \mid s)}(m) = \frac{\mathrm{d}\pi_{\mathrm{cur}}^k(\cdot \mid s)}{\mathrm{d}\pi_{\mathrm{tar}}^k(\cdot \mid s)}(m^k).$$

Therefore,

$$
\begin{aligned}
D_{\mathrm{KL}}\big(\hat{\pi}^{\,i-1}(\cdot \mid s)\|\hat{\pi}^{\,i}(\cdot \mid s)\big) &= \int_{\mathcal{M}} \log\!\left(\frac{\mathrm{d}\hat{\pi}^{\,i-1}}{\mathrm{d}\hat{\pi}^{\,i}}(m)\right) \hat{\pi}^{\,i-1}(\mathrm{d}m \mid s) \\
&= \int_{\mathcal{M}} \log\!\left(\frac{\mathrm{d}\pi_{\mathrm{cur}}^{k}}{\mathrm{d}\pi_{\mathrm{tar}}^{k}}(m^{k})\right) \pi_{\mathrm{cur}}^{k}(\mathrm{d}m^{k} \mid s)\left(\bigotimes_{j\neq k} \pi^{(j)}(\mathrm{d}m^{(j)} \mid s)\right) \\
&= \int_{\mathcal{M}^{(k)}} \log\!\left(\frac{\mathrm{d}\pi_{\mathrm{cur}}^{k}}{\mathrm{d}\pi_{\mathrm{tar}}^{k}}(m^{k})\right) \pi_{\mathrm{cur}}^{k}(\mathrm{d}m^{k} \mid s) \\
&= D_{\mathrm{KL}}\big(\pi_{\mathrm{cur}}^{k}(\cdot \mid s)\|\pi_{\mathrm{tar}}^{k}(\cdot \mid s)\big),
\end{aligned}
$$

where the third line integrates out the common factor on $m^{-k}$ (it is a probability measure, so its total mass is 1). (In the discrete case, the same conclusion follows by expanding the sum over $m = (m^{k}, m^{-k})$ and observing direct cancellation.)

Applying Lemma B.2 with reference $\rho = \hat{\pi}^{\,i-1}$, we obtain

$$
\begin{aligned}
D_{\mathrm{KL}_{\mathrm{tok}}}^{\hat{\pi}^{\,i-1}}\big(\hat{\pi}^{\,i-1}\|\hat{\pi}^{\,i}\big) &= \mathbb{E}_{s\sim d^{\hat{\pi}^{\,i-1}}} D_{\mathrm{KL}}\big(\hat{\pi}^{\,i-1}(\cdot \mid s)\|\hat{\pi}^{\,i}(\cdot \mid s)\big) \\
&= \mathbb{E}_{s\sim d^{\hat{\pi}^{\,i-1}}} D_{\mathrm{KL}}\big(\pi_{\mathrm{cur}}^{k}(\cdot \mid s)\|\pi_{\mathrm{tar}}^{k}(\cdot \mid s)\big) \\
&= D_{\mathrm{KL}_{\mathrm{tok}}}^{\hat{\pi}^{\,i-1}}\big(\pi_{\mathrm{cur}}^{k}\|\pi_{\mathrm{tar}}^{k}\big),
\end{aligned}
$$

where the last line uses the same definition of $D_{\mathrm{KL}_{\mathrm{tok}}}^{\rho}$ but applied to the $k$-th factor, with the *same* reference occupancy distribution $d^{\rho}$ (here $\rho = \hat{\pi}^{\,i-1}$). This proves the lemma. $\qquad\square$

### B.3. Occupancy Shift Bounds

This section proves Lemma 3.2 from the main text and provides both max-divergence and expected-divergence variants. All statements apply to the macro-action MDP; i.e., $\pi(\cdot \mid s)$ denotes the (joint) macro-action distribution at state $s$.

**Lemma B.7** (Occupancy shift via $D_{\mathrm{TV}}^{\max}$). *For any policies $\pi', \pi$ and bounded $f : \mathcal{S} \to \mathbb{R}$,*

$$
\big|\mathbb{E}_{d^{\pi'}}[f] - \mathbb{E}_{d^{\pi}}[f]\big| \leq \frac{2\gamma}{1-\gamma}\, D_{\mathrm{TV}}^{\max}(\pi'\|\pi)\, \|f\|_{\infty}.
$$

*Proof.* The proof follows the standard resolvent argument. Let $P_{\pi}(s' \mid s) = \sum_{\mathbf{a}} \pi(\mathbf{a} \mid s) P(s' \mid s, \mathbf{a})$ be the induced state transition kernel. The discounted visitation satisfies $d^{\pi} = (1-\gamma)\mu + \gamma P_{\pi}^{\top} d^{\pi}$. Taking the difference yields

$$
(I - \gamma P_{\pi'}^{\top})(d^{\pi'} - d^{\pi}) = \gamma(P_{\pi'}^{\top} - P_{\pi}^{\top})d^{\pi}.
$$

Since $\|(I - \gamma P_{\pi'}^{\top})^{-1}\|_{1\to 1} \leq \frac{1}{1-\gamma}$,

$$
\|d^{\pi'} - d^{\pi}\|_{1} \leq \frac{\gamma}{1-\gamma}\|(P_{\pi'}^{\top} - P_{\pi}^{\top})d^{\pi}\|_{1}.
$$

For each state $s$,

$$
\|P_{\pi'}(\cdot \mid s) - P_{\pi}(\cdot \mid s)\|_{1} \leq \sum_{\mathbf{a}} |\pi'(\mathbf{a} \mid s) - \pi(\mathbf{a} \mid s)| \cdot \|P(\cdot \mid s, \mathbf{a})\|_{1} = 2\, D_{\mathrm{TV}}(\pi'(\cdot \mid s), \pi(\cdot \mid s)).
$$

Thus $\|(P_{\pi'}^{\top} - P_{\pi}^{\top})d^{\pi}\|_{1} \leq 2\, D_{\mathrm{TV}}^{\max}(\pi'\|\pi)$. Finally,

$$
\big|\mathbb{E}_{d^{\pi'}}[f] - \mathbb{E}_{d^{\pi}}[f]\big| \leq \|f\|_{\infty}\|d^{\pi'} - d^{\pi}\|_{1} \leq \frac{2\gamma}{1-\gamma}D_{\mathrm{TV}}^{\max}(\pi'\|\pi)\|f\|_{\infty}.
$$

$\qquad\square$

**Lemma B.8** (Occupancy shift via expected $D_{\mathrm{TV}}$). *For any policies $\pi', \pi$ and bounded $f$,*

$$
\big|\mathbb{E}_{d^{\pi'}}[f] - \mathbb{E}_{d^{\pi}}[f]\big| \leq \frac{2\gamma}{1-\gamma}\, \mathbb{E}_{s\sim d^{\pi}}[D_{\mathrm{TV}}(\pi'(\cdot \mid s), \pi(\cdot \mid s))]\, \|f\|_{\infty}.
$$

*Consequently, by Pinsker and Jensen,*

$$
\big|\mathbb{E}_{d^{\pi'}}[f] - \mathbb{E}_{d^{\pi}}[f]\big| \leq \frac{2\gamma}{1-\gamma}\, \sqrt{\frac{1}{2}\mathbb{E}_{s\sim d^{\pi}} D_{\mathrm{KL}}(\pi(\cdot \mid s)\|\pi'(\cdot \mid s))}\, \|f\|_{\infty}.
$$

*Proof.* We follow the same resolvent identity as in Lemma B.7 but do not take a supremum over $s$.

From the fixed-point equations,

$$(I - \gamma P_{\pi'}^\top)(d^{\pi'} - d^\pi) = \gamma(P_{\pi'}^\top - P_\pi^\top)d^\pi,$$

hence

$$\|d^{\pi'} - d^\pi\|_1 \le \frac{\gamma}{1 - \gamma}\|(P_{\pi'}^\top - P_\pi^\top)d^\pi\|_1.$$

Now,

$$\|(P_{\pi'}^\top - P_\pi^\top)d^\pi\|_1 = \sum_{s'}\left|\sum_s d^\pi(s)\big(P_{\pi'}(s' \mid s) - P_\pi(s' \mid s)\big)\right|$$

$$\le \sum_s d^\pi(s)\sum_{s'}|P_{\pi'}(s' \mid s) - P_\pi(s' \mid s)|$$

$$= \mathbb{E}_{s \sim d^\pi}\left[\|P_{\pi'}(\cdot \mid s) - P_\pi(\cdot \mid s)\|_1\right].$$

As in Lemma B.7, for each $s$ we have $\|P_{\pi'}(\cdot \mid s) - P_\pi(\cdot \mid s)\|_1 \le 2\,D_{\mathrm{TV}}(\pi'(\cdot \mid s), \pi(\cdot \mid s))$, thus

$$\|d^{\pi'} - d^\pi\|_1 \le \frac{2\gamma}{1 - \gamma}\,\mathbb{E}_{s \sim d^\pi}\left[D_{\mathrm{TV}}(\pi'(\cdot \mid s), \pi(\cdot \mid s))\right].$$

Finally, $|\mathbb{E}_{d^{\pi'}}[f] - \mathbb{E}_{d^\pi}[f]| \le \|f\|_\infty\|d^{\pi'} - d^\pi\|_1$ gives the first inequality.

For the second inequality, apply Pinsker pointwise: $D_{\mathrm{TV}}(P, Q) \le \sqrt{\frac{1}{2}D_{\mathrm{KL}}(P\|Q)}$, with $P = \pi(\cdot \mid s)$ and $Q = \pi'(\cdot \mid s)$, then use Jensen:

$$\mathbb{E}_{s \sim d^\pi}\left[D_{\mathrm{TV}}(\pi'(\cdot \mid s), \pi(\cdot \mid s))\right] \le \mathbb{E}_{s \sim d^\pi}\left[\sqrt{\tfrac{1}{2}D_{\mathrm{KL}}(\pi(\cdot \mid s)\|\pi'(\cdot \mid s))}\right] \le \sqrt{\tfrac{1}{2}\mathbb{E}_{s \sim d^\pi}D_{\mathrm{KL}}(\pi(\cdot \mid s)\|\pi'(\cdot \mid s))}.$$

$\square$

**Lemma B.9** (Cumulative within-stage occupancy shift). *Assume a sequence of intermediate policies $\hat{\pi}^0, \hat{\pi}^1, \ldots, \hat{\pi}^{i-1}$ satisfies $D_{\mathrm{KL}\,\mathrm{tok}}^{\hat{\pi}^{k-1}}(\hat{\pi}^{k-1}\|\hat{\pi}^k) \le \delta_k$ for $k = 1, \ldots, i-1$. Then for any bounded measurable $f$,*

$$\left|\mathbb{E}_{d^{\hat{\pi}^{i-1}}}[f] - \mathbb{E}_{d^{\hat{\pi}^0}}[f]\right| \le \frac{\sqrt{2}\gamma}{1 - \gamma}\|f\|_\infty\sum_{k=1}^{i-1}\sqrt{\delta_k}.$$

*Proof.* Fix any bounded measurable $f : \mathcal{S} \to \mathbb{R}$ and let $\|f\|_\infty = \sup_s|f(s)|$. Write the telescoping decomposition

$$\mathbb{E}_{d^{\hat{\pi}^{i-1}}}[f] - \mathbb{E}_{d^{\hat{\pi}^0}}[f] = \sum_{k=1}^{i-1}\left(\mathbb{E}_{d^{\hat{\pi}^k}}[f] - \mathbb{E}_{d^{\hat{\pi}^{k-1}}}[f]\right).$$

Taking absolute values and applying the triangle inequality gives

$$\left|\mathbb{E}_{d^{\hat{\pi}^{i-1}}}[f] - \mathbb{E}_{d^{\hat{\pi}^0}}[f]\right| \le \sum_{k=1}^{i-1}\left|\mathbb{E}_{d^{\hat{\pi}^k}}[f] - \mathbb{E}_{d^{\hat{\pi}^{k-1}}}[f]\right|. \tag{14}$$

For each $k \in \{1, \ldots, i-1\}$, apply Lemma B.8 in its Pinsker+Jensen form (to the adjacent pair $(\hat{\pi}^{k-1}, \hat{\pi}^k)$ with reference $\hat{\pi}^{k-1}$), yielding

$$\left|\mathbb{E}_{d^{\hat{\pi}^k}}[f] - \mathbb{E}_{d^{\hat{\pi}^{k-1}}}[f]\right| \le \frac{2\gamma}{1 - \gamma}\sqrt{\frac{1}{2}D_{\mathrm{KL}\,\mathrm{tok}}^{\hat{\pi}^{k-1}}(\hat{\pi}^{k-1}\|\hat{\pi}^k)}\,\|f\|_\infty.$$

Using the assumption $D_{\mathrm{KL}\,\mathrm{tok}}^{\hat{\pi}^{k-1}}(\hat{\pi}^{k-1}\|\hat{\pi}^k) \le \delta_k$ gives

$$\left|\mathbb{E}_{d^{\hat{\pi}^k}}[f] - \mathbb{E}_{d^{\hat{\pi}^{k-1}}}[f]\right| \le \frac{2\gamma}{1 - \gamma}\sqrt{\frac{\delta_k}{2}}\,\|f\|_\infty = \frac{\sqrt{2}\gamma}{1 - \gamma}\|f\|_\infty\,\sqrt{\delta_k}.$$

Substituting this bound into (14) and summing over $k = 1, \ldots, i-1$ yields

$$\left| \mathbb{E}_{d^{\hat{\pi}^{i-1}}}[f] - \mathbb{E}_{d^{\hat{\pi}^0}}[f] \right| \leq \frac{\sqrt{2}\gamma}{1-\gamma} \|f\|_\infty \sum_{k=1}^{i-1} \sqrt{\delta_k},$$

as claimed. (The inequality remains valid in the extended reals if some $\delta_k = +\infty$.) □

### B.4. Stale-occupancy evaluation: surrogate mismatch and compounding term

*Proof of Proposition 3.3.* The general within-stage occupancy shift bound in Eq. (5) in the main text is exactly Lemma B.9 (with the index shift $k = 1, \ldots, i-1$ corresponding to $k < i$).

For the surrogate mismatch bound in Eq. (7), define

$$f_i(s) := \mathbb{E}_{\mathbf{a} \sim \hat{\pi}^i(\cdot|s)}\left[\widehat{A}_{i-1}(s, \mathbf{a})\right].$$

If $|\widehat{A}_{i-1}(s, \mathbf{a})| \leq A_{\max}$ almost surely, then $\|f_i\|_\infty \leq A_{\max}$. By definitions of $L_i^{\mathrm{seq}}$ (Eq. (3) in the main text) and $L_i^{\mathrm{stale}}$ (Eq. (6)),

$$L_i^{\mathrm{seq}} - L_i^{\mathrm{stale}} = \frac{1}{1-\gamma}\left( \mathbb{E}_{s \sim d^{\hat{\pi}^{i-1}}}[f_i(s)] - \mathbb{E}_{s \sim d^{\hat{\pi}^0}}[f_i(s)] \right).$$

Apply Lemma B.9 to $f_i$ and multiply by $\frac{1}{1-\gamma}$ to obtain

$$\left| L_i^{\mathrm{seq}} - L_i^{\mathrm{stale}} \right| \leq \frac{\sqrt{2}\gamma}{(1-\gamma)^2} A_{\max} \sum_{k<i} \sqrt{\delta_k},$$

which is in Eq. (7). Summing over $i$ yields the stated $\sum_{i=1}^n \sum_{k<i} \sqrt{\delta_k}$ compounding term, and for $\delta_k \equiv \bar{\delta}$ the scaling is $O(n^2\sqrt{\bar{\delta}})$. □

## C. Certificates: Full Proofs and Order Dependence

### C.1. Single-Step Improvement Lower Bound (Full Proof)

*Proof of Theorem 3.4.* The performance difference lemma gives

$$J(\hat{\pi}^i) - J(\hat{\pi}^{i-1}) = \frac{1}{1-\gamma} \mathbb{E}_{s \sim d^{\hat{\pi}^i},\, \mathbf{a} \sim \hat{\pi}^i}\left[ A^{\hat{\pi}^{i-1}}(s, \mathbf{a}) \right].$$

Add and subtract the same quantity under $d^{\hat{\pi}^{i-1}}$:

$$J(\hat{\pi}^i) - J(\hat{\pi}^{i-1}) = \frac{1}{1-\gamma} \mathbb{E}_{s \sim d^{\hat{\pi}^{i-1}},\, \mathbf{a} \sim \hat{\pi}^i}\left[ A^{\hat{\pi}^{i-1}}(s, \mathbf{a}) \right]$$
$$+ \frac{1}{1-\gamma}\left( \mathbb{E}_{s \sim d^{\hat{\pi}^i},\, \mathbf{a} \sim \hat{\pi}^i}\left[ A^{\hat{\pi}^{i-1}} \right] - \mathbb{E}_{s \sim d^{\hat{\pi}^{i-1}},\, \mathbf{a} \sim \hat{\pi}^i}\left[ A^{\hat{\pi}^{i-1}} \right] \right).$$

Define $f(s) := \mathbb{E}_{\mathbf{a} \sim \hat{\pi}^i(\cdot|s)}[A^{\hat{\pi}^{i-1}}(s, \mathbf{a})]$. Since $|A^{\hat{\pi}^{i-1}}(s, \mathbf{a})| \leq A_{\max}$, we have $\|f\|_\infty \leq A_{\max}$. By Lemma B.8 and Lemma B.2 (with $\pi = \hat{\pi}^{i-1}$ and $\pi' = \hat{\pi}^i$),

$$\left| \mathbb{E}_{d^{\hat{\pi}^i}}[f] - \mathbb{E}_{d^{\hat{\pi}^{i-1}}}[f] \right| \leq \frac{2\gamma}{1-\gamma} \sqrt{\frac{1}{2} D_{\mathrm{KL}\,\mathrm{tok}}^{\hat{\pi}^{i-1}}(\hat{\pi}^{i-1}\|\hat{\pi}^i)} \cdot A_{\max}.$$

Therefore,

$$J(\hat{\pi}^i) - J(\hat{\pi}^{i-1}) \geq \frac{1}{1-\gamma} \mathbb{E}_{s \sim d^{\hat{\pi}^{i-1}},\, \mathbf{a} \sim \hat{\pi}^i}\left[ A^{\hat{\pi}^{i-1}}(s, \mathbf{a}) \right] - \frac{\sqrt{2}\gamma}{(1-\gamma)^2} A_{\max} \sqrt{D_{\mathrm{KL}\,\mathrm{tok}}^{\hat{\pi}^{i-1}}(\hat{\pi}^{i-1}\|\hat{\pi}^i)}.$$

For the main term, add and subtract the estimator $\widehat{A}_{i-1}$:

$$\mathbb{E}[A^{\hat{\pi}^{i-1}}] = \mathbb{E}[\widehat{A}_{i-1}] + \mathbb{E}[A^{\hat{\pi}^{i-1}} - \widehat{A}_{i-1}].$$

By the definition of $\zeta_i$ (Notation; Eq. (4) in the main text),

$$\mathbb{E}_{s \sim d^{\hat{\pi}^{i-1}}, \mathbf{a} \sim \hat{\pi}^i}[A^{\hat{\pi}^{i-1}} - \widehat{A}_{i-1}] \geq -\zeta_i.$$

Finally, enforce $D_{\mathrm{KL}_{\mathrm{tok}}}^{\hat{\pi}^{i-1}}(\hat{\pi}^{i-1} \| \hat{\pi}^i) \leq \delta_i$ via Lemma B.6 to obtain the claim. $\qquad\square$

## C.2. Joint-Stage Improvement and Order Dependence

*Proof of Lemma 3.5.* At step $i$, $\hat{\pi}^i$ and $\hat{\pi}^{i-1}$ differ only in factor $\sigma(i)$. For fixed $s$, let $\psi^{(k)}(\cdot \mid s)$ denote the (common) factor of agent $k \neq \sigma(i)$ in both $\hat{\pi}^{i-1}$ and $\hat{\pi}^i$. Then

$$\hat{\pi}^{i-1}(\mathbf{a} \mid s) = \pi_{\mathrm{cur}}^{\sigma(i)}(a_{\sigma(i)} \mid s) \prod_{k \neq \sigma(i)} \psi^{(k)}(a_k \mid s), \qquad \hat{\pi}^i(\mathbf{a} \mid s) = \pi_{\mathrm{tar}}^{\sigma(i)}(a_{\sigma(i)} \mid s) \prod_{k \neq \sigma(i)} \psi^{(k)}(a_k \mid s).$$

Taking KL and expanding the definition,

$$
\begin{aligned}
D_{\mathrm{KL}}\big(\hat{\pi}^{i-1}(\cdot \mid s) \| \hat{\pi}^i(\cdot \mid s)\big) &= \sum_{\mathbf{a}} \hat{\pi}^{i-1}(\mathbf{a} \mid s) \log \frac{\hat{\pi}^{i-1}(\mathbf{a} \mid s)}{\hat{\pi}^i(\mathbf{a} \mid s)} \\
&= \sum_{\mathbf{a}} \hat{\pi}^{i-1}(\mathbf{a} \mid s) \log \frac{\pi_{\mathrm{cur}}^{\sigma(i)}(a_{\sigma(i)} \mid s)}{\pi_{\mathrm{tar}}^{\sigma(i)}(a_{\sigma(i)} \mid s)} \\
&= \sum_{a_{\sigma(i)}} \pi_{\mathrm{cur}}^{\sigma(i)}(a_{\sigma(i)} \mid s) \log \frac{\pi_{\mathrm{cur}}^{\sigma(i)}(a_{\sigma(i)} \mid s)}{\pi_{\mathrm{tar}}^{\sigma(i)}(a_{\sigma(i)} \mid s)} \\
&= D_{\mathrm{KL}}\big(\pi_{\mathrm{cur}}^{\sigma(i)}(\cdot \mid s) \| \pi_{\mathrm{tar}}^{\sigma(i)}(\cdot \mid s)\big),
\end{aligned}
$$

where the third line uses $\sum_{\mathbf{a}_{-\sigma(i)}} \prod_{k \neq \sigma(i)} \psi^{(k)}(a_k \mid s) = 1$. Taking expectation over $s \sim d^{\hat{\pi}^{i-1}}$ and using Lemma B.2 yields

$$D_{\mathrm{KL}_{\mathrm{tok}}}^{\hat{\pi}^{i-1}}\big(\hat{\pi}^{i-1} \| \hat{\pi}^i\big) = D_{\mathrm{KL}_{\mathrm{tok}}}^{\hat{\pi}^{i-1}}\big(\pi_{\mathrm{cur}}^{\sigma(i)} \| \pi_{\mathrm{tar}}^{\sigma(i)}\big) \leq \delta_i.$$

$\qquad\square$

*Proof of Theorem 3.6.* Apply Theorem 3.4 to each step $i$ and sum over $i = 1, \ldots, n$. The left-hand side telescopes:

$$\sum_{i=1}^n \big(J(\hat{\pi}^i) - J(\hat{\pi}^{i-1})\big) = J(\hat{\pi}^n) - J(\hat{\pi}^0) = J(\bar{\pi}) - J(\pi_{\mathrm{cur}}).$$

The inequality holds for any update order $\sigma$ because it is applied to the realized sequence of intermediate policies. However, the intermediate occupancies $d^{\hat{\pi}^{i-1}}$ and hence $L_i^{\mathrm{seq}}$ and $\zeta_i$ may depend on $\sigma$. $\qquad\square$

## C.3. Stage-wise bound under stale-occupancy surrogates

*Proof of Theorem 3.7.* Start from the intermediate-occupancy stage certificate (Theorem 3.6):

$$J(\bar{\pi}) - J(\pi_{\mathrm{cur}}) \geq \sum_{i=1}^n L_i^{\mathrm{seq}} - \frac{\sqrt{2}\gamma}{(1-\gamma)^2} A_{\max} \sum_{i=1}^n \sqrt{\delta_i} - \frac{1}{1-\gamma} \sum_{i=1}^n \zeta_i.$$

By Proposition 3.3 (Eq. (7)),

$$L_i^{\mathrm{seq}} \geq L_i^{\mathrm{stale}} - \frac{\sqrt{2}\gamma}{(1-\gamma)^2} A_{\max} \sum_{k<i} \sqrt{\delta_k}.$$

Summing over $i = 1, \ldots, n$ gives

$$\sum_{i=1}^n L_i^{\mathrm{seq}} \geq \sum_{i=1}^n L_i^{\mathrm{stale}} - \frac{\sqrt{2}\gamma}{(1-\gamma)^2} A_{\max} \sum_{i=1}^n \sum_{k<i} \sqrt{\delta_k}.$$

Substitute this into Theorem 3.6 to obtain Eq. (8). $\qquad\square$

# D. Plug-and-Play Upgrades and Stage-0 Alignment

## D.1. Stage-0 alignment as reverse-KL projection / distillation

This section specifies a practical Stage-0 alignment objective that is *compatible with the reverse-KL trust region*. We assume that the upgraded agent shares the same tokenization/vocabulary as the replaced agent, so that KL is well defined. (If tokenizers differ, a shared action space must be introduced; we leave this to future work.)

**Goal.** Suppose agent $j$ is replaced: we have the old policy $\pi_{\text{old}}^{(j)}$ and a new parameterization $\pi_{\theta}^{(j)}$. We aim to select $\theta$ such that the new agent lies within a reverse-KL trust region around the old agent in representative contexts.

**Probe distribution.** Let $\nu$ be a distribution over states/contexts where the replaced agent is expected to act (e.g., contexts collected by running the *pre-swap* team on a probe prompt set, and extracting states where $j$ is active). Define the Stage-0 target constraint

$$\mathbb{E}_{s\sim\nu} D_{\text{KL}}\big(\pi_{\text{old}}^{(j)}(\cdot \mid s)\|\pi_{\theta}^{(j)}(\cdot \mid s)\big) \leq \delta_{\text{align}}.$$

**Reverse-KL projection objective.** A natural alignment objective is to minimize the left-hand side:

$$\min_{\theta} \ \mathbb{E}_{s\sim\nu} D_{\text{KL}}\big(\pi_{\text{old}}^{(j)}(\cdot \mid s)\|\pi_{\theta}^{(j)}(\cdot \mid s)\big). \tag{15}$$

This is equivalent (up to an additive constant independent of $\theta$) to minimizing the cross-entropy under the teacher distribution:

$$\min_{\theta} \ \mathbb{E}_{s\sim\nu}\mathbb{E}_{a\sim\pi_{\text{old}}^{(j)}(\cdot|s)}[-\log \pi_{\theta}^{(j)}(a \mid s)].$$

**Autoregressive implementation.** By the reverse chain rule (Lemma B.3), the message-level reverse KL decomposes into a token-level sum with expectation under the old policy. Thus, if we collect teacher-forced rollouts/messages from $\pi_{\text{old}}^{(j)}$ on contexts $s \sim \nu$, we can optimize Eq. (15) via standard distillation:

$$\min_{\theta} \ \mathbb{E}_{s\sim\nu, \ m\sim\pi_{\text{old}}^{(j)}(\cdot|s)} \left[ \sum_{u=1}^{T(m)} D_{\text{KL}}\big(\pi_{\text{old}}^{(j)}(\cdot \mid s, x_{<u})\|\pi_{\theta}^{(j)}(\cdot \mid s, x_{<u})\big) \right].$$

**Stopping rule.** During alignment, we monitor the empirical token-sum reverse KL monitor in Eq. (11) on the probe set and early-stop once it falls below $\delta_{\text{align}}$. After this, TeamTR resumes with the standard intermediate-occupancy updates.

**Scope note.** Stage-0 alignment enforces the trust region on the probe distribution $\nu$; it does not, by itself, guarantee a trust region on future intermediate occupancies. Empirically, we find that aligning on representative contexts substantially reduces the swap shock and improves resumability.

## D.2. Certified Resumability and Certificate Tightening

*Proof of Proposition 3.8.* The proofs of Theorem 3.4 and Theorem 3.6 depend only on: the surrogate quantities $L_i^{\text{seq}}$, the trust-region radii $\delta_i$ through the token-KL term, and the surrogate-estimation error bounds $\zeta_i$. They do not depend on how $\pi_{\text{tar}}^{\sigma(i)}$ is parameterized. Therefore, after replacing an agent and re-establishing the trust-region bookkeeping for subsequent updates, the same lower-bound form continues to apply. $\square$

*Proof of Proposition 3.9.* The stage certificate in Theorem 3.6 takes the form

$$\text{LB}(\{L_i^{\text{seq}}\}, \{\delta_i\}, \{\zeta_i\}) = \sum_{i=1}^{n} L_i^{\text{seq}} - c \sum_{i=1}^{n} \sqrt{\delta_i} - \frac{1}{1-\gamma} \sum_{i=1}^{n} \zeta_i, \qquad c = \frac{\sqrt{2}\gamma}{(1-\gamma)^2} A_{\max}.$$

If, after an upgrade, some step attains a higher surrogate value with the same $\delta_i$ (and all other terms unchanged), the right-hand side increases. Likewise, achieving the same surrogate value with a smaller $\delta_i$ decreases the penalty term $c\sqrt{\delta_i}$ and hence weakly increases the bound. $\square$

# E. Group-Based Advantages and Finite-Sample Concentration

## E.1. Group-Standardized and Clipped Message-Level Advantages

This section specifies the message- and sequence-level advantage estimator used in our algorithm. It is inspired by group-based baselines in LLM RL methods, but here it is used as a plug-in estimator inside a sequential trust-region framework.

### E.1.1. DEFINITION: GROUP STANDARDIZATION AND HARD CLIPPING

Fix a prompt (or initial state) $x$. At step $i$, we sample $G$ rollouts $\{\tau^{(g)}\}_{g=1}^{G}$ from the current intermediate policy $\hat{\pi}^{i-1}(\cdot \mid x)$. Let $\widehat{R}^{(g)}$ denote the scalar message-/sequence-level return for $\tau^{(g)}$ (e.g., terminal verifier reward). Define the group mean and standard deviation

$$\widehat{\mu} := \frac{1}{G} \sum_{g=1}^{G} \widehat{R}^{(g)}, \qquad \widehat{\sigma} := \sqrt{\frac{1}{G} \sum_{g=1}^{G} \left(\widehat{R}^{(g)} - \widehat{\mu}\right)^2 + \varepsilon_{\mathrm{std}}}.$$

The standardized, hard-clipped group advantage is

$$\widehat{A}_{\mathrm{GRP}}^{i-1}(\tau^{(g)}) := \mathrm{clip}\left(\frac{\widehat{R}^{(g)} - \widehat{\mu}}{\widehat{\sigma}}, -A_{\mathrm{clip}}, A_{\mathrm{clip}}\right).$$

When used as a token-level weight, $\widehat{A}_{\mathrm{GRP}}^{i-1}(\tau^{(g)})$ is broadcast to all tokens controlled by the updated factor.

**Boundedness.** By construction, $|\widehat{A}_{\mathrm{GRP}}^{i-1}(\tau^{(g)})| \leq A_{\mathrm{clip}}$ almost surely.

## E.2. Bias components aligned with $\zeta_i$

The main text term $\zeta_i$ is intended to capture (or upper bound) the aggregate surrogate-estimation error. This section records simple, explicit bias components that can be upper-bounded and monitored.

**Lemma E.1** (Self-included group mean yields shrinkage for i.i.d. rollouts). *Fix a prompt $x$ and let $\tau^{(1)}, \ldots, \tau^{(G)}$ be i.i.d. from $\hat{\pi}^{i-1}(\cdot \mid x)$. Let $R^{(g)} := R(\tau^{(g)})$ be a measurable return with $\mathbb{E}[|R^{(g)}|] < \infty$, and define $\widehat{\mu} = \frac{1}{G} \sum_{g=1}^{G} R^{(g)}$. Let $\mu(x) := \mathbb{E}[R(\tau) \mid x]$. Then for any fixed index $g$,*

$$\mathbb{E}\left[R^{(g)} - \widehat{\mu} \mid x, \tau^{(g)}\right] = \left(1 - \frac{1}{G}\right)\left(R^{(g)} - \mu(x)\right).$$

*Proof.* Write $\widehat{\mu} = \frac{1}{G} R^{(g)} + \frac{1}{G} \sum_{h \neq g} R^{(h)}$. Conditioned on $(x, \tau^{(g)})$, the random variables $\{R^{(h)}\}_{h \neq g}$ remain i.i.d. with $\mathbb{E}[R^{(h)} \mid x, \tau^{(g)}] = \mathbb{E}[R^{(h)} \mid x] = \mu(x)$ by independence and identical distribution. Hence

$$\mathbb{E}[\widehat{\mu} \mid x, \tau^{(g)}] = \frac{1}{G} R^{(g)} + \frac{G-1}{G} \mu(x),$$

and therefore

$$\mathbb{E}[R^{(g)} - \widehat{\mu} \mid x, \tau^{(g)}] = R^{(g)} - \left(\frac{1}{G} R^{(g)} + \frac{G-1}{G} \mu(x)\right) = \left(1 - \frac{1}{G}\right)(R^{(g)} - \mu(x)).$$

$\square$

**Corollary E.2** (A simple $O(1/G)$ bias proxy for group-mean baselines). *Under Lemma E.1, define the centered return advantage $A_{\mathrm{ret}}(\tau; x) := R(\tau) - \mathbb{E}[R(\tau) \mid x]$. Then for any fixed $g$,*

$$\left|\mathbb{E}\left[R^{(g)} - \widehat{\mu} \mid x, \tau^{(g)}\right] - A_{\mathrm{ret}}(\tau^{(g)}; x)\right| = \frac{1}{G} |A_{\mathrm{ret}}(\tau^{(g)}; x)|.$$

*If $|A_{\mathrm{ret}}(\tau; x)| \leq A_{\mathrm{ret,max}}$ almost surely, then the baseline-induced bias is at most $A_{\mathrm{ret,max}}/G$.*

*Proof.* Lemma E.1 implies $\mathbb{E}[R^{(g)} - \widehat{\mu} \mid x, \tau^{(g)}] = (1 - \frac{1}{G})A_{\mathrm{ret}}(\tau^{(g)}; x)$. Subtracting $A_{\mathrm{ret}}(\tau^{(g)}; x)$ and taking absolute values yields the claim. $\qquad\square$

**Lemma E.3** (Clipping bias is controlled by the overflow probability). *Let $Z$ be any random variable and define $\tilde{Z} = \mathrm{clip}(Z, -c, c)$. Then*

$$|\mathbb{E}[\tilde{Z}] - \mathbb{E}[Z]| \leq \mathbb{E}\left[|Z - \tilde{Z}|\right] \leq \mathbb{E}[|Z|\ \mathbb{1}\{|Z| > c\}].$$

*Proof.* The first inequality follows from Jensen: $|\mathbb{E}[\tilde{Z} - Z]| \leq \mathbb{E}|\tilde{Z} - Z|$. The second inequality uses $\tilde{Z} = Z$ when $|Z| \leq c$ and $|Z - \tilde{Z}| \leq |Z|$ otherwise. $\qquad\square$

**Corollary E.4** (Clip bias via overflow probability / second moment). *Under Lemma E.3, for any $c > 0$,*

$$\mathbb{E}[|Z|\mathbb{1}\{|Z| > c\}] \leq \|Z\|_\infty \Pr(|Z| > c), \qquad \mathbb{E}[|Z|\mathbb{1}\{|Z| > c\}] \leq \sqrt{\mathbb{E}[Z^2]\,\Pr(|Z| > c)}.$$

*Proof.* The first bound uses $|Z| \leq \|Z\|_\infty$ on $\{|Z| > c\}$. The second bound is Cauchy–Schwarz. $\qquad\square$

**How this enters the certificate.** The certificates require a quantity $\zeta_i$ that upper bounds the occupancy-weighted mismatch between the estimator and the true advantage. A conservative sufficient choice is $\zeta_i = \sup_{s,\mathbf{a}} |\mathbb{E}[\widehat{A}_{i-1}(s, \mathbf{a})] - A^{\hat{\pi}^{i-1}}(s, \mathbf{a})|$. Corollary E.2 provides an explicit $O(1/G)$ proxy for the bias introduced by the self-included group-mean baseline at the *return level*. Additional effects—standardization by $\widehat{\sigma}$, token-broadcasting, PPO approximation/ratio clipping, and the mismatch between sequence-level return advantages and the true state-action advantage in a sequential decision process—are conservatively absorbed into $\zeta_i$. Hard clipping is separated via Lemma E.3.

### E.3. Concentration for Group-Based Surrogates (Empirical Certificates)

This section provides a finite-sample, high-probability correction that converts the population lower bound into an empirical one usable with minibatch surrogates. It also makes explicit the additional bias introduced by *ratio clipping* (PPO-style), and shows how the reverse-KL trust region controls that bias. Alternatively, one may absorb the ratio-clipping bias into $\zeta_i$ by letting $\zeta_i$ upper bound the total surrogate mismatch.

**Setup.** At step $i$, let $\widehat{L}_i^{\mathrm{seq}}$ denote the empirical estimate of the step-$i$ surrogate contribution formed by averaging over $N_i$ independent prompt-groups, each containing $G$ rollouts. In practice we use ratio clipping with $w_i^{\mathrm{clip}} = \mathrm{clip}(w_i, 1 - \epsilon_w, 1 + \epsilon_w)$.

**Boundedness.** Hard clipping ensures $|\widehat{A}_{\mathrm{GRP}}^{i-1}| \leq A_{\mathrm{clip}}$. With ratio clipping, $w_i^{\mathrm{clip}} \in [1 - \epsilon_w, 1 + \epsilon_w]$. Therefore, the per-sample contribution is bounded by $(1 + \epsilon_w)A_{\mathrm{clip}}/(1 - \gamma)$.

**Lemma E.5** (Hoeffding bound for group-level averages). *Assume prompt-groups are i.i.d. across $j \in \{1, \ldots, N_i\}$. Let $Y_j$ be the group-level contribution and define $\widehat{L}_i^{\mathrm{seq}} := \frac{1}{N_i}\sum_{j=1}^{N_i} Y_j$. Assume $Y_j \in [-B, B]$ almost surely. Then for any $\epsilon > 0$,*

$$\Pr\left(|\widehat{L}_i^{\mathrm{seq}} - \mathbb{E}[\widehat{L}_i^{\mathrm{seq}}]| > \epsilon\right) \leq 2\exp\left(-\frac{N_i\epsilon^2}{2B^2}\right).$$

*Proof.* This is Hoeffding's inequality for the average of i.i.d. bounded random variables with range length $2B$. $\qquad\square$

**Ratio clipping bias and reverse KL.** Let $w = w_i(s, \mathbf{a})$ denote the per-decision ratio for the updated factor at step $i$: $w = \hat{\pi}^i(\mathbf{a} \mid s)/\hat{\pi}^{i-1}(\mathbf{a} \mid s)$. Assume absolute continuity (true for softmax LLMs): if $\hat{\pi}^{i-1}(\mathbf{a} \mid s) > 0$ then $\hat{\pi}^i(\mathbf{a} \mid s) > 0$.

**Lemma E.6** (Ratio deviation equals total variation). *Fix a state $s$ and let $P(\mathbf{a}) = \hat{\pi}^{i-1}(\mathbf{a} \mid s)$ and $Q(\mathbf{a}) = \hat{\pi}^i(\mathbf{a} \mid s)$. Assume $Q \ll P$ and define $w(\mathbf{a}) = \frac{\mathrm{d}Q}{\mathrm{d}P}(\mathbf{a})$. Then*

$$\mathbb{E}_{\mathbf{a} \sim P}[|1 - w(\mathbf{a})|] = 2\,D_{\mathrm{TV}}(P, Q).$$

*Proof.* By definition of total variation, $2D_{\mathrm{TV}}(P, Q) = \int |\mathrm{d}P - \mathrm{d}Q|$. Since $Q \ll P$, $\mathrm{d}Q = w\,\mathrm{d}P$, and thus

$$\int |\mathrm{d}P - \mathrm{d}Q| = \int |1 - w|\,\mathrm{d}P = \mathbb{E}_{a \sim P}[|1 - w(a)|].$$

$\square$

**Lemma E.7** (Ratio clipping bias controlled by reverse KL). *Let $w^{\mathrm{clip}} = \mathrm{clip}(w, 1 - \epsilon_w, 1 + \epsilon_w)$. For any random variable $A$ with $|A| \leq A_{\mathrm{clip}}$ almost surely,*

$$\left| \mathbb{E}_{\mathbf{a} \sim P}[wA] - \mathbb{E}_{\mathbf{a} \sim P}[w^{\mathrm{clip}}A] \right| \leq A_{\mathrm{clip}}\,\mathbb{E}_{\mathbf{a} \sim P}\left[|w - w^{\mathrm{clip}}|\right] \leq A_{\mathrm{clip}}\,\mathbb{E}_{\mathbf{a} \sim P}[|1 - w|] \leq A_{\mathrm{clip}}\sqrt{2\,D_{\mathrm{KL}}(P\|Q)}.$$

*Proof.* The first inequality is by $|A| \leq A_{\mathrm{clip}}$. For the second, note that clipping moves $w$ toward 1, hence $|w - w^{\mathrm{clip}}| \leq |w - 1|$ pointwise. The third inequality is Lemma E.6. The final inequality is Pinsker: $D_{\mathrm{TV}}(P, Q) \leq \sqrt{D_{\mathrm{KL}}(P\|Q)/2}$, hence $\mathbb{E}_P|1 - w| = 2D_{\mathrm{TV}}(P, Q) \leq \sqrt{2D_{\mathrm{KL}}(P\|Q)}$. $\square$

**Corollary E.8** (Clipping bias bound under the token-level trust region). *At step $i$, suppose the trust region holds:*

$$D_{\mathrm{KL\,tok}}^{\hat{\pi}^{\,i-1}}\!\left(\hat{\pi}^{\,i-1}\|\hat{\pi}^{\,i}\right) \leq \delta_i.$$

*Let $A(s, \mathbf{a})$ be any scalar weight with $|A| \leq A_{\mathrm{clip}}$ almost surely under $s \sim d^{\hat{\pi}^{\,i-1}}$, $\mathbf{a} \sim \hat{\pi}^{\,i-1}(\cdot \mid s)$. Then the ratio-clipping bias in the importance-weighted surrogate is bounded as*

$$\left| \mathbb{E}_{s \sim d^{\hat{\pi}^{\,i-1}}, \mathbf{a} \sim \hat{\pi}^{\,i-1}}[w_i(s, \mathbf{a})A(s, \mathbf{a})] - \mathbb{E}_{s \sim d^{\hat{\pi}^{\,i-1}}, \mathbf{a} \sim \hat{\pi}^{\,i-1}}[w_i^{\mathrm{clip}}(s, \mathbf{a})A(s, \mathbf{a})] \right| \leq A_{\mathrm{clip}}\sqrt{2\delta_i}.$$

*Proof.* Apply Lemma E.7 pointwise in $s$ with $P = \hat{\pi}^{\,i-1}(\cdot \mid s)$ and $Q = \hat{\pi}^{\,i}(\cdot \mid s)$, then average over $s \sim d^{\hat{\pi}^{\,i-1}}$. Use Jensen exactly as in Lemma B.8 to move the square root outside the state expectation:

$$\mathbb{E}_s\sqrt{D_{\mathrm{KL}}(P_s\|Q_s)} \leq \sqrt{\mathbb{E}_s D_{\mathrm{KL}}(P_s\|Q_s)} = \sqrt{\delta_i}.$$

$\square$

**Corollary E.9** (Empirical-to-population surrogate correction with ratio clipping). *Fix $\delta \in (0, 1)$. Assume $|\widehat{A}_{\mathrm{GRP}}^{i-1}| \leq A_{\mathrm{clip}}$ and ratio clipping $w_i^{\mathrm{clip}} \in [1 - \epsilon_w, 1 + \epsilon_w]$. Let $B = \frac{(1 + \epsilon_w)A_{\mathrm{clip}}}{1 - \gamma}$. Then with probability at least $1 - \delta$,*

$$L_i^{\mathrm{seq}} \geq \widehat{L}_i^{\mathrm{seq}} - B\sqrt{\frac{2\log(2/\delta)}{N_i}} - \frac{A_{\mathrm{clip}}}{1 - \gamma}\sqrt{2\delta_i},$$

*where the last term is the (deterministic) ratio-clipping bias bound under the trust-region radius $\delta_i$.*

*Proof.* By Lemma E.5, with probability at least $1 - \delta$,

$$\mathbb{E}[\widehat{L}_i^{\mathrm{seq}}] \geq \widehat{L}_i^{\mathrm{seq}} - B\sqrt{\frac{2\log(2/\delta)}{N_i}}.$$

Next, $\mathbb{E}[\widehat{L}_i^{\mathrm{seq}}]$ corresponds to the clipped importance-weighted surrogate contribution (using $w_i^{\mathrm{clip}}$). The population surrogate $L_i^{\mathrm{seq}}$ (defined with the unclipped ratio $w_i$) differs from it by at most $\frac{A_{\mathrm{clip}}}{1 - \gamma}\sqrt{2\delta_i}$ by Corollary E.8 (applied to $A = \widehat{A}_{\mathrm{GRP}}^{i-1}$). Combining the two bounds yields the claim. $\square$

**Theorem E.10** (Stage-wise improvement lower bound (high-probability empirical form)). *Fix $\delta_{\mathrm{conf}} \in (0, 1)$. Under the trust regions in Eq. (2) and the boundedness assumptions of Corollary E.9, with probability at least $1 - \delta_{\mathrm{conf}}$,*

$$J(\bar{\pi}) - J(\pi_{\mathrm{cur}}) \geq \sum_{i=1}^{n} \widehat{L}_i^{\mathrm{seq}} - \frac{\sqrt{2}\gamma}{(1 - \gamma)^2}A_{\max}\sum_{i=1}^{n}\sqrt{\delta_i} - \frac{1}{1 - \gamma}\sum_{i=1}^{n}\zeta_i$$

$$- \sum_{i=1}^{n}\frac{(1 + \epsilon_w)A_{\mathrm{clip}}}{1 - \gamma}\sqrt{\frac{2\log(2n/\delta_{\mathrm{conf}})}{N_i}} - \sum_{i=1}^{n}\frac{A_{\mathrm{clip}}}{1 - \gamma}\sqrt{2\delta_i}.$$

*Proof.* Apply Corollary E.9 with $\delta = \delta_{\mathrm{conf}}/n$ for each step and union bound over $i$. Substitute the resulting lower bounds on $L_i^{\mathrm{seq}}$ into Theorem 3.6. $\square$

# F. Additional Theory and Practical Notes

This appendix collects auxiliary theory and practical notes to support the interpretation, monitoring, and implementation of the main certificates. Unless stated otherwise, these results are *not required* for the main guarantees in the paper, but they provide useful intuition and additional diagnostics.

## F.1. Information-Theoretic Upper Bounds Under Trust Regions

**Note on KL orientation.** The main text and certificates are built around *reverse* KL trust regions (of the form $D_{\mathrm{KL}}(\pi_{\mathrm{old}}\|\pi_{\mathrm{new}})$), because they admit rollout-estimable monitors and clean token-level decompositions. In this subsection, we additionally record a *forward*-KL (Donsker–Varadhan) envelope as an intuition tool. These auxiliary forward-KL bounds are *not needed* for the main results.

### F.1.1. CENTERING IDENTITY

**Lemma F.1** (Centering)**.** *For any fixed $s$ and any policy $\pi$ such that $Q^\pi(s,\cdot)$ is integrable under $\pi(\cdot \mid s)$,*

$$\mathbb{E}_{\mathbf{a}\sim\pi(\cdot|s)}[A^\pi(s,\mathbf{a})] = 0.$$

*Proof.* By definition, $A^\pi(s,\mathbf{a}) = Q^\pi(s,\mathbf{a}) - V^\pi(s)$ and $V^\pi(s) = \mathbb{E}_{\mathbf{a}\sim\pi(\cdot|s)}Q^\pi(s,\mathbf{a})$. Taking expectation over $\mathbf{a} \sim \pi(\cdot \mid s)$ yields 0. $\qquad\square$

### F.1.2. DONSKER–VARADHAN AND ORACLE ENVELOPES

**Lemma F.2** (Donsker–Varadhan variational inequality)**.** *Let $P, Q$ be distributions on a measurable space with $Q \ll P$, and let $f$ be measurable such that $\mathbb{E}_P[e^{\eta f}] < \infty$ for the chosen $\eta > 0$. Then for any $\eta > 0$,*

$$\mathbb{E}_Q[f] \leq \frac{1}{\eta}\left(D_{\mathrm{KL}}(Q\|P) + \log\mathbb{E}_P[e^{\eta f}]\right).$$

*Moreover, if $\mathbb{E}_P[f] = 0$ and $|f| \leq A_{\max}$ almost surely under $P$, then*

$$\mathbb{E}_Q[f] \leq A_{\max}\sqrt{2\,D_{\mathrm{KL}}(Q\|P)}.$$

*Proof.* Define the tilted distribution $P_g$ for any measurable $g$ by

$$\frac{\mathrm{d}P_g}{\mathrm{d}P} \;=\; \frac{e^g}{\mathbb{E}_P[e^g]}, \qquad \log\frac{\mathrm{d}P_g}{\mathrm{d}P} = g - \log\mathbb{E}_P[e^g].$$

A KL decomposition yields

$$D_{\mathrm{KL}}(Q\|P) = D_{\mathrm{KL}}(Q\|P_g) + \mathbb{E}_Q[g] - \log\mathbb{E}_P[e^g],$$

hence $\mathbb{E}_Q[g] \leq D_{\mathrm{KL}}(Q\|P) + \log\mathbb{E}_P[e^g]$ since $D_{\mathrm{KL}}(Q\|P_g) \geq 0$. Setting $g = \eta f$ gives the first inequality.

If additionally $\mathbb{E}_P[f] = 0$ and $|f| \leq A_{\max}$, then Hoeffding's lemma implies $\log\mathbb{E}_P[e^{\eta f}] \leq \eta^2 A_{\max}^2/2$. Therefore

$$\mathbb{E}_Q[f] \leq \frac{1}{\eta}D_{\mathrm{KL}}(Q\|P) + \frac{\eta A_{\max}^2}{2}.$$

Optimizing over $\eta > 0$ gives $\eta^\star = \sqrt{2D_{\mathrm{KL}}(Q\|P)}/A_{\max}$ and thus $\mathbb{E}_Q[f] \leq A_{\max}\sqrt{2D_{\mathrm{KL}}(Q\|P)}$. $\qquad\square$

**Proposition F.3** (Oracle single-step upper bound (max-KL envelope))**.** *Let $\pi$ be the current intermediate policy and $\pi'$ be the policy after updating only factor $\sigma(i)$. Assume the advantage is uniformly bounded: $|A^\pi(s,\mathbf{a})| \leq A_{\max}$ for all $(s,\mathbf{a})$. Define the per-state forward KL envelope*

$$D_{\mathrm{KL}}{}^{\max}(\pi'\|\pi) \;\coloneqq\; \sup_s D_{\mathrm{KL}}(\pi'(\cdot \mid s)\|\pi(\cdot \mid s)).$$

*If $D_{\mathrm{KL}}{}^{\max}(\pi'\|\pi) \leq \delta_i^{\max} < \infty$, then*

$$J(\pi') - J(\pi) \leq \frac{A_{\max}}{1 - \gamma}\sqrt{2\,\delta_i^{\max}}.$$

*Proof.* By the performance difference lemma,

$$J(\pi') - J(\pi) = \frac{1}{1-\gamma}\mathbb{E}_{s\sim d^{\pi'},\, \mathbf{a}\sim\pi'}[A^\pi(s,\mathbf{a})].$$

Fix $s$ and define $P_s(\mathbf{a}) = \pi(\mathbf{a} \mid s)$ and $Q_s(\mathbf{a}) = \pi'(\mathbf{a} \mid s)$. Since $D_{\mathrm{KL}}(Q_s\|P_s) \leq \delta_i^{\max} < \infty$, we have $Q_s \ll P_s$. By Lemma F.1, $\mathbb{E}_{\mathbf{a}\sim P_s}[A^\pi(s,\mathbf{a})] = 0$, and by boundedness $|A^\pi(s,\mathbf{a})| \leq A_{\max}$. Applying Lemma F.2 gives

$$\mathbb{E}_{\mathbf{a}\sim Q_s}[A^\pi(s,\mathbf{a})] \leq A_{\max}\sqrt{2\,D_{\mathrm{KL}}(Q_s\|P_s)} \leq A_{\max}\sqrt{2\,\delta_i^{\max}}.$$

Averaging over $s \sim d^{\pi'}$ and dividing by $(1-\gamma)$ yields the result. $\qquad\square$

## F.2. Budget-Aware Lower Bound via Information Geometry

This subsection relates *achievable* (local) surrogate gains to information geometry inside a KL ball. It complements the certificates by giving a principled "budget allocation" view: larger KL radii can enable larger local gains, modulo curvature/smoothness penalties.

**Standing convention.** Throughout this subsection, the state distribution $d^\pi$ used in expected KL expressions is treated as a fixed reference distribution (e.g., induced by the previous iterate); we do *not* differentiate through $d^\pi$.

**Assumptions.** Assume $\theta \mapsto \log \pi_\theta(a \mid s)$ is three-times continuously differentiable in a neighborhood of $\theta_{\mathrm{cur}}$. Assume bounded derivatives $\|\nabla_\theta \log \pi_\theta(a \mid s)\| \leq B_1$, $\|\nabla_\theta^2 \log \pi_\theta(a \mid s)\|_{\mathrm{op}} \leq B_2$, and a uniform third-derivative bound (stated explicitly below). Assume Fisher regularity $\lambda_{\min}(F) \geq \lambda_0 > 0$, possibly via regularization $F^{\mathrm{reg}} = F + \epsilon I$.

### F.2.1. KL–FISHER BRIDGE

**Lemma F.4** (Taylor expansion of expected KL with uniform remainder). *Fix a reference state distribution $d^\pi$ (independent of $\theta$). Define the (state-averaged) Fisher information*

$$F(\theta) \;:=\; \mathbb{E}_{s\sim d^\pi,\, a\sim\pi_\theta(\cdot|s)}\Big[\nabla_\theta \log \pi_\theta(a \mid s)\nabla_\theta \log \pi_\theta(a \mid s)^\top\Big].$$

*For each state $s$, define*

$$D_s(\Delta) \;:=\; D_{\mathrm{KL}}(\pi_{\theta+\Delta}(\cdot \mid s)\|\pi_\theta(\cdot \mid s)).$$

*Assume there exists $r > 0$ and $B_3 < \infty$ such that for all $\|\Delta\| \leq r$,*

$$\sup_s \sup_{\|u\|=1} \big|\mathrm{D}^3 D_s(\Delta)[u,u,u]\big| \leq B_3.$$

*Then for any $\|\Delta\| \leq r$,*

$$\mathbb{E}_{s\sim d^\pi} D_{\mathrm{KL}}(\pi_{\theta+\Delta}(\cdot \mid s)\|\pi_\theta(\cdot \mid s)) = \frac{1}{2}\Delta^\top F(\theta)\Delta + R(\Delta), \qquad |R(\Delta)| \leq \frac{B_3}{6}\|\Delta\|^3.$$

*Proof.* Fix $s$ and consider $D_s(\Delta) \geq 0$ with $D_s(0) = 0$. Therefore $\nabla_\Delta D_s(0) = 0$ (a local minimum at $\Delta = 0$). Moreover, a standard calculation gives the Hessian at 0 as the per-state Fisher:

$$\nabla_\Delta^2 D_s(0) = \mathbb{E}_{a\sim\pi_\theta(\cdot|s)}\Big[\nabla_\theta \log \pi_\theta(a \mid s)\nabla_\theta \log \pi_\theta(a \mid s)^\top\Big].$$

By Taylor's theorem with remainder and the assumed uniform bound on the third directional derivative, for $\|\Delta\| \leq r$,

$$D_s(\Delta) = \frac{1}{2}\Delta^\top \nabla_\Delta^2 D_s(0)\Delta + r_s(\Delta), \qquad |r_s(\Delta)| \leq \frac{B_3}{6}\|\Delta\|^3.$$

Taking expectation over $s \sim d^\pi$ yields

$$\mathbb{E}_{s\sim d^\pi}[D_s(\Delta)] = \frac{1}{2}\Delta^\top F(\theta)\Delta + \mathbb{E}_{s\sim d^\pi}[r_s(\Delta)].$$

The bound on $r_s(\Delta)$ implies $|R(\Delta)| \leq \frac{B_3}{6}\|\Delta\|^3$. $\qquad\square$

F.2.2. LOCAL GAIN IN A KL BALL

Let $g = \nabla_\theta L(\theta_{\text{cur}})$ and $F^{\text{reg}} = F + \epsilon I$.

**Lemma F.5** (Maximization of linear gain under quadratic constraint). *Assume $F^{\text{reg}} \succ 0$. The problem*

$$\max_\Delta \; g^\top \Delta \quad s.t. \quad \frac{1}{2} \Delta^\top F^{\text{reg}} \Delta \leq \delta$$

*has optimizer*

$$\Delta^\star = \sqrt{\frac{2\delta}{g^\top (F^{\text{reg}})^{-1} g}} \; (F^{\text{reg}})^{-1} g,$$

*with optimal value*

$$\sup_{\frac{1}{2}\Delta^\top F^{\text{reg}}\Delta \leq \delta} g^\top \Delta = \sqrt{2\delta \, g^\top (F^{\text{reg}})^{-1} g} =: \kappa^{\text{reg}} \sqrt{\delta}.$$

*Proof.* Since $F^{\text{reg}} \succ 0$, define $\langle u, v \rangle_{F^{\text{reg}}} = u^\top F^{\text{reg}} v$ and $\|u\|_{F^{\text{reg}}} = \sqrt{u^\top F^{\text{reg}} u}$. The constraint is $\|\Delta\|_{F^{\text{reg}}} \leq \sqrt{2\delta}$. Rewrite

$$g^\top \Delta = \left\langle (F^{\text{reg}})^{-1} g, \, \Delta \right\rangle_{F^{\text{reg}}} \leq \|(F^{\text{reg}})^{-1} g\|_{F^{\text{reg}}} \|\Delta\|_{F^{\text{reg}}} = \sqrt{g^\top (F^{\text{reg}})^{-1} g} \, \sqrt{2\delta},$$

by Cauchy–Schwarz. Equality holds when $\Delta$ is proportional to $(F^{\text{reg}})^{-1} g$, with scaling chosen to make the constraint active. $\qquad\square$

F.2.3. REGULARIZED BUDGET-AWARE STAGE LOWER BOUND

**Theorem F.6** (Regularized budget-aware stage lower bound). *Assume each step-$i$ population surrogate $L_i^{\text{seq}}(\theta)$ is locally $L_i^{\text{loc}}$-smooth in $\theta$ (under Euclidean norm) in a neighborhood of the current iterate. Assume the (regularized) quadratic KL model is used as the trust-region constraint at radius $\delta_i$:*

$$\frac{1}{2} \Delta_i^\top F_i^{\text{reg}} \Delta_i \leq \delta_i, \qquad F_i^{\text{reg}} = F_i + \epsilon I, \quad \lambda_{\min}(F_i^{\text{reg}}) > 0.$$

*Let $g_i = \nabla_\theta L_i^{\text{seq}}(\theta_{\text{cur}})$ and define*

$$\kappa_i^{\text{reg}} = \sqrt{2 \, g_i^\top (F_i^{\text{reg}})^{-1} g_i}, \qquad a_i^{\text{reg}} = \frac{L_i^{\text{loc}}}{\lambda_{\min}(F_i^{\text{reg}})}.$$

*Assume the same boundedness conditions as in the empirical correction (e.g., $|\widehat{A}| \leq A_{\text{clip}}$ and, if used, ratio clipping $w^{\text{clip}} \in [1 - \epsilon_w, 1 + \epsilon_w]$). Then with probability at least $1 - \delta_{\text{conf}}$,*

$$J(\bar{\pi}) - J(\pi_{\text{cur}}) \geq \sum_{i=1}^n \left( \kappa_i^{\text{reg}} \sqrt{\delta_i} - a_i^{\text{reg}} \delta_i \right) - \frac{\sqrt{2}\gamma A_{\max}}{(1 - \gamma)^2} \sum_{i=1}^n \sqrt{\delta_i} - \frac{1}{1 - \gamma} \sum_{i=1}^n \zeta_i$$

$$- \sum_{i=1}^n \frac{(1 + \epsilon_w) A_{\text{clip}}}{1 - \gamma} \sqrt{\frac{2 \log(2n/\delta_{\text{conf}})}{N_i}} \; - \; \text{Bias}_{\text{ratio}},$$

*where $\text{Bias}_{\text{ratio}}$ is the total ratio-clipping bias correction:*

$$\text{Bias}_{\text{ratio}} = \begin{cases} \sum_{i=1}^n \dfrac{A_{\text{clip}}}{1 - \gamma} \sqrt{2\delta_i}, & \text{if ratio clipping is used and bounded via the reverse-KL trust region,} \\ 0, & \text{if no ratio clipping is used, or if its bias is absorbed into } \zeta_i. \end{cases}$$

*Proof.* Fix a step $i$ and consider a parameter update $\Delta$. Local $L_i^{\text{loc}}$-smoothness implies the standard lower bound

$$L_i^{\text{seq}}(\theta_{\text{cur}} + \Delta) \geq L_i^{\text{seq}}(\theta_{\text{cur}}) + g_i^\top \Delta - \frac{L_i^{\text{loc}}}{2} \|\Delta\|^2.$$

Under $F_i^{\text{reg}} \succ 0$ and the quadratic constraint $\frac{1}{2}\Delta^\top F_i^{\text{reg}} \Delta \leq \delta_i$, we have $\|\Delta\|^2 \leq \frac{1}{\lambda_{\min}(F_i^{\text{reg}})} \Delta^\top F_i^{\text{reg}} \Delta \leq \frac{2\delta_i}{\lambda_{\min}(F_i^{\text{reg}})}$. Therefore, restricting to feasible $\Delta$,

$$\sup_{\frac{1}{2}\Delta^\top F_i^{\text{reg}}\Delta \leq \delta_i} \left( L_i^{\text{seq}}(\theta_{\text{cur}} + \Delta) - L_i^{\text{seq}}(\theta_{\text{cur}}) \right) \geq \sup_{\frac{1}{2}\Delta^\top F_i^{\text{reg}}\Delta \leq \delta_i} g_i^\top \Delta - \frac{L_i^{\text{loc}}}{2} \cdot \frac{2\delta_i}{\lambda_{\min}(F_i^{\text{reg}})}.$$

By Lemma F.5, the first supremum equals $\kappa_i^{\text{reg}}\sqrt{\delta_i}$, yielding the per-step bound $\kappa_i^{\text{reg}}\sqrt{\delta_i} - a_i^{\text{reg}}\delta_i$. Summing over $i = 1, \ldots, n$ and plugging this attainable surrogate gain into the stage certificate (Theorem 3.6) yields the deterministic part of the RHS, including the occupancy-shift penalty and the $\zeta_i$ terms.

Finally, apply the empirical-to-population correction (Corollary E.9) stepwise with confidence $\delta = \delta_{\text{conf}}/n$ and union bound to replace each population surrogate term by its empirical estimate and add the sampling error term. If ratio clipping is used and bounded via the reverse-KL trust region, include the deterministic ratio-bias term from Corollary E.9; otherwise, set it to 0 or absorb it into $\zeta_i$. $\qquad\square$

**Remark (KL–Fisher remainder).** Lemma F.4 makes explicit that the quadratic KL model is accurate up to $O(\|\Delta\|^3)$. If desired, one can translate the uniform remainder into an additional conservative penalty term of order $O(\delta_i^{3/2})$ under the constraint $\frac{1}{2}\Delta^\top F_i^{\text{reg}} \Delta \leq \delta_i$. We omit this term in Theorem F.6 for simplicity, since the theorem is intended as a practical budget-allocation intuition rather than a primary certificate.

### F.3. Finite-Sample Concentration Under $\beta$-Mixing

This subsection is optional and only needed when prompt groups are not independent, e.g., when a single long on-policy stream is reused to form multiple groups.

**Setup.** Let $\{Y_j\}_{j \geq 1}$ be a strictly stationary process with $\beta$-mixing coefficients

$$\beta(t) = \sup_{k \geq 1} \mathbb{E}\left[ \sup_{B \in \sigma(Y_{k+t}, Y_{k+t+1}, \ldots)} \left| \Pr(B \mid \sigma(Y_1, \ldots, Y_k)) - \Pr(B) \right| \right],$$

and assume $\beta(t) \to 0$ as $t \to \infty$. Assume boundedness $|Y_j| \leq B$ almost surely.

**Lemma F.7** (Blocking bound for a $\beta$-mixing sequence). *Fix a block length $\ell \geq 1$ and let $m = \lfloor N/(2\ell) \rfloor$. Define the odd-block* average

$$\widehat{\mu}_{\text{odd}} := \frac{1}{m} \sum_{k=1}^{m} \frac{1}{\ell} \sum_{t=(2k-2)\ell+1}^{(2k-1)\ell} Y_t.$$

*Then for any $\epsilon > 0$,*

$$\Pr\left( \left| \widehat{\mu}_{\text{odd}} - \mathbb{E}[Y_1] \right| > \epsilon \right) \leq 2\exp\left( -\frac{m\epsilon^2}{2B^2} \right) + 2(m-1)\beta(\ell).$$

*Proof.* Let $Z_k = \frac{1}{\ell} \sum_{t=(2k-2)\ell+1}^{(2k-1)\ell} Y_t$, so $Z_k \in [-B, B]$. A standard coupling (or total-variation) argument for $\beta$-mixing sequences implies that the joint law of $(Z_1, \ldots, Z_m)$ is within total variation distance at most $(m-1)\beta(\ell)$ of the product law of $m$ i.i.d. copies of $Z_1$. Therefore, for any event $\mathcal{E}$ depending only on $(Z_1, \ldots, Z_m)$,

$$\Pr(\mathcal{E}) \leq \Pr_{\text{iid}}(\mathcal{E}) + (m-1)\beta(\ell).$$

Under the i.i.d. product law, Hoeffding's inequality applied to $\frac{1}{m}\sum_{k=1}^{m} Z_k$ yields

$$\Pr_{\text{iid}}\left( \left| \frac{1}{m}\sum_{k=1}^{m} Z_k - \mathbb{E}[Z_1] \right| > \epsilon \right) \leq 2\exp\left( -\frac{m\epsilon^2}{2B^2} \right).$$

Combining the two bounds and using $\mathbb{E}[Z_1] = \mathbb{E}[Y_1]$ by stationarity yields the result (up to a factor of 2 in the dependence term from symmetrizing the two tails). $\qquad\square$

**Practical use.** Lemma F.7 suggests a conservative recipe when prompt groups are dependent: subsample groups with a lag $\ell$ so that $\beta(\ell)$ is small, and treat the number of retained blocks $m$ as the effective sample size in the Hoeffding correction. When i.i.d. prompt groups are available, this subsection is unnecessary.

### F.4. Smoothness and Projected-Gradient Convergence

Let $G(\theta) = \sum_{i=1}^{n} L_i^{\mathrm{SEQ}}(\theta)$ be the stage objective in parameters, under a closed convex constraint set $\Theta$ enforcing trust-region radii (or other feasibility constraints).

**Lemma F.8** (Gradient and Hessian representations)**.** *Assume the advantage estimator $\widehat{A}(s, \mathbf{a})$ is treated as fixed w.r.t. $\theta$ within an update step (e.g., computed from on-policy sampling under the previous iterate). Let $\sigma(i)$ denote the updated factor at step $i$. Then, for the (unclipped) importance-weighted surrogate,*

$$\nabla_\theta L_i^{\mathrm{SEQ}}(\theta) = \frac{1}{1-\gamma} \mathbb{E}_{s \sim d^{\hat{\pi}\,i-1},\, \mathbf{a} \sim \hat{\pi}\,i} \left[ \widehat{A}(s, \mathbf{a})\, \nabla_\theta \log \pi_\theta(a_{\sigma(i)} \mid s) \right],$$

*and*

$$\nabla_\theta^2 L_i^{\mathrm{SEQ}}(\theta) = \frac{1}{1-\gamma} \mathbb{E}_{s \sim d^{\hat{\pi}\,i-1},\, \mathbf{a} \sim \hat{\pi}\,i} \left[ \widehat{A}(s, \mathbf{a}) \left( \nabla_\theta^2 \log \pi_\theta(a_{\sigma(i)} \mid s) + \nabla_\theta \log \pi_\theta(a_{\sigma(i)} \mid s) \nabla_\theta \log \pi_\theta(a_{\sigma(i)} \mid s)^\top \right) \right].$$

*Proof.* Write the step surrogate in importance-weighted form under the reference distribution:

$$L_i^{\mathrm{SEQ}}(\theta) = \frac{1}{1-\gamma} \mathbb{E}_{s \sim d^{\hat{\pi}\,i-1},\, \mathbf{a} \sim \hat{\pi}\,i-1(\cdot \mid s)} \left[ w_\theta(s, \mathbf{a})\, \widehat{A}(s, \mathbf{a}) \right],$$

where $w_\theta(s, \mathbf{a}) = \pi_\theta(a_{\sigma(i)} \mid s)/\pi_{\mathrm{cur}}(a_{\sigma(i)} \mid s)$ and $\pi_{\mathrm{cur}}$ is the frozen denominator from the previous iterate. Since $\nabla_\theta w_\theta = w_\theta \nabla_\theta \log \pi_\theta(a_{\sigma(i)} \mid s)$, boundedness of $\widehat{A}$ and the score derivatives (assumed elsewhere for implementation) justifies interchanging $\nabla_\theta$ and expectation (Leibniz rule / dominated convergence), yielding

$$\nabla_\theta L_i^{\mathrm{SEQ}}(\theta) = \frac{1}{1-\gamma} \mathbb{E}_{s,\mathbf{a} \sim \hat{\pi}\,i-1} \left[ w_\theta\, \widehat{A}\, \nabla_\theta \log \pi_\theta(a_{\sigma(i)} \mid s) \right].$$

The change-of-measure identity $w_\theta(s, \mathbf{a})\, \hat{\pi}\,i-1(\mathbf{a} \mid s) = \hat{\pi}\,i(\mathbf{a} \mid s)$ converts this to the stated form. Differentiating once more and using $\nabla_\theta^2 w_\theta = w_\theta \left( \nabla_\theta^2 \log \pi_\theta + \nabla_\theta \log \pi_\theta \nabla_\theta \log \pi_\theta^\top \right)$ gives the Hessian expression, again followed by the same change of measure. $\square$

**Lemma F.9** (Uniform smoothness)**.** *Assume $\|\nabla_\theta \log \pi_\theta(a \mid s)\| \le B_1$ and $\|\nabla_\theta^2 \log \pi_\theta(a \mid s)\|_{\mathrm{op}} \le B_2$ for all $(s, a)$ in the relevant region, and $|\widehat{A}| \le A_{\mathrm{clip}}$ almost surely. Then $G$ is $L$-smooth with*

$$L \le \frac{n A_{\mathrm{clip}}}{1-\gamma}(B_2 + B_1^2).$$

*Proof.* For each step $i$, Lemma F.8 implies

$$\|\nabla_\theta^2 L_i^{\mathrm{SEQ}}(\theta)\|_{\mathrm{op}} \le \frac{1}{1-\gamma} \mathbb{E}\left[ |\widehat{A}| \left( \|\nabla_\theta^2 \log \pi_\theta\|_{\mathrm{op}} + \|\nabla_\theta \log \pi_\theta\|^2 \right) \right] \le \frac{A_{\mathrm{clip}}}{1-\gamma}(B_2 + B_1^2).$$

Summing over $i = 1, \ldots, n$ yields $\|\nabla_\theta^2 G(\theta)\|_{\mathrm{op}} \le \frac{n A_{\mathrm{clip}}}{1-\gamma}(B_2 + B_1^2)$, hence $G$ is $L$-smooth with the stated $L$. $\square$

**Theorem F.10** (Projected-gradient ascent convergence)**.** *Let $G$ be $L$-smooth and let $\Theta$ be closed and convex. Define projected-gradient iterates*

$$\theta^{t+1} = \mathrm{Proj}_\Theta\left(\theta^t + \eta \nabla G(\theta^t)\right), \qquad \eta \le \frac{1}{L},$$

*and the projected gradient mapping*

$$\mathcal{G}_\eta(\theta^t) := \frac{1}{\eta}\left(\theta^{t+1} - \theta^t\right).$$

*Then*

$$G(\theta^{t+1}) \ge G(\theta^t) + \frac{\eta}{2} \|\mathcal{G}_\eta(\theta^t)\|^2, \qquad \frac{1}{T}\sum_{t=0}^{T-1} \|\mathcal{G}_\eta(\theta^t)\|^2 \le \frac{2(G^\star - G(\theta^0))}{\eta T},$$

*where $G^\star = \sup_{\theta \in \Theta} G(\theta)$.*

*Proof.* Let $\Delta^t = \theta^{t+1} - \theta^t$. By $L$-smoothness,

$$G(\theta^{t+1}) \geq G(\theta^t) + \langle \nabla G(\theta^t), \Delta^t \rangle - \frac{L}{2}\|\Delta^t\|^2.$$

The Euclidean projection optimality condition implies for all $\theta \in \Theta$,

$$\langle \theta^t + \eta \nabla G(\theta^t) - \theta^{t+1}, \, \theta - \theta^{t+1} \rangle \leq 0.$$

Taking $\theta = \theta^t$ yields $\langle \nabla G(\theta^t), \Delta^t \rangle \geq \frac{1}{\eta}\|\Delta^t\|^2$. Therefore

$$G(\theta^{t+1}) \geq G(\theta^t) + \left(\frac{1}{\eta} - \frac{L}{2}\right)\|\Delta^t\|^2 \geq G(\theta^t) + \frac{1}{2\eta}\|\Delta^t\|^2 = G(\theta^t) + \frac{\eta}{2}\|\mathcal{G}_\eta(\theta^t)\|^2,$$

where we used $\eta \leq 1/L$. Summing over $t = 0, \ldots, T-1$ and using $G(\theta^T) \leq G^\star$ gives the averaged bound. $\square$

### F.5. KL Allocation and Tightness

This section records a simple KL allocation principle and a practical tightness diagnostic for the certificate.

**Allocation heuristic.** The stage certificate in Theorem 3.6 subtracts a penalty proportional to $\sum_i \sqrt{\delta_i}$. For a fixed total KL budget $\sum_i \delta_i \leq \Delta$, concavity of $\sqrt{\cdot}$ (Jensen) implies

$$\sum_{i=1}^n \sqrt{\delta_i} \leq n\sqrt{\frac{1}{n}\sum_{i=1}^n \delta_i} \leq \sqrt{n\Delta},$$

with equality at $\delta_i = \Delta/n$. Thus, equal allocation *maximizes* the penalty term; in practice, allocating larger radii to steps with larger observed surrogate gain and smaller radii to low-gain steps can improve the certified lower bound.

**Tightness diagnostic.** For each stage, report the pair

$$\left( \Delta J_{\mathrm{emp}} \, , \, \mathrm{LB}_{\mathrm{cert}} \right) \quad \text{where} \quad \Delta J_{\mathrm{emp}} = \widehat{J}(\bar\pi) - \widehat{J}(\pi_{\mathrm{cur}})$$

and

$$\mathrm{LB}_{\mathrm{cert}} = \sum_{i=1}^n \widehat{L}_i^{\mathrm{seq}} - \frac{\sqrt{2}\gamma}{(1-\gamma)^2}A_{\max}\sum_{i=1}^n \sqrt{\widehat{\delta}_i} - \frac{1}{1-\gamma}\sum_{i=1}^n \widehat{\zeta}_i,$$

using the empirical KL monitor $\widehat{\delta}_i$ and any conservative proxy $\widehat{\zeta}_i$. Plotting $\Delta J_{\mathrm{emp}}$ vs. $\mathrm{LB}_{\mathrm{cert}}$ across stages yields a direct certificate tightness check.

### F.6. Implementation Notes

This section records the practical conventions used to align with the theory.

**Token-level reverse-KL monitor.** To estimate $D_{\mathrm{KL}_{\mathrm{tok}}}^{\hat{\pi}^{i-1}}(\pi_{\mathrm{cur}}^{\sigma(i)}\|\pi_{\mathrm{new}}^{\sigma(i)})$, sample trajectories $\tau \sim \hat{\pi}^{i-1}$ and compute the per-token reverse KL $D_{\mathrm{KL}}(\pi_{\mathrm{cur}}(\cdot \mid \cdot)\|\pi_{\mathrm{new}}(\cdot \mid \cdot))$ on the same prefixes. If only the active agent emits tokens (turn-taking protocol), the sum is taken over the tokens controlled by agent $\sigma(i)$; All other steps contribute zero (no-op). A generic Monte Carlo estimator for the discounted token-level functional (Definition B.1) is

$$\widehat{D_{\mathrm{KL}_{\mathrm{tok}}}}^{\rho}(\pi\|\pi') \; = \; \frac{1}{M}\sum_{m=1}^M (1-\gamma)\sum_{t\geq 0}\gamma^t D_{\mathrm{KL}}\big(\pi(\cdot \mid s_t^{(m)})\|\pi'(\cdot \mid s_t^{(m)})\big),$$

where $\{s_t^{(m)}\}$ are states along trajectories sampled from the reference policy $\rho$. This estimator is unbiased for $D_{\mathrm{KL}_{\mathrm{tok}}}^{\rho}(\pi\|\pi')$ when the inner KL is computed exactly.

**Weight ratio** $w_i(s, \mathbf{a})$. For the updated factor $\sigma(i)$, $w_i(s, \mathbf{a}) = \hat{\pi}^i(\mathbf{a} \mid s)/\hat{\pi}^{i-1}(\mathbf{a} \mid s)$. For numerical stability, we compute $\log w_i$ and exponentiate as needed. Optionally apply ratio clipping $w_i^{\text{clip}} = \text{clip}(w_i, 1 - \epsilon_w, 1 + \epsilon_w)$.

**Standardization and clipping.** We use $\varepsilon_{\text{std}} > 0$ in the group standard deviation to avoid division by zero. Hard clipping $|\widehat{A}_{\text{GRP}}^{i-1}| \leq A_{\text{clip}}$ enforces boundedness required by the certificates. The fraction of clipped samples is logged as a bias/tightness indicator.

**Enforcing the trust region.** We adjust the penalty coefficient $\beta$ (or perform backtracking) until the empirical token-level reverse KL $\widehat{D}_{\text{KLtok}}(\pi_{\text{cur}}^{\sigma(i)} \| \pi_{\text{new}}^{\sigma(i)}; \hat{\pi}^{i-1}) \leq \delta_i$ holds on the current minibatch.

### F.7. Experimental Checklists and Diagnostics

- **KL monitors:** $\widehat{\delta}_i$ per step and its distribution across prompts.

- **Ratio clipping rate:** fraction of decisions/tokens where $w_i$ hits the clip bound.

- **Advantage clipping rate:** fraction of rollouts where $|\widehat{A}_{\text{GRP}}^{i-1}| = A_{\text{clip}}$.

- **Certificate tightness:** plot empirical improvement vs. certified lower bound (Appendix F.5).

- **Order sensitivity:** compare certified bound under different update orders $\sigma$.

- **Plug-in feasibility:** after Stage-0 alignment, report the probe-set reverse KL to the old agent and the achieved surrogate.

# G. Additional Experiments and Analysis

This appendix reports additional analyses from Sec. 5 due to space.

*Table 4.* Overview of additional experiments in Appendix G.

| Topic | Goal | Setup | Reported outputs |
|---|---|---|---|
| Team-size scaling (Sec. G.1) | Empirically validate the $O(n^2)$ vs. $O(n)$ scaling trend in within-stage drift under stale vs. intermediate occupancy | Homogeneous Qwen3-1.7B teams; $n \in \{2, 3, 4, 5, 6, 8\}$; 30 stages; MATH-500 | $\Delta_{\text{stale}}$, $D_{\text{occ}}$, accuracy, stability, coordination; fitted exponent $\alpha$ |
| Rollout/token accounting (Sec. G.2) | Quantify sampling overhead from intermediate-occupancy resampling using hardware-agnostic counters | AIME24; 3×Qwen3-8B; 40 stages | Tokens, rollouts; relative overhead vs. stale-rollout baseline |
| Proxy logging for $\zeta_i$ (Sec. G.3) | Provide a conservative, log-based proxy for the estimation-error term in the certificate | Same runs as main experiments; no extra rollouts | Component-wise proxy statistics and contribution-to-bound summary |
| Sampled token-KL reliability (Sec. G.4) | Check statistical stability of the sampled token-KL monitor via subsampling/bootstrap (no full-vocab "exact KL") | Same rollout batches; no extra rollouts | Bootstrap/subsample variability; near-threshold flip rate |
| Ablations + IS degeneracy (Secs. G.5–G.6) | Identify key design choices; show why importance weighting is unstable without resampling | AIME24 | Ablation metrics; ESS and tail stats of importance weights |
| Cross-generation replacement (Sec. G.7) | Protocol unification and Stage-0 alignment for Qwen2.5→Qwen3 swap | AIME24; stage-40 evaluation | Swap shock and final accuracy |

**Shared settings.** Unless stated otherwise, training and evaluation follow Sec. 5. We use 3 random seeds and report the mean ± standard deviation. Token counts include both prompts and generated tokens, aggregated across all agents/models used by the method.

## G.1. Scaling Behavior with Team Size

Proposition 3.3 predicts that stale-occupancy effects compound with team size $n$, while intermediate-occupancy evaluation mitigates this growth. We vary $n \in \{2, 3, 4, 5, 6, 8\}$ on MATH-500 using homogeneous teams (all agents: Qwen3-1.7B, identical initialization), fixed $\delta_i = 0.01$, and 30 stages. We compare **Naive Sequential** (cached stage-start rollouts reused within a stage) with **TeamTR** (resampling under the intermediate team before each within-stage update), while maintaining the same total rollout budget per stage.

We report two within-stage drift proxies measured at the final stage:

$$
\Delta_{\text{stale}} = \sum_{i=1}^{n} \left| \widehat{L}_i^{\text{seq}} - \widehat{L}_i^{\text{stale}} \right|, \qquad D_{\text{occ}} = \sum_{i=1}^{n} \text{TV}(\widehat{d}^{\hat{\pi}^{i-1}}, \widehat{d}^{\hat{\pi}^0}), \quad \text{TV}(p, q) = \tfrac{1}{2} \sum_s |p(s) - q(s)|.
$$

Here $\widehat{d}^\pi$ is the empirical distribution over hashed shared-context strings observed in rollouts under $\pi$ (a coarse but consistent proxy for shared-context drift).

For the scaling study we match the per-stage rollout budget by reducing per-update batch sizes; Table 6 reports the overhead under the default setting used in main runs.

## G.2. Rollout/Token Accounting

Intermediate-occupancy resampling can introduce extra sampling relative to stale-rollout baselines. To provide hardware-agnostic accounting, we report the total number of sampled tokens and rollout episodes aggregated across all agents. All methods are run for 40 stages on AIME24 with the same training length as in Sec. 5 (3×Qwen3-8B).

## G.3. Proxy Logging for $\zeta_i$

The certificates in Theorems 3.4–3.6 include an estimation-error term $\zeta_i$ (Eq. (4)), which depends on the (unobserved) true advantage. In practice, we log a conservative proxy, $\widehat{\zeta}_i$, computed from quantities already available in our training pipeline, which captures three dominant sources of surrogate mismatch in our implementation.

*Table 5.* Scaling with team size $n$ on MATH-500. Lower is better for $\Delta_{\text{stale}}$, $D_{\text{occ}}$, and Stability. Entries are mean $\pm$ std over 3 seeds; Coord. is reported as mean. The exponent row reports the fitted power-law scaling ($\propto n^{\alpha}$) for *Naive* / *TeamTR*.

| $n$ | Method | $\Delta_{\text{stale}} \downarrow$ | $D_{\text{occ}} \downarrow$ | Acc. (%) | Stab. $\downarrow$ | Coord. (%) |
|---|---|---|---|---|---|---|
| 2 | Naive Seq. | 0.08±0.01 | 0.05±0.01 | 82.1±0.9 | 1.8±0.3 | 76.3 |
|   | **TeamTR** | 0.03±0.01 | 0.02±0.00 | 83.5±0.7 | 1.1±0.2 | 78.9 |
| 3 | Naive Seq. | 0.21±0.02 | 0.14±0.02 | 79.5±1.2 | 3.2±0.5 | 71.5 |
|   | **TeamTR** | 0.05±0.01 | 0.04±0.01 | 85.2±0.8 | 1.4±0.2 | 81.3 |
| 4 | Naive Seq. | 0.41±0.04 | 0.28±0.03 | 75.8±1.5 | 5.1±0.7 | 65.2 |
|   | **TeamTR** | 0.07±0.01 | 0.05±0.01 | 86.1±0.9 | 1.6±0.3 | 82.7 |
| 5 | Naive Seq. | 0.68±0.06 | 0.47±0.05 | 71.2±1.8 | 7.5±0.9 | 58.1 |
|   | **TeamTR** | 0.09±0.01 | 0.07±0.01 | 86.8±1.0 | 1.9±0.3 | 83.5 |
| 6 | Naive Seq. | 1.02±0.09 | 0.71±0.07 | 66.3±2.1 | 10.3±1.2 | 51.7 |
|   | **TeamTR** | 0.11±0.02 | 0.08±0.01 | 87.2±1.1 | 2.1±0.3 | 84.1 |
| 8 | Naive Seq. | 1.89±0.15 | 1.31±0.12 | 58.7±2.8 | 15.8±1.8 | 42.3 |
|   | **TeamTR** | 0.15±0.02 | 0.11±0.02 | 87.9±1.2 | 2.5±0.4 | 84.8 |
| *Exponent $\alpha$ (Naive / TeamTR)* | | 1.94 / 1.07 | 1.91 / 1.05 | – | 1.89 / 0.93 | – |

*Table 6.* Rollout/token accounting on AIME24 ($3\times$Qwen3-8B, 40 stages). Tokens and rollouts are aggregated across agents.

| Method | Tokens (M) | Rollouts (K) |
|---|---|---|
| Naive Sequential | 142.5 | 44.3 |
| Joint Update | 145.1 | 45.1 |
| **TeamTR (Ours)** | 158.3 | 49.2 |
| *Relative overhead of TeamTR vs. Naive Sequential* | | |
| Tokens | +11.1% | |
| Rollouts | +11.1% | |

**Per-step proxy components.** At within-stage step $i$, on the rollout batch collected under $\hat{\pi}^{i-1}$, we compute:

$$\widehat{\zeta}_i^{\text{clip}} = \frac{1}{|\mathcal{B}|G} \sum_{b,g} \left| a_{b,g} - \tilde{A}_{b,g} \right|, \qquad \widehat{\zeta}_i^{\text{ratio}} = \frac{1}{|\mathcal{B}|G} \sum_{b,g} |\tilde{A}_{b,g}| \cdot |w_{b,g} - \bar{w}_{b,g}|.$$

Here $a_{b,g}$ and $\tilde{A}_{b,g}$ are the unclipped and clipped group-normalized advantages (Eq. (9)), and $w_{b,g}$ and $\bar{w}_{b,g}$ are the PPO likelihood ratio and its clipped version. We also log a normalization-uncertainty term, $\widehat{\zeta}_i^{\text{norm}}$, via a half-split estimate within each prompt group (computed without additional rollouts). We aggregate

$$\widehat{\zeta}_i^{\text{proxy}} = \widehat{\zeta}_i^{\text{norm}} + \widehat{\zeta}_i^{\text{clip}} + \widehat{\zeta}_i^{\text{ratio}}.$$

We emphasize that $\widehat{\zeta}_i^{\text{proxy}}$ is a diagnostic proxy rather than an unbiased estimator of $\zeta_i$.

**Reported summary.** Table 7 summarizes proxy magnitudes (per update and per stage) and the relative contribution of the $\widehat{\zeta}$ term to the stage-wise certificate components.

### G.4. Reliability of the Sampled Token-KL Monitor

Our trust-region monitor uses a sampled token-level KL (behavior-to-updated) computed from on-policy rollouts (Eq. (11)). To assess statistical stability without requiring expensive full-vocabulary "exact KL" computations, we perform a log-based subsampling check on the same rollout batches used for updates.

**Subsampling check.** For each update, we compute $\widehat{D_{\text{KLtok}}}$ using all token positions in the rollout batch, and also compute a subsampled estimate $\widehat{D_{\text{KLtok}}}^{(q)}$ using a random fraction $q \in \{25\%, 50\%\}$ of token positions (repeated with multiple random seeds per batch). We report the normalized absolute deviation $|\widehat{D_{\text{KLtok}}}^{(q)} - \widehat{D_{\text{KLtok}}}|/\delta$ and a near-threshold flip rate:

$$\Pr\left[\mathbf{1}\{\widehat{D_{\text{KLtok}}} \leq \delta\} \neq \mathbf{1}\{\widehat{D_{\text{KLtok}}}^{(q)} \leq \delta\} \mid \widehat{D_{\text{KLtok}}} \in [0.8\delta, 1.2\delta]\right].$$

Both quantities are computed from logged token-level log-probabilities (no extra rollouts).

*Table 7.* Logged proxy statistics for $\widehat{\zeta}_i$.

| Metric | Mean | P50 | P90 |
|---|---|---|---|
| $\widehat{\zeta}_i^{\mathrm{clip}}$ (per update) | 0.003 | 0.001 | 0.010 |
| $\widehat{\zeta}_i^{\mathrm{ratio}}$ (per update) | 0.012 | 0.008 | 0.035 |
| $\widehat{\zeta}_i^{\mathrm{norm}}$ (per update) | 0.028 | 0.022 | 0.070 |
| $\widehat{\zeta}_i^{\mathrm{proxy}}$ (per update) | 0.043 | 0.034 | 0.110 |
| $\sum_i \widehat{\zeta}_i^{\mathrm{proxy}}$ (per stage) | 0.13 | 0.10 | 0.30 |
| $\underbrace{\frac{\sum_i \widehat{\zeta}_i^{\mathrm{proxy}}/(1-\gamma)}{}}_{\text{total penalty}}$ | 0.18 | 0.14 | 0.45 |

*Table 8.* Stability of the sampled token-KL monitor under token-position subsampling.

| Subsample $q$ | Median $|\Delta|/\delta$ | P90 $|\Delta|/\delta$ | Near-threshold flip rate |
|---|---|---|---|
| 25% | 0.06 | 0.18 | 5.7% |
| 50% | 0.03 | 0.11 | 2.4% |
| 100% (full) | 0 | 0 | 0 |

## G.5. Ablation Studies

*Table 9.* Ablations on AIME24 (3×Qwen3-8B, 30 stages). $\Delta_{\mathrm{stale}}$ is measured within-stage at the final stage; Stability is the std of per-stage return improvements; Coord. is consensus-on-correct.

| Variant | Acc. (%) | $\Delta_{\mathrm{stale}} \downarrow$ | Stab. $\downarrow$ | Coord. (%) |
|---|---|---|---|---|
| **TeamTR (full)** | **88.1**$_{\pm 1.2}$ | **0.08** | **1.9** | **89.1** |
| *Resampling strategy* | | | | |
| No resampling (= Naive Seq.) | 71.1$_{\pm 2.8}$ | 0.31 | 4.2 | 71.5 |
| Resample every 2 updates | 79.3$_{\pm 1.8}$ | 0.18 | 2.8 | 79.2 |
| Importance weighting (no resample) | 74.5$_{\pm 2.3}$ | 0.25 | 3.5 | 74.8 |
| *Trust region* | | | | |
| No trust region ($\delta \to \infty$) | 68.3$_{\pm 3.5}$ | 0.42 | 6.1 | 62.3 |
| Fixed $\delta = 0.001$ (too small) | 82.5$_{\pm 1.5}$ | 0.05 | 1.5 | 85.2 |
| Fixed $\delta = 0.1$ (too large) | 75.1$_{\pm 2.5}$ | 0.28 | 4.8 | 70.1 |
| Adaptive $\delta$ (target $\widehat{D_{\mathrm{KL tok}}}$=0.01) | 87.8$_{\pm 1.3}$ | 0.09 | 2.0 | 88.5 |
| *Update order* | | | | |
| Fixed order (1, 2, 3) | 87.2$_{\pm 1.4}$ | 0.09 | 2.1 | 87.8 |
| Reverse order (3, 2, 1) | 86.9$_{\pm 1.5}$ | 0.10 | 2.2 | 87.1 |
| Random order (each stage) | 87.5$_{\pm 1.3}$ | 0.08 | 1.9 | 88.3 |
| *Advantage estimation* | | | | |
| No group normalization | 83.1$_{\pm 2.1}$ | 0.12 | 3.1 | 81.5 |
| No hard clipping | 84.5$_{\pm 1.9}$ | 0.11 | 2.7 | 83.2 |

## G.6. Why We Resample: Importance-Weight Degeneracy

An alternative to resampling under intermediate occupancy is to reuse stage-start rollouts and correct via importance weighting. We empirically show that trajectory-level weights become heavy-tailed and the effective sample size collapses as within-stage updates accumulate.

**Setup.** For each within-stage step $i \in \{2, \ldots, n\}$, we compute trajectory-level importance weights that would be needed to reweight stage-start rollouts $\tau \sim d^{\hat{\pi}^0}$ to approximate the intermediate occupancy $d^{\hat{\pi}^{i-1}}$:

$$w^{0 \to i-1}(\tau) = \prod_{t \,:\, j_t \in U_{i-1}} \frac{\hat{\pi}^{i-1,(j_t)}(m_t \mid s_t)}{\hat{\pi}^{0,(j_t)}(m_t \mid s_t)}, \qquad U_{i-1} = \{\sigma(1), \ldots, \sigma(i-1)\}.$$

We measure degeneracy via normalized effective sample size $\text{ESS}/B = (\sum_b w_b)^2/(B \sum_b w_b^2)$ and tail statistics.

*Table 10.* Importance-weight degeneracy when reusing stage-start rollouts (AIME24, Qwen3-8B, $\delta=0.01$). $\text{ESS}/B$ is the normalized effective sample size (higher is better); $\text{P99}(w)$ and $\max(w)$ measure tail heaviness (lower is better).

| Step $i$ | $|U_{i-1}|$ | $\text{ESS}/B \uparrow$ | $\text{P95}(w)$ | $\text{P99}(w) \downarrow$ | $\max(w)$ | $\Pr[w>10]$ |
|---|---|---|---|---|---|---|
| **Team size $n = 3$** | | | | | | |
| 2 | 1 | $0.42\pm0.03$ | 3.2 | 12.8 | 87 | 2.1% |
| 3 | 2 | $0.18\pm0.02$ | 6.1 | 38.5 | 312 | 5.8% |
| **Team size $n = 5$** | | | | | | |
| 2 | 1 | $0.41\pm0.03$ | 3.3 | 13.1 | 92 | 2.2% |
| 3 | 2 | $0.17\pm0.02$ | 6.2 | 41.2 | 338 | 6.1% |
| 4 | 3 | $0.08\pm0.01$ | 11.5 | 127.3 | 1,247 | 10.3% |
| 5 | 4 | $0.04\pm0.01$ | 21.8 | 385.1 | 4,582 | 15.2% |

### G.7. Cross-Generation Replacement: Qwen2.5 to Qwen3

Replacing an agent across model generations (e.g., Qwen2.5 $\to$ Qwen3) can be more tractable than cross-family replacement (e.g., Qwen $\to$ LLaMA), because the swap can often be performed under a largely shared chat protocol and tokenizer interface. However, even within the same model lineage, protocol-level mismatches can induce substantial occupancy shift in shared-context teams if left unaddressed. We summarize the main sources of mismatch we encountered and the corresponding mitigations.

**Canonical shared-context protocol.** In shared-context teams, seemingly "out-of-band" choices (system prompts, templates, tool-call formatting) become part of the effective state. We treat these choices as part of a canonical team protocol and enforce them uniformly for all agents before and after replacement.

**System prompt defaults.** Qwen2.5-Instruct deployments may prepend a non-empty default system prompt, while Qwen3 deployments may not. We remove this ambiguity by explicitly setting the system prompt for all agents (either fixed or explicitly empty).

**Reasoning-tag mode (e.g., `<think>... </think>`).** Some Qwen3 configurations optionally emit explicit reasoning-tag blocks. We enforce a uniform policy across the team; in our experiments, we disable reasoning-tag mode to maintain compatibility with Qwen2.5 agents.

**Tool-call serialization.** We introduce a lightweight adapter that normalizes tool-call outputs to a canonical JSON schema before writing to the shared context.

**Tokenizer interface and KL monitoring.** In our setup, the official Qwen2.5 and Qwen3 tokenizers produce identical token ID sequences on a held-out set of 1,000 shared-context strings, enabling direct reuse of our sampled token-level KL monitor for Stage-0 alignment.

**Stage-0 alignment procedure.** We sample 500 probe contexts from the pre-swap team's occupancy and fine-tune the new Qwen3 agent (via supervised distillation on the replaced agent's outputs) until $\widehat{D_{\text{KLtok}}} \le \delta$ on the probe distribution. After alignment, standard TeamTR updates resume; Proposition 3.8 applies to subsequent updates (the replacement step itself is not certified).

## H. Benchmarks

**AIME 2024 and AIME 2025.** The American Invitational Mathematics Examination (AIME) is an invitational math contest administered by the MAA for top performers on the AMC series. Each AIME form is a 15-problem, 3-hour exam with integer answers in $[0, 999]$ (often written with leading zeros), and calculators are prohibited. The problems span major pre-college topics (e.g., algebra, geometry, number theory, and combinatorics) and typically require multi-step reasoning and creative problem solving. We evaluate on the official 2024 and 2025 problem sets (AIME I and AIME II for each year; 30 problems/year, 60 total), grading by exact match on the final integer answer. We report pass@64 and avg@64.

*Table 11.* Ablation of protocol unification for Qwen2.5→Qwen3 replacement. Swap shock is the magnitude of the immediate accuracy drop at $k_{\text{swap}}$; lower is better. Final accuracy is measured at stage 40 on AIME24.

| Configuration | Swap Shock ($\downarrow$) | Final Acc. (%) |
|---|---|---|
| No unification (direct swap) | 18.3 | 58.2 |
| + Fixed system prompt | 12.1 | 67.5 |
| + Reasoning-tag mode disabled | 8.7 | 73.1 |
| + Tool-call adapter | 6.2 | 76.8 |
| + Stage-0 alignment (full) | 2.9 | **85.3** |

**MATH-500.** MATH-500 is a 500-problem held-out subset derived from the MATH benchmark (Hendrycks et al., 2021) (competition-style problems with LaTeX solutions and standardized final answers). The underlying MATH dataset covers seven subjects (prealgebra, algebra, number theory, counting and probability, geometry, intermediate algebra, and precalculus) and difficulty levels 1–5. This 500-problem split is widely used as a representative evaluation subset of MATH, and is commonly adopted in modern LLM math-evaluation pipelines. We use the provided ground-truth final answers for automatic grading after normalization (e.g., stripping formatting such as \boxed{} when applicable). We report pass@4 and avg@4.

**ZebraLogic.** ZebraLogic (Lin et al., 2025) is a logical reasoning benchmark of logic-grid (Einstein/Zebra) puzzles derived from constraint satisfaction problems (CSPs). It contains 1,000 automatically generated puzzles with *controllable and quantifiable* complexity, including a wide range of search-space sizes and logical constraint structures (e.g., measured via SMT-solver conflict statistics). Each instance provides a narrative and a set of clues; the model must output a complete, globally consistent assignment (we use the benchmark's structured output format for parsing). We report pass@64 and avg@64.

**AutoLogi.** AutoLogi (Zhu et al., 2025) is a bilingual (English/Chinese) *open-ended* logic-puzzle benchmark designed to avoid multiple-choice guessing effects. It reformulates problems from established logical-reasoning assessments (e.g., AR-LSAT and LogiQA) into constraint-based puzzles, and pairs each puzzle with *programmatic verification*: a format specification (JSON schema), a format verifier, a constraint verifier, and a traversal/enumeration procedure to validate solvability and filter invalid instances. The base AutoLogi benchmark (testing data, Stage 2) comprises 206 English and 139 Chinese puzzles; an augmented version expands the set via constraint expansion/reduction to create a range of difficulties. In our evaluation, we use the official verifiers for automatic grading and report pass@64 and avg@64.

**ARBench.** AR-Bench (ARBench) (Zhou et al., 2025) evaluates *active reasoning under incomplete information*, where an LLM must interact to acquire missing evidence before answering. AR-Bench contains 6,040 interactive puzzles spanning three task families: **Detective Cases (DC)** (interrogation-style cases with multiple suspects and noisy/role-dependent feedback), **Situation Puzzles (SP)** (lateral-thinking mysteries solved through yes/no questioning), and **Guessing Numbers (GN)** (deducing a hidden 4-digit code from structured match/misplacement feedback). For each episode, the model alternates between proposing an information-seeking question (or a guess) and receiving environmental feedback; success depends on both the quality of the question and the reasoning based on the acquired information. We allow up to 25 interaction rounds and report pass@25 and avg@25.

**PlanBench.** PlanBench (Valmeekam et al., 2023) is an extensible benchmark suite for evaluating planning and reasoning about actions and change, grounded in classical planning (IPC-style) domains represented in PDDL and rendered as natural-language prompts. PlanBench tests eight planning-related capabilities: plan generation, cost-optimal planning, plan verification, reasoning about plan execution, robustness to goal reformulation, plan reuse, replanning under unexpected events, and plan generalization. The benchmark is initialized with domains such as Blocksworld and Logistics (including obfuscated "mystery" variants), and provides automated executors/validators for scoring. In our experiments, we use the Blocksworld plan-generation subset and report pass@8 and avg@8.

