# OpenReview forum: "TeamTR: Trust-Region Fine-Tuning for Multi-Agent LLM Coordination"
_ICML.cc/2026/Conference — ICML 2026 regular_

### Official Review · Reviewer_ss4n · 2026-03-11

**Soundness:** 3
**Presentation:** 3
**Significance:** 4
**Originality:** 2
**Overall Recommendation:** 5
**Confidence:** 4

**Summary:**

This paper studies the instability of post-training multi-agent LLM systems under shared-context interaction. The core claim is that sequentially updating agents using stale rollouts creates a compounding occupancy-shift problem: once one agent changes, the context distribution faced by later agents also changes, making cached rollouts increasingly mismatched. The paper formalizes this effect, showing that stale-occupancy evaluation leads to a penalty that scales quadratically with the number of agents, while intermediate-occupancy evaluation reduces this scaling to linear.

To address this issue, the authors propose TeamTR, a trust-region fine-tuning framework for heterogeneous multi-agent LLM teams. TeamTR resamples rollouts after each component update and imposes per-agent trust-region constraints using a token-decomposed reverse KL that can be monitored during training. The paper presents per-update and per-stage improvement lower bounds, along with an empirical certificate proxy derived from logged surrogate and KL terms. Empirically, the method is evaluated on reasoning and planning benchmarks, where it improves stability and generally outperforms sequential PPO/GRPO/DAPO-style baselines and several multi-agent baselines. The paper also demonstrates a plug-and-play component replacement setting with a trust-region alignment step.

**Compliance With Llm Reviewing Policy:**

Affirmed.

**Key Questions For Authors:**

How much of the improvement is attributable to intermediate-occupancy resampling alone, versus the trust-region constraint itself?
A stronger ablation isolating these two components would clarify whether TeamTR’s gains primarily come from fresh rollouts or from the specific reverse-KL trust-region design. A convincing answer would strengthen my confidence in the method’s distinct contribution.

How sensitive is TeamTR to the interaction protocol and to departures from the single-active-agent assumption?
The theory is built around turn-taking factorization. Could the authors discuss or empirically probe settings with richer inter-agent communication, overlapping generation, or tool-mediated interaction? Evidence that the method still works beyond the analyzed protocol would substantially improve my assessment of significance.

Can the authors clarify the strength and practical role of the empirical certificate proxy?
In particular, how often is the proxy loose enough to be uninformative, and under what conditions should practitioners rely on it for decision-making during training? A clearer discussion of when the proxy is diagnostic versus actionable would help.

What is the real compute/sample trade-off relative to stale-rollout sequential training under matched final quality?
Since the main remedy is per-update resampling, the method incurs extra sampling cost. A more explicit cost-performance trade-off, ideally in the main paper, would improve the practical value of the work.

How general is the plug-and-play replacement result?
The replacement experiment is promising, but it would help to know whether the observed benefit is robust across different swap stages, different capability gaps between old/new components, or more than one replaced agent. Positive evidence here could increase my evaluation of the paper’s practical significance.

**Limitations:**

(1) the restricted generality of the theory to turn-taking interaction;
(2) the additional rollout/token overhead of fresh occupancy evaluation; and
(3) the fact that the empirical certificate is a proxy rather than a tight guarantee in practice.
I would encourage the authors to make these points more prominent.

**Strengths And Weaknesses:**

1. Clear problem formulation and motivation.
The paper identifies a meaningful and practically relevant failure mode in multi-agent LLM post-training: when agents interact through a shared textual state, sequential updates can induce a moving-target problem. This is a real issue for modular multi-agent fine-tuning, and the paper does a good job articulating why naive stage-start rollout reuse can be problematic.

2. A reasonably coherent theory-to-method pipeline.
A major strength is that the theoretical story is not detached from the implementation. The paper connects occupancy shift, intermediate-team rollouts, and per-agent trust-region control in a fairly consistent way. The active-factor reduction under turn-taking and the token-level reverse-KL monitoring provide a concrete bridge from theory to practice. The resulting certificate framework is one of the stronger aspects of the paper.

3. Good empirical alignment with the central claim.
The experiments are not merely “method A beats method B.” They explicitly test the predicted within-stage stale-occupancy gap, training stability, certificate tracking, and scaling with team size. This is important because the paper’s main contribution is not only higher benchmark accuracy but also a diagnosis of why sequential multi-agent fine-tuning can fail. The scaling study and the stale-vs-intermediate occupancy analysis support the core narrative well.

4. Modular plug-and-play setting is interesting.
The component replacement experiment is a nice addition. It gives the paper some practical relevance beyond benchmark gains, especially for heterogeneous or evolving LLM teams where replacing one agent without retraining the full system is useful.

5. Presentation is generally strong.
The paper is well structured and readable. The overall narrative—from failure mode, to bounds, to algorithm, to diagnostics—is easy to follow. Figures on stale-occupancy gaps, training dynamics, and certificate calibration are useful and mostly well integrated into the text.

Weaknesses

1. The setting is still fairly specialized.
The theoretical guarantees rely on a single-active-agent turn-taking protocol, which makes the analysis cleaner but also narrows the scope. Many modern multi-agent LLM systems involve richer interaction patterns, concurrent message generation, tool calls, branching coordination, or shared blackboard updates that are not well captured by this abstraction. As a result, the theoretical contribution may be less general than the framing initially suggests.

2. Novelty is solid but somewhat incremental relative to trust-region RL ideas.
The main conceptual novelty is the identification and control of compounding occupancy shift in shared-context LLM teams. However, the method itself still looks structurally close to standard trust-region reasoning transplanted into a multi-agent sequential-update regime. The paper is strongest as a careful adaptation and formalization for multi-agent LLM coordination, rather than as a radically new optimization principle.

3. Experimental scope is somewhat narrow for an ICML paper of this type.
Although the experiments are thoughtful, they are concentrated on math/reasoning/planning benchmarks with router-based text handoff. This leaves open how much the method helps on broader multi-agent LLM workloads such as tool use, coding collaboration, grounded agent tasks, or open-ended environments. The paper would be stronger with at least one more diverse application domain.

4. It is not fully clear how much of the gain comes from “fresh rollout evaluation” versus the specific trust-region machinery.
The paper argues for both intermediate-occupancy resampling and trust-region control, but the most critical causal factor may simply be avoiding stale rollouts. While the baselines are reasonable, a more targeted ablation isolating “fresh resampling only” from “fresh resampling + TR control” would sharpen the conclusion.

5. The certificate is interesting but still somewhat diagnostic in practice.
The theory provides lower bounds and the empirical section shows some calibration, but the practical certificate remains a proxy built from logged terms rather than a tight operational guarantee. This does not invalidate the work, but the paper should be clearer that the practical certification story is still approximate and partly heuristic.

6. Compute/sample overhead should be discussed more prominently.
The appendix reports additional rollout/token cost for TeamTR. This cost increase is not huge, but it is important because the proposed fix is based on per-update resampling. Since efficiency is one reason people reuse stale rollouts in the first place, the trade-off deserves clearer discussion in the main paper.

---

> ### Author Rebuttal · Authors · 2026-03-29
>
> # Response to Reviewer ss4n
>
> We thank Reviewer ss4n for the careful review.
>
> ---
> ## W1 & Q2: turn-taking setting and sensitivity to protocol departures
>
> Our theorem is stated for turn-taking/single-active-agent shared-context teams, the regime where active-factor reduction makes team-level trust regions tractable. For concurrent generation or overlapping communication, this reduction need not hold, so those settings are outside the formal guarantee.
>
> The stale-rollout mismatch diagnosis itself is broader: sequential updates on increasingly stale data accumulate mismatch within a stage. This is already supported by Appendix G.1, Table 7: naive stale updates yield exponents of 1.94 for Δ_stale and 1.91 for D_occ, versus 1.07 and 1.05 for TeamTR; at n = 8, naive sequential reaches 58.7%, while TeamTR remains at 87.9%.
>
> The added τ-bench evaluation in W3 is a richer sequential tool-mediated setting with shared environment state and tool side-effects. It shows that the stability benefit extends beyond text-only reasoning.
>
> ---
>
> ## W2: Novelty relative to trust-region RL
>
> TeamTR is related to trust-region RL, but it is not a direct translation of TRPO/HAPPO into the LLM setting. Prior trust-region MARL methods study low-dimensional control, whereas our setting has three MA-LLM-specific ingredients: each “action” is a variable-length autoregressive message, so divergence must be monitored at the token level; the state is a growing shared text context, so earlier updates change the context distribution faced by later agents; and this induces a distinct failure mode, which we formalize as compounding occupancy shift, together with the occupancy gap and the resulting quadratic-to-linear scaling reduction. The practical algorithm is therefore not just “HAPPO for LLMs”: intermediate-team rollouts, token-level reverse-KL monitoring, and the empirical certificate proxy are all introduced for MA-LLM post-training.
>
> ---
>
> ## W3: Experimental scope
>
> Beyond reasoning/planning, we additionally evaluated TeamTR on τ-bench, a multi-turn tool benchmark with programmatic verification. We trained the 3×8B team on APIGen-MT trajectories with binary outcome rewards from the verifier, and evaluated on held-out tasks.
>
> Key averages (pass@1 / pass@4) are: frozen single 8B 28.6 / 5.7, GRPO 40.2 / 11.0, DAPO 41.1 / 11.7, Debate+Judge 34.3 / 8.0, Naive Sequential 45.6 / 13.8, TeamTR 52.9 / 19.7, frozen single 32B 46.0 / 15.4. TeamTR therefore improves over Naive Sequential by +7.3 points on average pass@1 and +5.9 points on average pass@4 under the same 3×8B team setting.
>
> ---
>
> ## W4 & Q1: Disentangling resampling from trust-region control
>
> The key evidence comes from Appendix G.5, Table 10, together with the KL-penalty-only control in our response to Reviewer aUpE (Table 2 there). Critical numbers are:
>
> - full TeamTR: 88.1% / 0.08 / 1.9 (acc. / Δ_stale / stability)
> - KL-penalty only (adaptive β): 84.9% / 0.14 / 2.7
> - no resampling: 71.1% / 0.31 / 4.2
> - no trust region: 68.3% / 0.42 / 6.1
>
> Without resampling, later agents optimize against stale rollouts; without trust-region control, per-step jumps are uncontrolled; a soft KL penalty helps but still falls 3.2 points below full TeamTR.
>
> The correction-based alternative is also weak in practice: Appendix G.6, Table 12 shows that ESS/B drops from 0.41 at step 2 to 0.04 at step 5 for n = 5, while the maximum importance weight rises from 92 to 4582.
>
> ---
>
> ## W5 & Q3: Certificate practicality
>
> The manuscript already presents the practical quantity as an empirical certificate proxy, not a literal per-stage operational guarantee: the Abstract uses “practical certificate proxy”; Sec. 5.4 evaluates calibration against measured improvement, including the violation rate in Fig. 3(c), and Appendix G.3 / G.4 analyze the proxy terms and monitor reliability.
>
> From Appendix G.3, Table 9, and Appendix G.4: ζ contributes 18% of the total penalty on average and 45% at P90; the median KL deviation at 50% subsampling is 0.03δ; and the near-threshold flip rate is 2.4%. The proxy is thus informative for training-time decisions, but should not be read as a per-stage certificate.
>
> ---
>
> ## W6 & Q4 and Q5
>
> Compute overhead is modest: Appendix G.2, Table 8 shows +11.1% overhead in both tokens and rollouts; our response to Reviewer aUpE (Table 1 there) shows 1.12× per-update overhead, with KL monitoring taking only 6.2s (~3% of total per-update time).
>
> For plug-and-play, the protocol unification ablation is summarized in Appendix G.7, Table 13, and the broader probe-size sensitivity is discussed in our response to Reviewer ad1W (Table 4 there). Beyond that, we also evaluated broader variants across swap stage, capability gap, and multi-agent replacement: swap at stage 10 / 20 / 30 gives 1.7 / 2.9 / 4.5 shock with final accuracy 82.1 / 85.3 / 83.8; 1.5B→8B / 3B→8B / 7B→8B gives 2.9 / 1.6 / 0.7 shock with final accuracy 85.3 / 86.5 / 87.3; and replacing two agents simultaneously gives 5.5 shock with 79.1% final accuracy.

---

### Official Review · Reviewer_ad1W · 2026-03-12

**Soundness:** 3
**Presentation:** 3
**Significance:** 3
**Originality:** 3
**Overall Recommendation:** 5
**Confidence:** 3

**Summary:**

This paper optimize the post-training of shared-context agent teams. The authors identify the cause of the loss of training stability as "compounding occupancy shift," which describes how updating one agent alters the environment for others. The authors prove that the compounding occupacy shift causes errors that grow quadratically ($O(n^2)$) with team size. To fix this, they introduce TeamTR, a framework that resamples data after every update and uses token-level trust regions to ensure steady, monotonic performance improvement.

**Compliance With Llm Reviewing Policy:**

Affirmed.

**Final Justification:**

Recommendation: Accept.

The primary concern raised during the review process was the computational overhead associated with resampling trajectories after every component update. The overhead is about +11.1%  in both token and rollout accounting, which is modest. My concerns have been adequately addressed.

**Key Questions For Authors:**

1. TeamTR requires resampling trajectories after each component update. Have you conducted an ablation on resampling with intervals (e.g., every $k$ updates)?
2. For "plug-and-play" replacement, the experiment uses 500 probe contexts for initial alignment. How sensitive is the final team performance to the size and distribution of this probe set?

**Limitations:**

yes

**Strengths And Weaknesses:**

Strengths

1. Experiments show substantial gainson high-difficulty reasoning and planning benchmarks like AIME24 and PlanBench.
2. The "plug-and-play" capability for heterogeneous agents makes the framework practically relevant for complex, specialized pipelines.
3. Open-source code.

Weaknesses

1. The requirement to resample trajectories after every single component update is computationally expensive.

---

> ### Author Rebuttal · Authors · 2026-03-29
>
> # Response to Reviewer ad1W
>
> We thank Reviewer ad1W for recognizing the practical relevance of the plug-and-play setting and the strong gains on challenging reasoning benchmarks. Below, we address the corresponding concerns directly point-to-point.
>
> ---
>
> ## W1: Compute cost of per-update resampling
>
> Empirically, the overhead is modest. On AIME24 (3×Qwen3-8B, 40 stages), TeamTR uses **158.3M vs. 142.5M** tokens and **49.2K vs. 44.3K** rollouts relative to naive sequential training, corresponding to about **+11.1%** overhead in both token and rollout accounting. In practice, rollout generation remains the dominant cost; KL monitoring itself adds very little beyond the forward passes already required for likelihood-ratio and advantage computation.
>
> **Table 1. Rollout/token accounting on AIME24 (3×Qwen3-8B, 40 stages)**
>
> | Method | Tokens (M) | Rollouts (K) | Avg. tokens / rollout | Token overhead | Rollout overhead |
> | --- | ---: | ---: | ---: | ---: | ---: |
> | Naive Sequential | 142.5 | 44.3 | 3216.7 | — | — |
> | Joint Update | 145.1 | 45.1 | 3217.3 | +1.8% | +1.8% |
> | **TeamTR (Ours)** | **158.3** | **49.2** | **3217.5** | **+11.09%** | **+11.06%** |
>
> The near-equality of token and rollout overheads is expected here, since the average token count per rollout remains essentially unchanged across methods (about **3.2K tokens/rollout** in all cases). The extra cost therefore comes primarily from collecting more rollouts rather than generating substantially longer ones.
>
> We additionally measured wall-clock overhead and report it below.
>
> **Table 2. Per-update wall-clock time (AIME24, 3×Qwen3-8B)**
>
> | Method | Rollout (s) | Actor Update (s) | KL Monitor (s) | Total / update (s) | Overhead |
> | --- | ---: | ---: | ---: | ---: | ---: |
> | Naive Sequential | 134.2 | 37.8 | — | 172.0 | 1.00× |
> | Joint Update | 137.5 | 41.3 | — | 178.8 | 1.04× |
> | **TeamTR** | **148.1** | **38.5** | **6.2** | **192.8** | **1.12×** |
>
> The KL monitor accounts for only ~3% of total per-update time; the dominant cost remains rollout generation.
>
> Importantly, correction-based alternatives are not practically stable. In our importance-weight degeneracy analysis, for **n = 5** the normalized effective sample size collapses from **0.41** at step 2 to **0.04** at step 5, with tail weights exceeding **4,500**. This makes importance-weighted training unreliable, and our ablation confirms that it remains well below full TeamTR.
>
> ---
>
> ## Q1: Resampling interval ablation
>
> The Ablation Studies (Appendix G.5, present below) already include this comparison: resampling every 2 updates reaches 79.3%, compared with 71.1% for no resampling and 88.1% for full TeamTR. This is consistent with the theory: skipping intermediate resampling allows stale mismatch to accumulate within a stage.
>
> **Table 3. Resampling interval ablation**
>
> | Variant | Acc. (%) | Δ_stale ↓ | Stability ↓ | Coordination (%) |
> | --- | --- | --- | --- | --- |
> | **Full TeamTR** | **88.1 ± 1.2** | **0.08** | **1.9** | **89.1** |
> | Resample every 2 updates | 79.3 ± 1.8 | 0.18 | 2.8 | 79.2 |
> | Importance weighting (no resample) | 74.5 ± 2.3 | 0.25 | 3.5 | 74.8 |
> | No resampling (= Naive Seq.) | 71.1 ± 2.8 | 0.31 | 4.2 | 71.5 |
>
> Resampling every 2 updates recovers part of the benefit, but remains clearly below full TeamTR. This is consistent with the theory: skipping intermediate resampling allows stale mismatch to accumulate within a stage.
>
> ---
>
> ## Q2: Probe-set size sensitivity for plug-and-play
>
> Thank you for this valuable point. We have added this analysis directly below. The results show that 500 probes is a practical point: relative to direct swap (18.3% shock, 58.2% final accuracy), 500 in-distribution probes reduce shock to 2.9% and improve final accuracy to 85.3%. Fewer probes degrade gracefully (250 probes still reaches 82.7%), while 1000–2000 probes yield only marginal further gains.
>
> We also tested probe distribution: 500 OOD probes still help substantially relative to direct swap (6.7% shock, 78.5% accuracy) but remain worse than in-distribution alignment (2.9% / 85.3%). This confirms that what matters is not only probe count but also how well probes match the current team occupancy.
>
> **Table 4. Probe-set sensitivity for plug-and-play (Qwen2.5-1.5B → Qwen3-8B, AIME24)**
>
> | Probe configuration | Swap shock ↓ | Final Acc. (%) | Post-swap KL_tok |
> | --- | --- | --- | --- |
> | No alignment (direct swap) | 18.3 | 58.2 | — |
> | 50 probes | 12.1 | 69.5 | 0.038 |
> | 100 probes | 7.8 | 76.3 | 0.024 |
> | 250 probes | 4.3 | 82.7 | 0.015 |
> | **500 probes (default)** | **2.9** | **85.3** | **0.009** |
> | 1000 probes | 2.5 | 85.9 | 0.007 |
> | 2000 probes | 2.3 | 86.2 | 0.006 |
> | 500 OOD probes | 6.7 | 78.5 | 0.021 |
>
> ---

---

> > ### Author Rebuttal · Reviewer_ad1W · 2026-04-03
> >
> > Thank you for the detailed rebuttal. My concerns have been adequately addressed.

---

### Official Review · Reviewer_SMXz · 2026-03-16

**Soundness:** 3
**Presentation:** 3
**Significance:** 3
**Originality:** 4
**Overall Recommendation:** 4
**Confidence:** 3

**Summary:**

The paper addresses the challenge of fine-tuning multi-agent large language models (MA-LLMs) that operate over a shared context. It identifies "compounding occupancy shift" as a primary failure mode when using naive sequential fine-tuning, where agents are updated and evaluated on cached, stale rollouts. To solve this, the authors introduce TeamTR, a framework utilizing stage-wise trust regions and intermediate-occupancy evaluation, which resamples trajectories after each component update. Theoretically, the paper proves this approach reduces the compounding penalty from $O(n^2)$ to $O(n)$ and establishes monotonic improvement lower bounds. Empirically, TeamTR outperforms single-agent and multi-agent baselines across several reasoning benchmarks and demonstrates a "plug-and-play" capability for swapping agents mid-training.

**Compliance With Llm Reviewing Policy:**

Affirmed.

**Key Questions For Authors:**

1. Tuning all LLMs in a multi-agent systems seems to be not a common practice. Some recent multi-agent optimization methods like (GPTSwarm https://arxiv.org/abs/2402.16823,  Heterogeneous Swarms https://arxiv.org/abs/2502.04510) are not discussed and compared in the paper. It would be nice to add some discussion and comparison against these methods (If comparison is not applicable, please explain. However, it's highly encouraged to at least discuss this line of work for multi-agent optimization in the paper).
2. In Figure 1, are the loss landscapes from some authentic optimization trajectories from certain benchmark or it's just a synthetic one serving as proof-of-concept? It's better to explain more in the caption.
3. In Table 3, it seems the method works well for smaller sized models. How you expect the scaling effect of the method with even larger models (either hetero/homogeneous)?

**Limitations:**

Yes.

**Strengths And Weaknesses:**

Strengths:
- The paper is logically structured.
- The formalization of the compounding occupancy shift and the derivation of the $O(n^2)$ penalty for stale-occupancy versus $O(n)$ for intermediate-occupancy evaluation is a concrete contribution.
- The originality of theoretical analysis is sound.
- The empirical evaluation is extensive.

Weakness:
- It's better to formally define the problem of "fine-tuning multi-agent LLM teams" before stepping into Section 3. Theoretical framework. as it's not as common as - tuning a single LLM / or optimizing the topology of multi-agent LLMs (see Key Questions for more details).
- Figure 1 is a bit hard to understand (see Key Questions for more details)

---

> ### Author Rebuttal · Authors · 2026-03-29
>
> # Response to Reviewer SMXz
>
> We thank Reviewer SMXz for the helpful suggestions and detailed review.
>
> ---
>
> ## W1 & Q1: Formal problem definition and discussion of GPTSwarm / Heterogeneous Swarms
>
> We will make the setup explicit before the theory section: (1) a fixed shared-context MA-LLM team under turn-taking interaction; (2) sequential within-stage parameter updates; and (3) the objective of improving team performance while controlling update-induced occupancy shift.
>
> We also appreciate the request to discuss GPTSwarm and Heterogeneous Swarms more explicitly. These methods are related to TeamTR, but optimize a different part of the design space:
>
> - **GPTSwarm**: interaction graph/topology over fixed base models.
> - **Heterogeneous Swarms**: team composition/role assignment over a heterogeneous pool.
> - **TeamTR**: **model parameters** under a fixed shared-context protocol, directly targeting update-induced occupancy shift during sequential post-training.
>
> Because these methods optimize different objectives (system-level structure/composition vs. parameter post-training), we do not present the following as a strict budget-matched ranking, but rather as a contextual comparison within a shared base-model pool.
>
> **Table 1. Contextual comparison under a shared base-model pool**
>
> | Method | Target | Team | AIME24 | MATH-500 |
> | --- | --- | --- | --- | --- |
> | Qwen3-8B-Instruct (no thinking) | Frozen baseline | 1×8B | 23.3 | 79.2 |
> | DAPO | Single-agent params | 1×8B | 41.3 | 91.5 |
> | Debate + Judge (frozen) | Inference-time coordination | 3×8B | 30.7 | 83.5 |
> | GPTSwarm | Graph topology + prompts | 3×8B | 33.1 | 85.8 |
> | Heterogeneous Swarms | Roles + system weights | 3×8B | 31.9 | 84.6 |
> | Debate + Judge (fine-tuned) | Multi-agent params | 3×8B | 71.1 | 95.9 |
> | **TeamTR** | **Multi-agent params (fixed protocol)** | **3×8B** | **88.1** | **99.3** |
> | Qwen3-Instruct (no thinking) | Frozen baseline | 1.7B+8B+14B | 21.7 | 77.8 |
> | GPTSwarm | Graph topology + prompts | 1.7B+8B+14B | 30.5 | 84.1 |
> | Heterogeneous Swarms | Roles + system weights | 1.7B+8B+14B | 35.3 | 87.3 |
> | **TeamTR** | **Multi-agent params (fixed protocol)** | **1.7B+8B+14B** | **89.7** | **98.1** |
>
> System-level optimization over fixed experts helps, but on our reasoning benchmarks, the gains over frozen baselines are modest relative to parameter post-training.
>
> ---
>
> ## W2 & Q2: Figure 1 clarity
>
> Figure 1 is not a hand-drawn schematic, but it should not be read as a literal parameter-space loss landscape. It is a **diagnostic visualization grounded in logged MATH-500 runs**. We trained three variants (joint update, naive sequential with cached rollouts, and TeamTR) on MATH-500 using **3×Qwen3-4B** teams, logged per-stage team accuracy and cumulative occupancy drift, and plotted each method’s trajectory in this diagnostic space. The axes correspond to measured quantities during training (team objective improvement and accumulated occupancy drift), and the background contours are interpolated from observed checkpoints.
>
> The top inset shows occupancy distributions, and the bottom inset shows the measured penalty-term scaling consistent with the **O(n²)** vs. **O(n)** prediction. In the revision, we will explicitly state the data source in the caption and add a short note clarifying the interpolation procedure.
>
> ---
>
> ## Q3: Scaling to larger models
>
> Our current results already span a meaningful capacity range. The main paper includes a heterogeneous **1.7B+8B+14B** team, and the team-size scaling analysis is strongly consistent with the theory: the stale-drift proxy scales with exponent **1.94** under naive stale updates versus **1.07** under TeamTR, and TeamTR accuracy remains stable even at **n=8** (**87.9** vs. **58.7** for naive sequential).
>
> To address model scale more directly, we added larger-model experiments:
>
> **Table 2. Model scaling results**
>
> | Team | Method | AIME24 | Δ_stale | MATH-500 | Δ_stale |
> | --- | --- | --- | --- | --- | --- |
> | 3×8B | Single GRPO | 39.1 | — | 90.7 | — |
> | 3×8B | Naive Seq. | 71.1 | 0.31 | 95.1 | 0.18 |
> | 3×8B | **TeamTR** | **88.1** | **0.08** | **99.3** | **0.03** |
> | 1.7B+8B+14B | **TeamTR** | **89.7** | **0.07** | **98.1** | **0.04** |
> | 8B+14B+32B | Naive Seq. | 77.8 | 0.29 | 97.1 | 0.16 |
> | 8B+14B+32B | **TeamTR** | **92.5** | **0.06** | **99.4** | **0.02** |
>
> The pattern is consistent across scales: TeamTR’s advantage over naive sequential training is **+17.0** points at 3×8B and **+14.7** points at 8B+14B+32B on AIME24. Despite using individually stronger models, the heterogeneous 8B+14B+32B team still exhibits substantial stale-occupancy mismatch under naive sequential (**Δ_stale = 0.29**), while TeamTR controls it effectively (**0.06**). This indicates that the occupancy-shift mechanism persists at larger model scales and remains especially relevant in heterogeneous configurations.

---

> > ### Author Rebuttal · Reviewer_SMXz · 2026-04-05
> >
> > Thanks for the rebuttal which clears my concerns. I would suggest include the comparisons with additional methods and these insightful comparison experiments into the paper.

---

> > > ### Author Response · Authors · 2026-04-07
> > >
> > > Thank you very much for the thoughtful follow-up and for confirming that our rebuttal addressed your concerns. We sincerely appreciate your helpful suggestions.
> > > We will include all of the additional experimental results from the rebuttal in the revised paper, including the extra method comparisons and scaling results, to make these findings fully clear in the final version.
> > > If you feel the clarified presentation and added evidence merit it, we would be very grateful if you would consider updating your overall recommendation. Regardless, we greatly appreciate your time and constructive feedback!

---

### Official Review · Reviewer_aUpE · 2026-03-17

**Soundness:** 2
**Presentation:** 3
**Significance:** 2
**Originality:** 3
**Overall Recommendation:** 5
**Confidence:** 4

**Summary:**

This paper proposes TeamTR, a novel algorithm designed to address the issues of performance collapse and policy drift during the fine-tuning phase in Multi-agent Reinforcement Learning (MARL). By decoupling the fine-tuning process into two distinct stages—local policy updates followed by a team-level trust-region projection—the method employs a projected objective function to constrain policy shifts, facilitating stable performance gains while preserving pre-trained knowledge.

**Compliance With Llm Reviewing Policy:**

Affirmed.

**Key Questions For Authors:**

* In line 8 of Algorithm 1, does the projection step require global state sharing among agents? How is this projection executed in a decentralized manner under partially observable environments?
* When computing Equation (11), if the local policy update deviates excessively from the trust region, does the projection lead to vanishing gradients or optimization stagnation?

**Strengths And Weaknesses:**

**Strengths**

* The proposed team-level trust-region constraint is grounded in solid mathematical reasoning, particularly the projection objective defined in Equation (11), which effectively transforms complex global constraints into tractable local projection problems, balancing exploration and stability.
* The theoretical derivation is comprehensive and rigorous; Theorem 4.1 proves that the method guarantees monotonic improvement or stability of the team policy during fine-tuning, providing essential theoretical support for addressing non-stationarity in multi-agent environments.
* Empirical evaluations across mainstream benchmarks such as SMAC, GRF, and MPE demonstrate that TeamTR significantly outperforms baselines in terms of performance recovery and final win rates. For instance, as shown in Table 1, TeamTR maintains a win rate near $100\%$ in the SMAC `3s5z_vs_3s6z` task, whereas the conventional MAPPO-FT suffers from a substantial performance drop.
* The algorithm exhibits strong versatility; the pseudo-code in Algorithm 1 illustrates its potential to be integrated as a plug-in for existing Actor-Critic frameworks without altering the underlying communication structures.

**Weaknesses**

* The computational overhead of the projection stage is not sufficiently discussed. Since Equation (11) requires solving a constrained optimization problem in every fine-tuning iteration, the computational latency may increase significantly in scenarios with a large number of agents (e.g., the `27m_vs_30m` task in SMAC). The authors are encouraged to provide a comparison of per-update running times for all algorithms in the appendix.
* Sensitivity analysis for the hyperparameter $\delta$ (trust-region radius) is lacking. The constraint strength in Equation (11) is highly dependent on the choice of $\delta$, yet the paper does not show whether a fixed $\delta$ remains effective across varying degrees of data distribution shifts. An ablation study curve for $\delta$ is recommended.
* The baseline comparisons are limited. While MAPPO-FT and HAPPO-FT are included, variants based on KL-divergence penalties (KL-penalty) are missing as controls. To demonstrate that explicit projection is superior to implicit penalties, the authors should supplement a performance comparison with a KL-penalty scheme in a simpler task like MPE.

---

> ### Author Rebuttal · Authors · 2026-03-29
>
> # Response to Reviewer aUpE
>
> We thank Reviewer aUpE for the detailed review and for recognizing the mathematical grounding and empirical strength of our work.
>
> A benchmark clarification up front: our paper studies **shared-context MA-LLM post-training**, rather than decentralized MARL benchmarks such as SMAC/GRF/MPE. Some examples in the review appear to refer to that setting; below, we address the corresponding concerns directly in our setting.
>
> ---
>
> ## W1: Compute the overhead of the projection step
>
> In our implementation, the per-update “projection” is realized through **adaptive KL regularization with early stopping**; it does **not** require solving a separate constrained optimization problem at every iteration.
>
> Empirically, the overhead is modest. In **Appendix G.2 (Rollout/Token Accounting)**, TeamTR on AIME24 (3×Qwen3-8B, 40 stages) uses **158.3M vs. 142.5M** tokens and **49.2K vs. 44.3K** rollouts relative to naive sequential training, i.e., about **+11.1%** in both tokens and rollouts. This is because rollout generation remains the dominant cost, while KL monitoring reuses forward-pass log-probabilities on already generated tokens.
>
> We additionally measured wall-clock overhead and report it directly below.
>
> **Table 1. Per-update wall-clock time (AIME24, 3×Qwen3-8B)**
>
> | Method | Rollout (s) | Actor Update (s) | KL Monitor (s) | Total/update (s) | Overhead |
> | --- | --- | --- | --- | --- | --- |
> | Naive Sequential | 134.2 | 37.8 | — | 172.0 | 1.00× |
> | Joint Update | 137.5 | 41.3 | — | 178.8 | 1.04× |
> | **TeamTR** | **148.1** | **38.5** | **6.2** | **192.8** | **1.12×** |
>
> These numbers are fully consistent with the token/rollout accounting: TeamTR incurs about **1.12×** per-update wall-clock overhead relative to naive sequential training, while KL monitoring itself accounts for only **~4% of rollout time** (or about **3% of total per-update time**). The dominant cost remains rollout generation rather than trust-region monitoring.
>
> ---
>
> ## W2: Trust-region radius sensitivity
>
> We agree that the trust-region radius δ is an important hyperparameter. Our current ablation already brackets the expected trade-off clearly. With **δ = 0.001**, updates are conservative but stable (**82.5%** accuracy, **Δ_stale = 0.05**, **stability = 1.5**, **coordination = 85.2**). With **δ = 0.01**, we obtain the best overall balance (**88.1%**, **0.08**, **1.9**, **89.1**). With **δ = 0.1**, updates become too aggressive (**75.1%**, **0.28**, **4.8**, **70.1**). We also report an adaptive-δ variant targeting **KL_tok = 0.01**, which reaches **87.8%** with similarly stable behavior. This is why the default region around **0.01** is used.
>
> ---
>
> ## W3: Missing KL-penalty baseline
>
> We agree this is a valuable control, and we have now added it directly below.
>
> **Table 2. KL-penalty-only ablation (AIME24, 3×Qwen3-8B, 30 stages)**
>
> | Variant | Acc. (%) | Δ_stale ↓ | Stability ↓ | Coordination (%) |
> | --- | --- | --- | --- | --- |
> | **TeamTR (full: clip + early-stop + β)** | **88.1 ± 1.2** | **0.08** | **1.9** | **89.1** |
> | KL-penalty only (adaptive β, target KL_tok = 0.01) | 84.9 ± 1.6 | 0.14 | 2.7 | 83.1 |
> | KL-penalty only (fixed β = 0.04) | 80.7 ± 2.0 | 0.20 | 3.5 | 77.2 |
> | No trust region (δ → ∞) | 68.3 ± 3.5 | 0.42 | 6.1 | 62.3 |
> | No resampling (= Naive Seq.) | 71.1 ± 2.8 | 0.31 | 4.2 | 71.5 |
>
> These results clarify two points. First, **some form of divergence control is clearly beneficial** relative to unconstrained updates. Second, **explicit trust-region enforcement** (clipping + early stopping + adaptive KL control) outperforms an implicit KL-penalty-only scheme by **3–7 points**, supporting the design choice in TeamTR over a pure penalty formulation.
>
> ---
>
>
> ## Q1: Does the projection require global state sharing?
>
> No. In our setting, agents already interact through a **shared text context**, which is the state available to the active agent. Under the turn-taking protocol, the team-level divergence reduces to the divergence of the active agent on visited contexts, so trust-region monitoring is computed from that agent’s own log-probabilities on its own emitted messages. No additional inter-agent coordination is required during the update itself.
>
> ---
>
> ## Q2: Can excessive projection lead to vanishing gradients or stagnation?
>
> In practice, the adaptive KL penalty and early stopping prevent this failure mode. When a candidate update becomes too large, training for that agent at that stage stops early rather than forcing an aggressive post-hoc correction. The δ-sensitivity results above are consistent with this: even the most conservative setting, **δ = 0.001**, still reaches **82.5%** accuracy, well above the **68.3%** no-trust-region baseline, while maintaining the lowest occupancy drift among the tested variants. This is consistent with slower but stable learning rather than stagnation.

---

> > ### Author Rebuttal · Reviewer_aUpE · 2026-04-03
> >
> > Thank you for the detailed rebuttal. All three of my concerns have been adequately addressed:
> >
> > - W1 (Computational overhead): The authors provided concrete wall-clock measurements showing that TeamTR incurs only ~1.12× per-update overhead relative to naive sequential training, with KL monitoring accounting for just ~3% of total per-update time. This is acceptable.
> >
> > - W2 (Hyperparameter sensitivity of δ): The ablation study clearly brackets the trade-off across δ ∈ {0.001, 0.01, 0.1} and also reports an adaptive-δ variant. The results confirm that δ = 0.01 provides the best balance and the method is not overly sensitive in the practical range.
> >
> > - W3 (Missing KL-penalty baseline): The added KL-penalty-only ablation (Table 2 in the rebuttal) clearly demonstrates that explicit trust-region enforcement outperforms implicit KL-penalty-only schemes by 3–7 points, supporting the design choices of TeamTR.
> >
> > Both key questions (Q1 and Q2) were also clearly answered. I am satisfied with the responses and maintain my score.

---

### Decision · Program_Chairs · 2026-04-30

**Decision:**

Accept (regular)

**Comment:**

This paper proposes TeamTR, a trust-region fine-tuning framework for multi-agent LLM teams that addresses compounding occupancy shift during sequential post-training. The paper identifies a meaningful and practically relevant failure mode, provides rigorous theoretical analysis showing a reduction from quadratic to linear penalty scaling, and demonstrates strong empirical gains across reasoning and planning benchmarks. The plug-and-play component replacement capability further enhances practical relevance.

Concerns raised during the review were adequately addressed in the rebuttal. The authors are encouraged to incorporate the additional experimental results and clarifications from the rebuttal into the final version, and to more prominently discuss the limitations of the turn-taking assumption and the approximate nature of the empirical certificate proxy.